# On the Identifiability and Interpretability of Gaussian Process Models

**Jiawen Chen**
Department of Biostatistics
University of North Carolina at Chapel Hill
jiawenn@email.unc.edu

**Wancen Mu**
Department of Biostatistics
University of North Carolina at Chapel Hill
wancen@live.unc.edu

**Yun Li**
Department of Biostatistics
University of North Carolina at Chapel Hill
yun_li@med.unc.edu

**Didong Li**
Department of Biostatistics
University of North Carolina at Chapel Hill
didongli@unc.edu

## Abstract

In this paper, we critically examine the prevalent practice of using additive mixtures of Matérn kernels in single-output Gaussian process (GP) models and explore the properties of multiplicative mixtures of Matérn kernels for multi-output GP models. For the single-output case, we derive a series of theoretical results showing that the smoothness of a mixture of Matérn kernels is determined by the least smooth component and that a GP with such a kernel is effectively equivalent to the least smooth kernel component. Furthermore, we demonstrate that none of the mixing weights or parameters within individual kernel components are identifiable. We then turn our attention to multi-output GP models and analyze the identifiability of the covariance matrix $A$ in the multiplicative kernel $K(x,y) = AK_0(x,y)$, where $K_0$ is a standard single output kernel such as Matérn. We show that $A$ is identifiable up to a multiplicative constant, suggesting that multiplicative mixtures are well suited for multi-output tasks. Our findings are supported by extensive simulations and real applications for both single- and multi-output settings. This work provides insight into kernel selection and interpretation for GP models, emphasizing the importance of choosing appropriate kernel structures for different tasks.

## 1 Introduction

Gaussian processes (GPs) have emerged as a powerful and popular tool in machine learning, spatial statistics, functional data analysis, etc., due to their versatility, flexibility, and interpretability as a nonparametric method (Rasmussen and Williams, 2006; Banerjee et al., 2014). GPs provide an intuitive means of modeling uncertainty, allowing the generation of predictive distributions for unseen data points without requiring explicit model specification. Consequently, GPs have found applications in a variety of contexts.

The key to harnessing the power of GPs lies in the choice of kernel functions, also known as covariance functions. Rather than specifying a model, the GP framework revolves around selecting an appropriate kernel, transforming the problem into one of parameter estimation. Over time, researchers have developed various kernels, each tailored to specific scenarios, such as spatial data, time series data, and others (Genton, 2001). For instance, for spatiotemporal, a variety of kernels have been developed to capture unique characteristics and patterns (Gneiting, 2002; Stein, 2005).

The Radial Basis Function (RBF) kernel and Matérn kernels serve as prime examples of the diversity within the kernel family (Cressie and Wikle, 2015). The RBF kernel, renowned for its infinite differentiability, yields smooth functions, while Matérn kernels control the degree of smoothness by a kernel parameter, thereby accommodating both smooth and nonsmooth functions (Stein (1999)).

37th Conference on Neural Information Processing Systems (NeurIPS 2023).

With each kernel bearing its own set of advantages and disadvantages, kernel selection, as a nontrivial task, often demands extensive domain knowledge.

Inspired by the potential of harnessing the strengths of multiple kernels concurrently, researchers have ventured into methods that involve kernel combinations, such as spectral mixtures (Wilson and Adams, 2013; Samo and Roberts, 2015; Remes et al., 2017), addition and/or multiplication of kernels (Duvenaud et al., 2011) such as mixture of RBF (Duvenaud et al., 2013), mixture of RBF, periodic (Per), linear (Lin), and rational quadratic (RQ) (Kronberger and Kommenda, 2013), mixture of RBF, RQ, Matérn and Per (Cheng et al., 2019; Verma and Engelhardt, 2020), mixture of RBF, Matérn 1/2, 3/2, 5/2, as well as more sophisticated methods like Neural Kernel Networks (NKN,Sun et al. (2018) and Automatic Bayesian Covariance Discovery (ABCD, Lloyd et al. (2014). For instance, within the realm of spectral mixtures, Remes et al. (2017) proposed the Generalized Spectral Mixture (GSM) kernel, which is a product of three components all parametrized by GPs, representing frequencies,length-scales and mixture weights. In the category of summation and/or multiplication, Verma and Engelhardt (2020) utilized the sum of the RBF kernel and Matérn kernel with varying smoothness as the final kernel in their t-distribution Gaussian process latent variable model (tGPLVM), an extension to GPLVM (Lawrence, 2003; Lalchand et al., 2022), to characterize the latent features in single-cell RNA sequencing data.

In addition, methodological advances have facilitated the efficient discovery of optimal kernel mixtures. For example, NKN is reported to be more efficient than the Automatic Statistician, a gradient-based method. Simpson et al. (2021) utilized a transformer-based framework to generate mixture kernel recommendations. These studies underscore the advantages of employing mixed kernels in GP modeling, showcasing improved model fitting and more accurate predictions. By combining multiple kernels, the strengths of individual kernels can be leveraged to capture complex data patterns and relationships, potentially exceeding the capabilities of single kernels. Furthermore, the automatic selection of kernels optimizes model performance, reducing the reliance on extensive prior knowledge.

Another essential benefit of using mixture kernels in GP modeling lies in the improved interpretability. In contrast to complex, data-driven kernels, mixture kernels allow the decomposition of intricate patterns into simpler, distinct base kernels that are more readily interpretable. This technique, often termed decomposition, enables a more comprehensive understanding of the underlying data structure by simplifying complex patterns into their constitutive components. An illustrative example of this approach is the work of Duvenaud et al. (2013), who applied this decomposition technique for structure discovery in time series data. Their proposed mixture kernel comprised several base kernels, including RBF, periodic, linear, and rational quadratic kernels, enabling the dissection of time-series data patterns into components such as long-term trends, annual periodicity, and medium-term deviations. In a similar vein, the ABCD approach also employed decomposition, albeit with a different set of base kernels.

The identifiability of parameters within a single Matérn kernel has previously been explored, with the microergodic parameter uniquely identified as the only identifiable parameter, as outlined in Stein (1999). Tang et al. (2022) examined the identifiablity of parameters within a single Matérn kernel with nuggets. Despite the prevalent use and interpretation of mixture kernels, their theoretical properties including identifiability and interpretability of parameters within kernel components, to the best of our knowledge, remain underexplored. A related critical question pertains to the common practice of using kernels with varying degrees of smoothness for enhanced flexibility. In this work, we turn our attention to the additive kernel in univariate GPs and multiplicative (separable) kernel in multivariate GPs, specifically focusing on the mixture of the widely-used Matérn kernel. We highlight the following novel findings:

- The smoothness of an additive mixture kernel is completely determined by the least smooth component.

- For additive mixture of Matérn kernels, the identifiability is confined only to a single parameter, also known as the microergodic parameter that is associated with the least smooth kernel component.

- For multivariate GPs with multiplicative separable kernels, the multiplicative matrix that controls the correlation structure among the response variables is identifiable up to a multiplicative constant.

- Our conclusions extend beyond the specific case of Matérn kernels, demonstrating applicability to a wider range of mixture kernels.

Our study aims to deepen the understanding of mixture kernel identifiability and interpretability, as well as to clarify the practical benefits of using kernels with varying degrees of smoothness. Our theoretical assertions are supported by both simulations and real-world applications. Details regarding the proofs of our theories and numerical experiments can be found in the supplementary material.

## 2 Gaussian process, kernels and parameter inference

This section serves to define key terms and outline the parameter inference algorithm integral to the forthcoming theory and simulation sections.

A GP is a random function where any finite set of its realizations follows a multivariate normal distribution, characterized by a mean function $\mu$ and a covariance function $K$ (Rasmussen and Williams, 2006).

**Definition 1.** *$f$ is said to follow a Gaussian process in domain $\Omega$ with a mean function $\mu : \Omega \to \mathbb{R}$ and a covariance function $K : \Omega \times \Omega \to \mathbb{R}$ if for any $x_1, \ldots, x_n \in \Omega$,*

$$[f(x_1), \ldots, f(x_n)]^\top \sim N(v, \Sigma), \text{ where } v = [\mu(x_1), \ldots, \mu(x_n)]^\top, \Sigma_{ij} = K(x_i, x_j).$$

For the purpose of this study, we assume $\mu = 0$, adhering to common practice and for the sake of simplicity. If the mean is not zero, a data transformation can be applied to achieve this. This approach is often adopted in machine learning and statistical analysis to simplify calculations and comply with certain algorithmic requirements (Rasmussen and Williams, 2006; Murphy, 2012).

As a consequence, the behavior of a GP is primarily determined by its kernel function. In this study, we assume the domain $\Omega = \mathbb{R}^p$ and our focus is on the selection of kernels that are widely recognized and commonly used in machine learning.

**Definition 2.** *The RBF kernel, also known as the squared exponential kernel of Gaussian kernel is given by:*

$$K(x, x') = \sigma^2 \exp(-\alpha \|x - x'\|^2).$$

The parameter $\sigma^2$ is called the spatial variance or partial sill that controls the point-wise variance, while $\alpha$ is called the (length) scale, range, or decay that controls the spatial dependency. A key characteristic of RBF is its smoothness, defined as follows:

**Definition 3.** *A Gaussian process $f$ is said to be mean-square continuous (MSC) if $\mathbb{E}(f(x + h) - f(x))^2 \xrightarrow{h \to 0} 0$ for any $x \in \mathbb{R}^p$. $f$ is said to be mean-square differentiable (MSD) if $\lim_{h \to 0} \frac{f(x+h) - f(x)}{h}$ exists in the mean-square topology, and the limiting process is called the derivative process of $f$, denoted by $f'$. Similarly, the $d$-times mean-square differentiable ($d$-MSD) GPs can be defined inductively.*

In particular, the above-defined RBF kernel is infinitely differentiable. However, oversmoothing can be problematic in prediction (Stein, 1999), leading to the popularity of the following flexible family of kernels that allow for varying smoothness:

**Definition 4.** *The Matérn kernel is given by*

$$K(x, x') = \frac{\sigma^2 2 \left( \frac{\alpha}{2} \|x - x'\| \right)^\nu}{\Gamma(\nu)} K_\nu(\alpha \|x - x'\|)$$

*where $K_\nu$ is the modified Bessel function of the second kind. When $\nu = 1/2$, the Matérn kernel becomes the exponential kernel:*

$$K(x, x') = \sigma^2 \exp(-\alpha \|x - x'\|)$$

The additional parameter $\nu$ is called the smoothness, since the smoothness of a Matérn GP is exactly $\lceil \nu \rceil - 1$. Inferring $\nu$ is known to be a particularly challenging problem, both from the theoretical and empirical perspectives (Zhang, 2004). In practice, $\nu$ is usually set to $1/2, 3/2, 5/2, \cdots$ due to the simplified analytic form of the modified Bessel function.

The most straightforward estimator of the parameters $\sigma^2$ and $\alpha$ is the maximum likelihood estimator (MLE). In practice, optimizers such as Adam (Kingma and Ba, 2015) or stochastic gradient descent (SGD) are used to maximize the log-likelihood.

# 3 Univariate GP: additive kernels

In this section, we investigate the smoothness of mixture kernels and the identifiability of parameters in the additive mixture of Matérn kernels in the context of a univariate response variable. We note here that all the theorems presented are framed in an asymptotic context.

Consider $K_1, \cdots, K_L$ as $L$ kernels with $d_l$-MSD. Define $K = \sum_{l=1}^{L} w_l K_l$ where $\sum_{l=1}^{L} w_l = 1$ and $w_l \geq 0$.

**Theorem 1.** *$K$ is $d := \min_l\{d_l\}$-times MSD, but not $d + 1$-times MSD, so the smoothness of $K$ is determined by the least smooth component.*

Beyond smoothness, we discuss the identifiability of the parameters in the mixture kernel, which relies on the notion of equivalence of measures.

**Definition 5.** *Two measures $\mathbb{P}_1$ and $\mathbb{P}_2$ are said to be equivalent, i.e., $\mathbb{P}_1 \equiv \mathbb{P}_2$, if they are absolutely continuous with respect to each other, i.e., $\mathbb{P}_1(B) = 0 \iff \mathbb{P}_2(B) = 0$. Two GPs are equivalent if the corresponding Gaussian random measures are equivalent.*

As a consequence, two equivalent GPs cannot be distinguished by any finite number of realizations (Stein, 1999). Specifically, given a family of GPs parametrized by $\theta$, if $\mathbb{P}_{\theta_1} \iff \mathbb{P}_{\theta_2}$ with $\theta_1 \neq \theta_2$, then $\theta$ is not identifiable since we cannot distinguish between $\theta_1$ and $\theta_2$. As a corollary, there does not exist any **consistent** estimator for $\theta$.

Now we can study the identifiability of the mixture Matérn kernel $K = \sum_{l=1}^{L} w_l K_l$, where $K_l$ is the Matérn kernel with parameters $(\sigma_l^2, \alpha_l, \nu_l)$, given known $\nu_l$'s in ascending order $\nu_1 < \nu_2 \cdots < \nu_L$ with $\nu_l - \nu_{l-1} \geq 1$ for different smoothness.

**Theorem 2.** *Let $K = \sum_l w_l K_l$ and $\widetilde{K} = \sum_l \widetilde{w}_l \widetilde{K}_l$ be two kernels represented as linear combinations of Matérn kernels $K_l$'s and $\widetilde{K}_l$ with parameters $(\sigma_l^2, \alpha_l, \nu_l)$ and $(\widetilde{\sigma}_l^2, \widetilde{\alpha}_l, \nu_l)$.*

*(i) When $p = 1, 2, 3$, Then $K \equiv \widetilde{K}$ if $w_1 \sigma_1^2 \alpha_1^{2\nu_1} = \widetilde{w}_1 \widetilde{\sigma}_1^2 \widetilde{\alpha}_1^{2\nu_1}$.*

*(ii) When $p \geq 5$, Then $K \equiv \widetilde{K}$ if*

$$w_1 \sigma_1^2 \alpha_1^{2\nu_1} = \widetilde{w}_1 \widetilde{\sigma}_1^2 \widetilde{\alpha}_1^{2\nu_1}, \ w_2 \sigma_2^2 \alpha_2^{2\nu_2} - \nu_1 w_1 \sigma_1^2 \alpha_1^{2(\nu_1+1)} = \widetilde{w}_2 \widetilde{\sigma}_2^2 \widetilde{\alpha}_2^{2\nu_2} - \nu_1 \widetilde{w}_1 \widetilde{\sigma}_1^2 \widetilde{\alpha}_1^{2(\nu_1+1)}.$$

*Consequently, no single parameter is identifiable. The only identifiable parameter when $p \leq 3$ is $w_1 \sigma_1^2 \alpha_1^{2\nu_1}$, known as the microergodic parameter (Stein, 1999).*

Interestingly, the case of $p = 4$ is missing here. We hypothesize that it aligns with the case of $p \geq 5$. However, providing a proof remains challenging. It is also worth noting that the issue of identifiability persists even for a single Matérn kernel, as highlighted in the existing literature (Zhang, 2004; Anderes, 2010; Li et al., 2023).

**Corollary 1.** *The mixture kernel $K$ is equivalent to $K_1$, that is, the mixture kernel is equivalent to its least smooth component. Furthermore, the mean squared error (MSE) of the mixture kernel $K$ is asymptotically equal to the MSE of $K_1$.*

Theorem 2 implies that interpreting $w_l$ as the weight of each component is often misleading due to its nonidentifiability. Furthermore, Corollary 1 suggests the use of a single Matérn kernel in lieu of the linear mixture kernel. Both assertions are supported by the simulation study in Section 5 and the real-world application in Section 6.

An alternative strategy is to assume the same smoothness across mixing components, i.e., $\nu_1 = \nu_2, \cdots, \nu_L$. In this case, we prove that no single parameter is identifiable. The complete theorem, proof, and simulation are in the Supplement.

When considering real-world data, it is often the case that the observed outcomes are noisy, which is frequently modeled as an i.i.d. Gaussian with variance denoted by $\tau^2$, commonly referred as the "nugget". The inclusion of such a noise term is essential to capture the inherent variability in the data. Equivalently, the model can be formulated as noiseless by adjusting the kernel to be $K(x, x') + \tau^2 \mathbf{1}_{\{x=x'\}}$. The following corollary shows that the noise, or nugget, does not impact the identifiability and interpretability of the mixture of Matérn kernels as shown in Theorem 2.

**Corollary 2.** *Let $\tau^2$ and $\widetilde{\tau}^2$ be the noise variance of $K$ and $\widetilde{K}$. If $\tau^2 \neq \widetilde{\tau}^2$, then $K \not\equiv \widetilde{K}$. If $\tau^2 = \widetilde{\tau}^2$, the previous results in Theorem 2 hold.*

In essence, while the presence of noise or "nugget" captures real-world data variability, it does not obscure or alter the fundamental characteristics of the mixture of Matérn kernels as established in our theorems.

## 4 Multivariate GP: multiplicative kernels

In this section, we shift our focus to multivariate GPs, also known as multi-output or multi-task GPs, which are applicable to datasets with multiple response variables.

**Definition 6.** *$f : \Omega \to \mathbb{R}^m$ is said to follow a $m$-variate GP in domain $\Omega$ with zero mean and cross-covariance function $K : \Omega \times \Omega \to \mathrm{PD}(m)$, where $\mathrm{PD}(m)$ is the space of all $m$ by $m$ positive definite matrices, if for any $x_1, \cdots, x_n$:*

$$[f(x_1), \cdots, f(x_n)]^\top \sim MN(0, \Sigma),$$

*where $MN$ represents matrix Gaussian distribution and $\Sigma$ consists of $m \times m$ blocks $\Sigma_{ij} = K(x_i, x_j)$.*

The construction of valid cross-covariance kernels is a pivotal yet challenging aspect of multivariate GPs (Gneiting, 2002; Apanasovich and Genton, 2010; Genton and Kleiber, 2015). A popular kernel that is widely used due to its simplicity and seemingly interpretability is the multiplicative kernel, also known as a separable kernel. This kernel admits the form $K(x, y) = AK_0(x, y)$ where $K_0$ is a standard kernel for univariate GP like Matérn while $A \in \mathrm{PD}(m)$ is a positive definite matrix reflecting the correlation between response variables. The following theorem investigates the identifiability of the multiplicative cross-covariance kernel.

**Theorem 3.** *(i) When $K_0$ is Matérn with parameters $(\sigma^2, \alpha, \nu)$ where $\nu$ is given, then $K \equiv \widetilde{K}$ if $\sigma^2 \alpha^{2\nu} A = \widetilde{\sigma}^2 \widetilde{\alpha}^{2\nu} \widetilde{A}$. That is, the identifiable parameter is $\sigma^2 \alpha^{2\nu} A$. Hence $A$ is identifiable up to a multiplicative constant and the correlation structure is identifiable.*

*(ii) Let $K_0$ be an arbitrary kernel with spectral density $\rho_0$ satisfying:*

$$\exists \gamma \in \mathcal{W}_{[-b,b]^m}, \ 0 < b < \infty, \ c_1, c_2 > 0, \ s.t., \ c_1 \gamma(\omega)^2 \leq \rho_0(\omega) \leq c_2 \gamma(\omega)^2,$$

*where $\mathcal{W}_{[-b,b]^d}$ is the space of Fourier transforms of $L^2(\mathbb{R}^p)$ functions with compact support $[-b, b]^d$ (see the Supplement for more details). If $\theta$ is a mircoergodic parameter of $K_0$, such that $\int_{\mathbb{R}^p} \frac{1}{\gamma^4(\omega)} \left( \frac{\rho(\omega)}{\theta} - \frac{\widetilde{\rho}(\omega)}{\widetilde{\theta}} \right)^2 \mathrm{d}\omega < 0$ if $\theta = \widetilde{\theta}$, then $K \equiv \widetilde{K}$ if $\theta A = \widetilde{\theta} \widetilde{A}$. That is, the identifiable parameter is $\theta A$, hence $A$ is identifiable up to a multiplicative constant and the correlation structure is identifiable.*

Note that (i) is a special case of (ii), as the Matérn kernel satisfies the additional assumptions in (ii). This theorem positively supports the utilization of the multiplicative kernel and the interpretation that $A_{ij}$ measures the correlation between the $i$-th response variable and the $j$-th response variable.

## 5 Simulation

In this section, we will validate the proposed theories by conducting three simulation studies. The first simulation demonstrates Theorem 1 using a mixture of Matérn kernels, highlighting that the smoothness of the sample path (e.g., a realization of the GP) is determined by the least smooth component. The second simulation supports Theorem 2 using a mixture kernel of Matérn kernels with varying smoothness. It shows that none of the single parameters are identifiable, and only the proposed microergodic parameter can be identified. Finally, the third simulation showcases Theorem 3, suggesting that the multiplicative matrix is identifiable up to a multiplicative constant.

### 5.1 Simulation 1 - Univariate GP: Smoothness

In the first simulation, we show that the smoothness of the sample path is determined by the smoothness of the least smooth kernel. We generate 200 samples as an equidistant sequence ranging

from $0 - 10$. We consider the following mixture kernel: $K = \sum_{l=1}^{3} w_l K_l$, where $K_l$ is the Matérn kernel with smoothness parameter $\nu_l$ being $\frac{2l-1}{2}$ and $\sum_{l=1}^{L} w_l = 1$, $w_l \geq 0$. We randomly sample $Y$ from multivariate normal distribution with kernel function as the mixture kernel, Matérn with $\nu = 1/2, 3/2, 5/2$. The resulting sample paths are summarized in Figure 1.

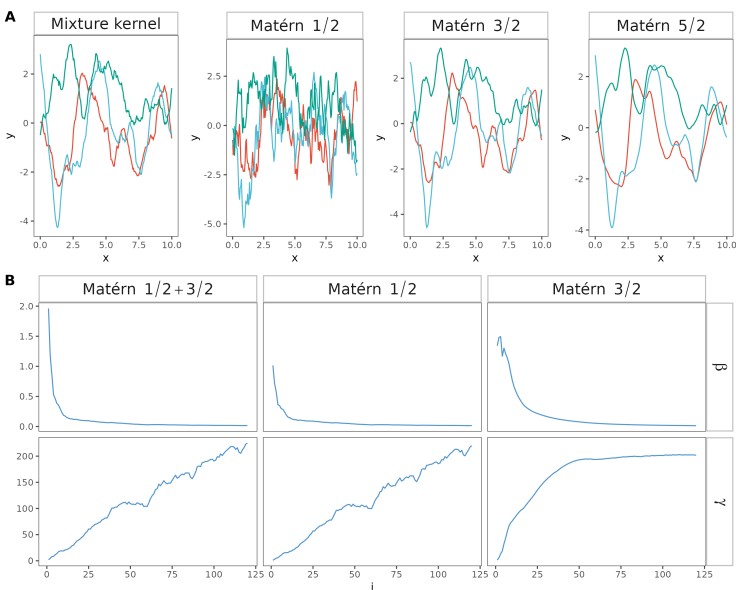

Figure 1: Smoothness of Matérn kernels. Sample path of (A) Mixture kernel (B) Matérn $1/2$ (C) Matérn $3/2$ (D) Matérn $5/2$. (E) Numerical examination of Theorem 1.

Our results clearly indicate that the Matérn kernel with $\nu = 1/2$ exhibits the least degree of smoothness (continuous but not differentiable), and the smoothness increases with $\nu$. The smoothness of the mixture kernel is predominantly influenced by the Matérn kernel with $\nu = 1/2$. When comparing the sample path of the mixture kernel to those of the Matérn kernels with $\nu = 3/2$ and $\nu = 5/2$, it is evident that the mixture kernel demonstrates a degree of smoothness similar to the Matérn kernels with $\nu = 1/2$.

To better demonstrate the smoothness of the mixture kernel and its least component, we further examine the continuity and MSD of the mixture kernel and its mixing components empirically. For a fixed $x_0 = 0$, let $x_i = 1/i$ for $i = 1, 2, \ldots$, and we generate $y_i^l$ from the GP, where $l = 1, \ldots, T$ denotes the index of replicates. This allows us to approximate $\lim_{x \to 0} \mathbb{E}(f(x) - f(0))^2$ by $\beta_i = \frac{1}{T} \sum_{l=1}^{T} (y_i^l - y_0^l)^2$. As per Definition 3, the GP is continuous if and only if $\beta_i \to 0$. Similarly, we can approximate $\lim_{x \to 0} \mathbb{E} \left( \frac{f(x) - f(0)}{x} \right)^2$ by $\gamma_i = \frac{1}{T} \sum_{l=1}^{T} \left( \frac{y_i^l - y_0^l}{x_i} \right)^2$. The GP is mean-square differentiable if and only if $\lim_{i \to \infty} \gamma_i$ exists. Specifically, for the mixture of Matérn $1/2$ and $3/2$ and Matérn $1/2$, while $\beta_i \to 0$, $\gamma_i$ does not converge (Figure 1 first two columns), which indicates that both the mixture kernel and Matérn $1/2$ are continuous but not differentiable. However, for Matérn $3/2$ (Figure 1 third column), $\beta_i \to 0$ and $\gamma_i$ converges, implying that Matérn $3/2$ is continuous and differentiable.

These empirical observations strongly support the claims made in Theorem 1, which suggests that the inclusion of smoother kernel components in the mixture does not inherently enhance the overall smoothness of the mixture kernel.

## 5.2 Simulation 2 - Univariate GP: Components with different smoothness

In the second simulation, our aim is to assess parameter identifiability in a GP with a mixture kernel consisting of Matérn kernels with distinct smoothness. The mixture kernel is denoted as $K = \sum_{l=1}^{3} w_l K_l$, where $K_l$ is a Matérn kernel with parameters $(\sigma_l^2, \alpha_l, \nu_l)$. For this simulation, we set $\nu_1 = 1/2, \nu_2 = 3/2, \nu_3 = 5/2$. Theorem 2 indicates that the only identifiable parameter is

$w_l\sigma_l^2\alpha_l$, while all other parameters remain unidentifiable. In this scenario, the identifiable parameter, also known as the microergodic parameter, is associated with the least smooth component, i.e., the Matérn $1/2$ kernel. We will assess parameter inference by comparing estimated values to their true values across different training sample sizes. For this experiment, we generate $n$ samples, ranging from $20, 50, 100, 500$. For each sample size, we replicate the simulation 100 times.

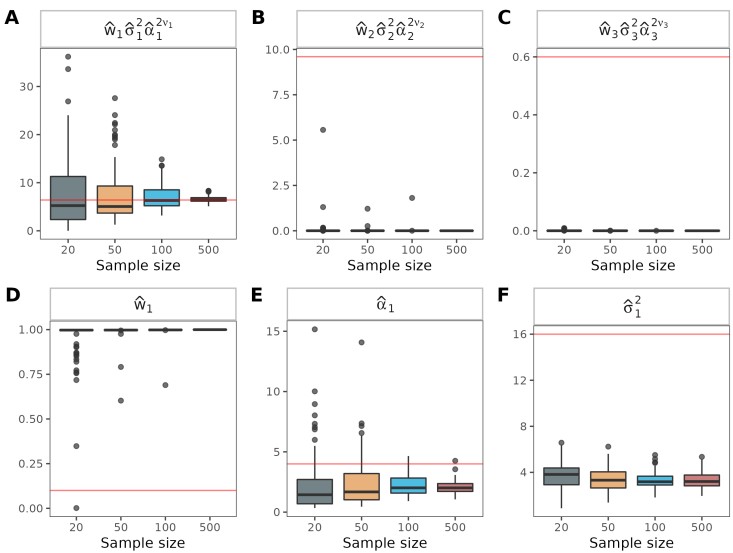

Figure 2: Parameter estimation. (A) Only the microergodic parameter $w_1\sigma_1^2\alpha_1$ is identifiable. (B-F) other parameters are not identifiable. The red line indicates the true value.

The results provide persuasive support for Theorem 2 that, in the context of additive mixture of Matérn kernels with distinct smoothness, only the MLE of the microergodic parameter $w_1\sigma_1^2\alpha_1$ for the least smooth component converges to the true value. The MLEs for all other parameters do not converge to their respective true values, highlighting the unique identifiability of the parameters in the least smooth component within the additive mixture kernel framework. Such results are robust in terms of optimizer choice and consistent as the sample size increases to 3000 (details in the Supplement).

## 5.3 Simulation 3 - Multivariate GP

Next, we turn our attention to multivariate GPs with a separable kernel. We define the separable kernel as $K = AK_0$. Here, we select Matérn $1/2$ as $K_0$, characterized by the parameters $(\sigma^2, \alpha)$. Our Theorem 3 suggests that only $A\alpha^{2\nu}\sigma^2$ is identifiable. To verify this, we adopt a bivariate setup where $A$ is a $2 \times 2$ positive definite matrix. In this simulation, the sample size $n$ varies from $50, 100, 200, 400$. For each sample size, we replicate the simulation 100 times.

The parameter estimates are summarized in Figure 3. Evidently, only the MLE of the microergodic parameter $A\sigma^2\alpha^{2\nu}$ converges to its true value. These findings reinforce the identifiability and interpretability of the separable kernel. This understanding is critical for the interpretation and application of separable kernels in real-world situations.

## 6 Application

Motivated by the findings in our theoretical and simulation studies, we proceed to compare the performance of the mixture kernel with varying smoothness and the least smooth component within that mixture only, in real-world data applications. In this section, we will revisit several applications in previous studies (Rasmussen and Williams (2006), Wilson et al. (2014), Sun et al. (2018)), illustrating that the mixture kernel achieves prediction performance comparable to the least smooth kernel component within the mixture.

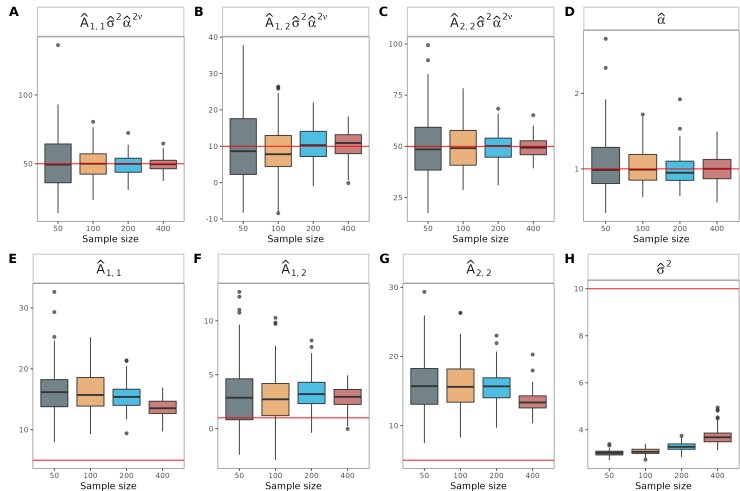

Figure 3: Parameter estimation. (A-C) Only the microergodic parameter $A\sigma^2\alpha^{2\nu}$ is identifiable (D-E) All other parameters are not identifiable. The red dotted line indicates the true value.

## 6.1 Application 1 - Image prediction

For our first real-data application, we delve into the domain of image analysis. GP has been widely used to predict missing image (Wilson et al. (2014), Sun et al. (2018)). This task of image prediction is an exemplary context for assessing kernel performance in discerning local correlation patterns, providing a visually intuitive framework for comparative analysis.

We employ a handwritten zero image from the MNIST dataset (LeCun et al. (1998)) for this analysis; we extract a $20 \times 20$ section from the center of the original $100 \times 100$ image to serve as our test image. This results in a training dataset of 9600 pixels and a testing dataset with 400 pixels (Figure 4A). The input for this analysis comprises the 2-D pixel locations, with the corresponding pixel intensities serving as the response. To reconstruct the missing area, we train GP regression models with both the mixture kernel with varying smoothness and a single Matérn $1/2$ kernel on the training dataset, and subsequently make predictions on the testing dataset. Specifically, we employ a mixture of three Matérn kernels of smoothness $1/2$, $3/2$, $5/2$. Our analysis seeks to demonstrate the similarities in performance between the mixture kernel and the least smooth component within that mixture, i.e., Matérn $1/2$, in terms of prediction in the missing image area.

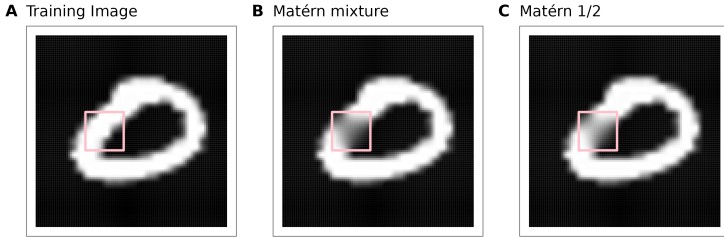

Figure 4: Texture exploration in diamond plate. (A) Training image. (B) Prediction of the mixture kernel. The area in the pink box represents the testing image area. (C) Predication of the Matérn $1/2$ kernel.

The predictions generated by both the mixture kernel and the Matérn $1/2$ kernel demonstrate a remarkable similarity, as shown in Figure 4. This observation suggests that the mixture kernel and a single Matérn $1/2$ kernel have comparable performance. The close alignment between their predictions implies that using the mixture kernel does not yield significant advantages over the least smooth kernel component in this specific application, further reinforcing our theoretical findings (Theorem 2) regarding the dominance of the least smooth kernel component in the mixture kernel.

## 6.2 Application 2 - Manua Loa CO$_2$

In this section, we extend our comparative analysis to the widely recognized Moana Loa CO$_2$ dataset from the Global Monitoring Laboratory's Repository as our benchmark for regression analysis (Tans and Keeling (2023)). Rasmussen and Williams (2006) employed this dataset to demonstrate that constructing a complex covariance function by combining several different types of simple covariance functions could yield an excellent fit to complex data. Consequently, we consider it to be a suitable choice for testing our theorem.

Specifically, we used the decimal date from 1960 to 2020 as our input and the monthly average CO$_2$ as our response. In this context, we compare the performance of Matérn mixture $1/2 + 3/2$, $1/2 + 3/2 + 5/2$, $3/2 + 5/2$ and three single Matérn kernels ($1/2$, $3/2$, $5/2$). The dataset consists of 720 records. We undertook an extensive assessment, varying the training sample size from $5\%$ to $95\%$ and keeping the test sample size as the remaining data. For each training sample size, we conducted 10 simulations using different random splits of the dataset as replications. This strategy allows for a robust and thorough evaluation of the prediction accuracy of both the mixture kernel and the single kernel component in the context of a regression problem with varying training sample sizes. We use the MSE to quantify the prediction accuracy (Figure 5).

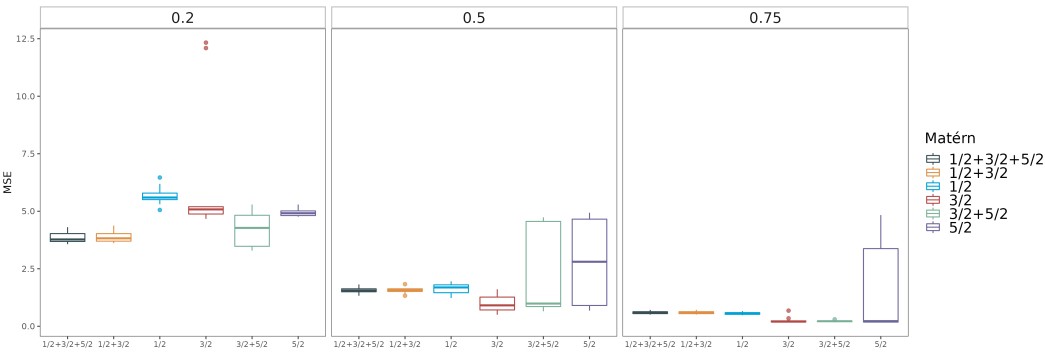

Figure 5: Manua Loa CO$_2$ prediction MSE in test dataset with 20%, 50% and 75% training dataset. Boxplot are colored by different Matérn kernels.

The Matérn mixture kernel $1/2 + 3/2 + 5/2$ aligned closely with $1/2 + 3/2$ across all experiments. Furthermore, these three Matérn mixture kernels ($1/2 + 3/2 + 5/2, 1/2 + 3/2, 1/2$) converge when the training sample size ratio reaches $50\%$. This empirical finding supports the theoretical equivalence of these three GPs as posited in Theorem 2. Similarly, the Matérn mixture $3/2 + 5/2$ kernel and the standalone Matérn $3/2$ kernel converge from a $65\%$ sample size, lending empirical weight to their theoretical equivalence and their equivalence in MSE as stated in Corollary 1.

Matérn $3/2$ kernel has exceptional performance across all experiments. However, its influential performance is diluted when coupled with the less smooth component, Matérn $1/2$. On the contrary, when mixed with the smoother Matérn $5/2$ component, the worse performance of the latter is overshadowed by Matérn $3/2$. These behaviors lend empirical weight to our Theorem 2, suggesting that in the asymptotic sense, the mixture kernel is dominated by its least smooth component.

A noteworthy trend is the inequivalance of the Matérn mixture kernel and its least component with a limited training sample (training sample size $< 50\%$). We consider such scenario as the "finite" sample scenario in contrast to our focus on the infinite sample scenario, i.e. asymptotic theory. Theoretically, while the Matérn mixture kernels $1/2 + 3/2 + 5/2, 1/2 + 3/2$, and the standalone Matérn $1/2$ kernel are deemed equivalent, and the Matérn mixture kernel $3/2 + 5/2$ is akin to the Matérn $3/2$ kernel, their practical agreement points, i.e. $50\%$ and $65\%$, differed with limited training samples. Although securing a conclusive theoretical support for the finite sample regime is challenging, we managed to provide some insights to interpret our findings. To do so, we have to delve further into the proof of Stein (1993)'s Theorem 1, which guides us to Theorem 3.1 in Stein (1990). The proof suggests that the relative difference between the MSEs rests on the tail of the series. This difference is influenced by both the sample size (denoted as $N$ in Stein (1990)) and by $b_{jk}$ and $\mu_j$, as defined on the same page. For a fixed sample size, smaller values of $(b_{jk} + \mu_j \mu_k)^2$ result in a

smaller relative difference between MSEs. By the definition of $b_{jk}$ and $\mu_j$, the more "different" the two kernels are, in other words, the more "significant" the additional smoother components become — the greater the difference between the MSEs will be.

# 7 Discussion and future work

In this study, we delved into the identifiability and interpretability of parameters in GPs with various mixture kernels in the asymptotic scenario and fixed domain, including the additive kernel and separable kernels. We formulated a series of theorems clarifying the identifiable parameters in these kernel structures, and further corroborated our theorems through multiple simulations and real-data applications. Our simulation results convincingly demonstrated that in GPs with mixture kernels, the only identifiable parameter, known as the microergodic parameter, is associated with the least smooth kernel component. This discovery has profound implications for parameter interpretation in the context of GPs with mixture kernels. Empirical evidence from image data analysis and Manua Loa $CO_2$ regression studies further reinforce these discoveries. Despite the inclusion of kernels with varying smoothness in the mixture kernel, the performance of the mixture kernel closely paralleled that of the least smooth kernel component within it when the sample size is large enough, regardless of its performance. This observation suggests that the inclusion of kernels with different smoothness does not necessarily improve the prediction accuracy. In fact, due to various real world factors including limited training samples, optimization and more, determining a clear winner in performance between Matérn mixture kernel and single Matérn kernel proves to be a challenging task. Lastly, in the case of multivariate GPs with separable kernels, our theoretical and simulation results show that the correlation structure is identifiable up to a multiplicative constant. This result underlines that the interpretability of separable kernels mainly resides in the relative correlation structure, rather than individual parameters.

Although our study has provided substantial insight on the identifiability and interpretability of parameters in GPs with various mixture kernel types, it also opens several exciting avenues for future research. First, our analysis primarily focuses on Matérn kernels, and it would be intriguing to extend this framework to other families of kernels, such as the periodic kernel. Such an extension would provide a more comprehensive understanding of parameter identifiability and predictive performance across a broader spectrum of kernel types. Second, our work has so far considered the cases where $p \leq 3$ or $p \geq 5$. However, extending this to $p = 4$ presents a substantial challenge due to the lack of mathematical tools to determine whether two Gaussian random measures are equivalent or not. Third, our observations in the Mauna Loa $CO_2$ example underscore the need for further exploration in the finite sample scenario. Potential avenues for future research include investigating the convergence rates of both the Matérn mixture kernel and the single Matérn kernel. Lastly, while we have identified the microergodic parameters that are theoretically identifiable in mixture kernels, an important direction for future work involves finding consistent estimators for these kernel parameters. This would entail developing novel estimation techniques or adapting existing ones to reliably estimate the parameters in practice, thereby enhancing the practical utility of our theoretical findings. These endeavors will not only extend the theoretical foundations of GPs with mixture kernels, but will also broaden their applicability across various real-world scenarios.

## Acknowledgments and Disclosure of Funding

We express our sincere gratitude to anonymous reviewers and the Area Chair for their insightful comments and constructive feedback, which significantly enhanced the quality and clarity of this work. DL was supported by the NIH/NCATS award UL1 TR002489, NIH/NHLBI award R01 HL149683 and NIH/NIEHS award P30 ES010126. YL was partially supported by NIH awards R01AG079291 and U01HG011720.

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
