# Supplement for "On the Identifiability and Interpretability of Gaussian Process Models"

**Jiawen Chen**
Department of Biostatistics
University of North Carolina at Chapel Hill
jiawenn@email.unc.edu

**Wancen Mu**
Department of Biostatistics
University of North Carolina at Chapel Hill
wancen@live.unc.edu

**Yun Li**
Department of Biostatistics
University of North Carolina at Chapel Hill
yun_li@med.unc.edu

**Didong Li**
Department of Biostatistics
University of North Carolina at Chapel Hill
didongli@unc.edu

## Abstract

Section 1 is the proof of Theorem 1. Section 2 is the proof of Theorem 2. Section 3 is the proof of Corollary 1. Section 4 is the proof for Corollary 2. Section 5 is the proof of the additional Theorem 4. Section 6 is the proof of Theorem 3. Section 7 is the experiment details for simulations and applications. All the codes for replicating the analysis and parameter estimates are available at: `https://github.com/JiawenChenn/GP_mixture_kernel`.

## 1 Proof of Theorem 1

We first state a Lemma for determining the smoothness for a stationary GP.

**Lemma 1** (Stein (1999)). *Let $\rho$ be the spectral density of a GP with kernel $K$, then $K$ is $d$-times mean squared differentiable if and only if*

$$\int \|\omega\|^{2d} \rho(\omega) \mathrm{d}\omega < \infty.$$

*Proof of Theorem 1.* Let $\rho_l$ be the spectral density of $K_l$, then the spectral density of $K$ is given by $\rho = \sum_{l=1}^{L} \rho_l$ by the linearity of Fourier transform. Then by the smoothness of $K_l$ and Lemma 1, we have $\int \|\omega\|^{2d_l} \rho_l(\omega) \mathrm{d}\omega < \infty$. As a result,

$$\int \|\omega\|^{2d} \rho(\omega) \mathrm{d}\omega = \sum_{l=1}^{L} \int \|\omega\|^{2d} \rho_l(\omega) \mathrm{d}\omega = \sum_{l=1}^{L} \int_{\|\omega\| \leq 1} \|\omega\|^{2d} \rho_l(\omega) \mathrm{d}\omega + \sum_{l=1}^{L} \int_{\|\omega\| \geq 1} \|\omega\|^{2d} \rho_l(\omega) \mathrm{d}\omega$$

$$\leq \sum_{l=1}^{L} \int_{\|\omega\| \leq 1} \rho_l(\omega) \mathrm{d}\omega + \sum_{l=1}^{L} \int \|\omega\|^{2d_l} \rho_l(\omega) \mathrm{d}\omega < \infty,$$

that is, $K$ is $d$-times MSD. Similarly, $K$ is not $d+1$-times MSD since

$$\int \|\omega\|^{2(d+1)} \rho(\omega) \mathrm{d}\omega = \sum_{l=1}^{L} \int \|\omega\|^{2(d+1)} \rho_l(\omega) \mathrm{d}\omega \geq \int \|\omega\|^{2(d+1)} \rho_1(\omega) \mathrm{d}\omega = \infty.$$

$\square$

37th Conference on Neural Information Processing Systems (NeurIPS 2023).

## 2 Proof of Theorem 2

We first state a Lemma, also known as the integral test to determine whether two GPs are equivalent or not.

**Lemma 2** (Yadrenko and Balakrishnan (1983), Zhang (2004)). *Let $K_1$ and $K_2$ be the kernels of two GPs $P_1$ and $P_2$ with spectral densities $\rho_1$ and $\rho_2$. Then $P_1 \equiv P_2$ if*

*1. There exists $r > 0$ such that $\|\omega\|^r \rho_1(\omega)$ is bounded away from both zero and infinity as $\omega \to \infty$.*

*2. There exists $\delta > 0$ such that $\int_{\|\omega\|>\delta} \left( \frac{\rho_2(\omega)-\rho_1(\omega)}{\rho_1(\omega)} \right)^2 d\omega < \infty$.*

*Proof of Theorem 2.* Recall that the spectral densities of $K_l$ are

$$\rho_l(\omega) = \frac{\sigma_l^2 \alpha_l^{2\nu_l}}{(\alpha_l^2 + \|\omega\|^2)^{\nu_l+\frac{p}{2}}}, \ \rho(\omega) = \sum_{l=1}^L w_l \frac{\sigma_l^2 \alpha_l^{2\nu_l}}{(\alpha_l^2 + \|\omega\|^2)^{\nu+\frac{p}{2}}}, \ \widetilde{\rho}(\omega) = \sum_{l=1}^L \widetilde{\omega}_l \frac{\widetilde{\sigma}_l^2 \widetilde{\alpha}_l^{2\nu_l}}{(\widetilde{\alpha}_l^2 + \|\omega\|^2)^{\nu_l+\frac{p}{2}}}.$$

To apply the integral test, first, let $r = 2(\nu_1 + \frac{p}{2})$, then

$$\rho(\omega)\|\omega\|^r = \sum_{l=1}^L w_l \frac{\sigma_l^2 \alpha_l^{2\nu_l}\|\omega\|^r}{(\alpha_l^2 + \|\omega\|^2)^{\nu_l+\frac{p}{2}}} \xrightarrow{\omega\to\infty} w_1 \sigma_1^2 \alpha_1^{2\nu_1},$$

that is, bounded away from both 0 and $\infty$ as $\omega \to \infty$. Then we check the relative difference between the spectral densities $\rho$ and $\widetilde{\rho}$. Observe that when,

$$\rho(\omega) = \sum_{l=1}^L w_l \frac{\sigma_l^2 \alpha_l^{2\nu_l}}{(\alpha_l^2 + \|\omega\|^2)^{\nu_l+\frac{p}{2}}} = \frac{\sum_{l=1}^L w_l \sigma_l^2 \alpha_l^{2\nu_l} \Pi_{j\neq l}(\alpha_j^2 + \|\omega\|^2)^{\nu_j+\frac{p}{2}}}{\Pi_{l=1}^L(\alpha_l^2 + \|\omega\|^2)^{\nu_l+\frac{p}{2}}} =: \frac{\sum_{l=1}^L w_l \sigma_l^2 \alpha_l^{2\nu_l} p_l(\omega)}{\Pi_{l=1}^L(\alpha_l^2 + \|\omega\|^2)^{\nu_l+\frac{p}{2}}},$$

where $p_l := \Pi_{j\neq l}(\alpha_j^2 + \|\omega\|^2)^{\nu_j+\frac{p}{2}}$. Similarly,

$$\widetilde{\rho}(\omega) = \frac{\sum_{l=1}^L \widetilde{w}_l \widetilde{\sigma}_l^2 \widetilde{\alpha}_l^{2\nu} \widetilde{p}_l(\omega)}{\Pi_{l=1}^L(\widetilde{\alpha}_l^2 + \|\omega\|^2)^{\nu_l+\frac{p}{2}}},$$

where $\widetilde{p}_l := \Pi_{j\neq l}(\widetilde{\alpha}_j^2 + \|\omega\|^2)^{\nu_j+\frac{p}{2}}$. Observe that when $\|\omega\| \to \infty$,

$$\frac{\Pi_{l=1}^L(\widetilde{\alpha}_l^2 + \|\omega\|^2)^{\nu_l+\frac{p}{2}}}{\Pi_{l=1}^L(\alpha_l^2 + \|\omega\|^2)^{\nu_l+\frac{p}{2}}} = \frac{\Pi_{l=1}^L \left(1 + \frac{\widetilde{\alpha}_l^2}{\|\omega\|^2}\right)^{\nu_l+\frac{p}{2}}}{\Pi_{l=1}^L \left(1 + \frac{\alpha_l^2}{\|\omega\|^2}\right)^{\nu_l+\frac{p}{2}}} = \frac{1 + O(\|\omega\|^{-2})}{1 + O(\|\omega\|^{-2})} = 1 + O(\|\omega\|^{-2}).$$

Furthermore, let $\gamma_l = \sum_{j\neq l}(\nu_j + \frac{p}{2})$, then $p_l(\omega) = O(\|\omega\|^{2\gamma_l})$ where $\gamma_1 > \gamma_2 \cdots > \gamma_L$ and $\gamma_l - \gamma_{l-1} \geq 1$. As a result, the leading term that dominates $\sum_{l=1}^L w_l \sigma_l^2 \alpha_l^{2\nu_l} p_l(\omega)$ is $w_1 \sigma_1^2 \alpha_1^{2\nu_1} p_1(\omega)$, as $\|\omega\| \to \infty$. Moreover, $\sum_{l=1}^L w_l \sigma_l^2 \alpha_l^{2\nu_l} p_l(\omega) = \|\omega\|^{2\gamma_1} \left(w_1 \sigma_1^2 \alpha_1^{2\nu_1} + O(\|\omega\|^{-2})\right)$ since $2\gamma_l - 2\gamma_1 \geq 2$, $\forall l > 1$.

Now we can analyze the relative difference:

$$\left| \frac{\rho(\omega) - \widetilde{\rho}(\omega)}{\rho(\omega)} \right| = \left| \frac{\widetilde{\rho}(\omega)}{\rho(\omega)} - 1 \right| = \left| \frac{\sum_{l=1}^L \widetilde{w}_l \widetilde{\sigma}_l^2 \widetilde{\alpha}_l^{2\nu_l} \widetilde{p}_l(\omega)}{\sum_{l=1}^L w_l \sigma_l^2 \alpha_l^{2\nu_l} p_l(\omega)} \frac{\Pi_{l=1}^L(\widetilde{\alpha}_l^2 + \|\omega\|^2)^{\nu_l+\frac{p}{2}}}{\Pi_{l=1}^L(\alpha_l^2 + \|\omega\|^2)^{\nu_l+\frac{p}{2}}} - 1 \right|$$

$$= \left| \frac{\sum_{l=1}^L \widetilde{w}_l \widetilde{\sigma}_l^2 \widetilde{\alpha}_l^{2\nu_l} \widetilde{p}_l(\omega)}{\sum_{l=1}^L w_l \sigma_l^2 \alpha_l^{2\nu_l} p_l(\omega)} (1 + O(\|\omega\|^{-2})) - 1 \right|$$

$$= \left| \frac{\|\omega\|^{2\gamma_1}(\widetilde{w}_1 \widetilde{\sigma}_1^2 \widetilde{\alpha}_1^{2\nu_1} + O(\|\omega\|^{-2}))}{\|\omega\|^{2\gamma_1}(w_1 \sigma_1^2 \alpha_1^{2\nu_1} + O(\|\omega\|^{-2}))} (1 + O(\|\omega\|^{-2})) - 1 \right|$$

$$= \left| \frac{\widetilde{w}_1 \widetilde{\sigma}_1^2 \widetilde{\alpha}_1^{2\nu_1}}{w_1 \sigma_1^2 \alpha_1^{2\nu_1}} (1 + O(\|\omega\|^{-2}))(1 + O(\|\omega\|^{-2})) - 1 \right|$$

$$= \left| \frac{\widetilde{w}_1 \widetilde{\sigma}_1^2 \widetilde{\alpha}_1^{2\nu_1}}{w_1 \sigma_1^2 \alpha_1^{2\nu_1}} - 1 + O(\|\omega\|^{-2}) \right|.$$

As a result, if $w_1 \sigma_1^2 \alpha_1^{2\nu_1} = \widetilde{w}_1 \widetilde{\sigma}_1^2 \widetilde{\alpha}_1^{2\nu_1}$,

$$\int_{\omega \in \mathbb{R}^p : \|\omega\| > 1} \left| \frac{\rho(\omega) - \widetilde{\rho}(\omega)}{\rho(\omega)} \right|^2 \mathrm{d}\omega \approx \int_{\omega \in \mathbb{R}^p : \|\omega\| > 1} \frac{1}{\|\omega\|^4} \mathrm{d}\omega < \infty,$$

when $p = 1, 2, 3$, so $K \equiv \widetilde{K}$. That is, none of the parameters are identifiable, while the parameter that might be identifiable is $w_1 \sigma_1^2 \alpha_1^{2\nu_1}$, also known as the microergodic parameter.

Now we turn to the case for $p \geq 5$. By Anderes (2010), it suffices to anaylize the principal irregular terms for $K$. For the spectral density of each individual kernel component, the principal irregular term is

$$G_{\nu_l}(\omega) = \frac{-\pi}{2^{2\nu_l} \sin(\nu_l \pi) \Gamma(\nu_l) \Gamma(\nu_l + 1)} \omega^{2\nu_l} =: C_l \omega^{2\nu_l}.$$

Furthermore,

$$K_l(x + h, x') = \sigma_l^2 G_{\nu_l}(|\alpha_l h|) - \nu_l \sigma_l^2 G_{\nu_l + 1}(|\alpha_l h|) + \iota_l(|\alpha_l h|),$$

where $\iota_l(t) = p(|h|) + o(G_{\nu_l + 1}(|h|))$ as $|h| \to 0$ and $p$ is a polynomial with even degree. By linearity, we have

$$
\begin{aligned}
K(x + h, x') &= \sum_{l=1}^{L} w_l K_l(x + h, x') \\
&= \sum_{l=1}^{L} w_l \left( \sigma_l^2 G_{\nu_l}(|\alpha_l h|) - \nu_l \sigma_l^2 G_{\nu_l + 1}(|\alpha_l h|) + \iota_l(t) \right) \\
&= w_1 \sigma_1^2 G_{\nu_1}(|\alpha_1 h|) - w_1 \nu_1 \sigma_1^2 G_{\nu_1 + 1}(|\alpha_1 h|) + w_2 \sigma_2^2 G_{\nu_2}(|\alpha_2 h|) + \iota(t),
\end{aligned}
$$

where $\iota(t) = p(|h|) + o(G_{\nu_2 + 1}(|h|))$ as $|h| \to 0$. They by Theorem 4 of Anderes (2010), there exists consistent estimators for $w_1 \sigma_1^2 \alpha_1^{2\nu_1}$ and $w_2 \sigma_2^2 \alpha_2^{2\nu_2} - \nu_1 w_1 \sigma_1^2 \alpha_1^{2(\nu_1 + 1)}$ when $0 < 2(2\nu_2 - 2\nu_1) < p$, that is $0 < 4 < p$. This completes the proof. $\qquad \square$

## 3 Proof of Corollary 1

We first state a Lemma for comparing MSE of two best linear predictor with two spectral density.

**Lemma 3** (Stein (1993)). *Suppose $Z$ is a zero mean stationary process in $\mathbb{R}^d$ and $x_1, x_2, \ldots$ is a dense sequence of points in a bounded subset of $\mathbb{R}^d$. $x_0$ is a point in the bounded set but not the sequence. Let $\hat{Z}(x_0, n, \rho)$ be the best linear predictor of $Z(x_0)$ based on $Z(x_1), \ldots, Z(x_n)$ assuming $\rho$ is the spectral density for $Z$. $e(x_0, n, \rho) := Z(x_0) - \hat{Z}(x_0, n, \rho)$. If there exists c>0, such that*

$$\lim_{|\omega| \to \infty} \frac{\rho_1(\omega)}{\rho_0(\omega)} = c$$

*and $\rho_0$ satisfies condition 1 in Lemma 2, then*

$$\lim_{n \to \infty} \frac{\mathbb{E}_{\rho_0} e(x_0, n, \rho_1)^2}{\mathbb{E}_{\rho_0} e(x_0, n, \rho_0)^2} = 1$$

*Proof of Corollary 1.* For $K_1 = \mathrm{Mat}(w_1 \sigma_1^2, \alpha_1, \nu_1)$, by Theorem 2, $K_1 \equiv K$ since the microergodic parameters match.

Regarding MSE, let $\rho$ be the spectral density of the mixture kernel $K$, and $\rho_1$ be the spectral density of the first mixing component $K_1$. We assume $\rho_1$ as the true spectral density. Then by the proof of Theorem 2,

$$\lim_{|\omega| \to \infty} \frac{\rho(\omega)}{\rho_1(\omega)} = \frac{w_1 \sigma_1^2 \alpha_1^{2\nu_1}}{w_1 \sigma_1^2 \alpha_1^{2\nu_1}} = 1,$$

then the MSE of GP with $K_1$ is asymptocially equal to the MSE of GP with $K$. $\qquad \square$

# 4 Proof of Corollary 2

We first state a Lemma regarding the equivalence of Gaussian measure with nuggets.

**Lemma 4** (Tang et al. (2022)). *Let $S$ be a closed set, $G_S(m, K)$ denotes the Gaussian measure of the random field on $S$ with mean function $m$ and covariance function $K$. Let $w(s)$ be a mean square continuous process on $S$ under $G_S(m_1, K_1)$ and $\chi$ be a dense sequence of points in $S$. Then*

*1. if $\tau_1^2 \neq \tau_2^2$, then $G_\chi(m_1, K_1, \tau_1^2) \perp G_\chi(m_2, K_2, \tau_2^2)$.*

*2. if $\tau_1^2 = \tau_2^2$, then $G_\chi(m_1, K_1, \tau_1^2) \equiv G_\chi(m_2, K_2, \tau_2^2)$ if and only if $G_S(m_1, K_1) \equiv G_S(m_2, K_2)$.*

*Proof of Corollary 2.* We denote the adjusted kernel with noise of $K$ and $\widetilde{K}$ as $K_\tau$ and $\widetilde{K}_{\widetilde{\tau}}$.

From Lemma 4(1), where we set $m_1 = m_2 = 0$, if $\tau^2 \neq \widetilde{\tau}^2$, then $K_\tau \not\equiv \widetilde{K}_{\widetilde{\tau}}$. If $\tau^2 = \widetilde{\tau}^2$, then $K_\tau \equiv \widetilde{K}_{\widetilde{\tau}}$ if and only if $K \equiv \widetilde{K}$. Then the previous results in Theorem 2 holds. $\square$

# 5 Additional theorem - mixture of Matérn kernels with the same smoothness

Here we introduce a special scenario for Theorem 4 where all component in the mixture kernel have same smoothness. The simulation study for the following theorems are included in Section 7: simulation 4.

**Theorem 4.** *For $K = \sum_{l=1}^{L} K_l$ where $K_l = \text{Mat}(\sigma_l^2, \alpha_l, \nu)$, then*

*(i) When $p = 1, 2, 3$, the only identifiable parameter is $\sum_{l=1}^{L} w_l \sigma_l^2 \alpha_l^{2\nu}$.*

*(ii) When $p > 4$, the only identifiable parameters are $\sum_{l=1}^{L} w_l \sigma_l^2 \alpha_l^{2\nu}$ and $\sum_{l=1}^{L} w_l \sigma_l^2 \alpha_l^{2\nu+2}$.*

*As a consequence, none of the parameters are identifiable.*

*Proof.* The proof is similar to the proof of Theorem 2. Recall that the spectral densities of $K_l$ are

$$\rho_l(\omega) = \frac{\sigma_l^2 \alpha_l^{2\nu}}{(\alpha_l^2 + \|\omega\|^2)^{\nu + \frac{p}{2}}}, \quad \rho(\omega) = \sum_{l=1}^{L} w_l \frac{\sigma_l^2 \alpha_l^{2\nu}}{(\alpha_l^2 + \|\omega\|^2)^{\nu + \frac{p}{2}}}, \quad \widetilde{\rho}(\omega) = \sum_{l=1}^{L} \widetilde{\omega}_l \frac{\sigma_l^2 \alpha_l^{2\nu}}{(\alpha_l^2 + \|\omega\|^2)^{\nu + \frac{p}{2}}}.$$

To apply the integral test, first, let $r = 2(\nu + \frac{p}{2})$, then

$$\rho(\omega)\|\omega\|^r = \sum_{l=1}^{L} w_l \frac{\sigma_l^2 \alpha_l^{2\nu} \|\omega\|^r}{(\alpha_l^2 + \|\omega\|^2)^{\nu + \frac{p}{2}}} \xrightarrow{\omega \to \infty} \sum_{l=1}^{L} w_l \sigma_l^2 \alpha_l^{2\nu},$$

that is, bounded away from both 0 and $\infty$ as $\omega \to \infty$. Then we check the relative difference between the spectral densities $\rho$ and $\widetilde{\rho}$. Observe that

$$\rho(\omega) = \sum_{l=1}^{L} w_l \frac{\sigma_l^2 \alpha_l^{2\nu}}{(\alpha_l^2 + \|\omega\|^2)^{\nu + \frac{p}{2}}} = \frac{\sum_{l=1}^{L} w_l \sigma_l^2 \alpha^{2\nu} \Pi_{j \neq l}(\alpha_j^2 + \|\omega\|^2)^{\nu + \frac{p}{2}}}{\Pi_{l=1}^{L}(\alpha_l^2 + \|\omega\|^2)^{\nu + \frac{p}{2}}} =: \frac{\sum_{l=1}^{L} w_l \sigma_l^2 \alpha^{2\nu} p_l(\omega)}{\Pi_{l=1}^{L}(\alpha_l^2 + \|\omega\|^2)^{\nu + \frac{p}{2}}},$$

where $p_l := \Pi_{j \neq l}(\alpha_j^2 + \|\omega\|^2)^{\nu + \frac{p}{2}}$. Observe that

$$|p_l(\omega)| \geq \|\omega\|^{2(L-1)(\nu + \frac{p}{2})} \tag{1}$$

$$\frac{p_l(\omega)}{\|\omega\|^{2(L-1)(\nu + \frac{p}{2})}} = \Pi_{j \neq l}\left(1 + \frac{\alpha_j^2}{\|\omega\|^2}\right)^{\nu + \frac{p}{2}} = 1 + O(\|\omega\|^{-2}). \tag{2}$$

Now we can analyze the relative difference:

$$
\left|\frac{\rho(\omega)-\widetilde{\rho}(\omega)}{\rho(\omega)}\right| = \left|\frac{\frac{\sum_{l=1}^{L} w_l \sigma_l^2 \alpha^{2\nu} p_l(\omega)}{\Pi_{l=1}^{L}(\alpha_l^2+\|\omega\|^2)^{\nu+\frac{p}{2}}} - \frac{\sum_{l=1}^{L} \widetilde{w}_l \sigma_l^2 \alpha^{2\nu} p_l(\omega)}{\Pi_{l=1}^{L}(\alpha_l^2+\|\omega\|^2)^{\nu+\frac{p}{2}}}}{\frac{\sum_{l=1}^{L} w_l \sigma_l^2 \alpha^{2\nu} p_l(\omega)}{\Pi_{l=1}^{L}(\alpha_l^2+\|\omega\|^2)^{\nu+\frac{p}{2}}}}\right|
$$

$$
= \left|\frac{\sum_{l=1}^{L} w_l \sigma_l^2 \alpha^{2\nu} p_l(\omega) - \sum_{l=1}^{L} \widetilde{w}_l \sigma_l^2 \alpha^{2\nu} p_l(\omega)}{\sum_{l=1}^{L} w_l \sigma_l^2 \alpha^{2\nu} p_l(\omega)}\right|
$$

$$
\leq \frac{1}{\sum_{l=1}^{L} w_l \sigma_l^2 \alpha^{2\nu}} \left|\frac{\sum_{l=1}^{L} w_l \sigma_l^2 \alpha^{2\nu} p_l(\omega) - \sum_{l=1}^{L} \widetilde{w}_l \sigma_l^2 \alpha^{2\nu} p_l(\omega)}{\|\omega\|^{2(L-1)(\nu+\frac{p}{2})}}\right|
$$

$$
= \frac{1}{\sum_{l=1}^{L} w_l \sigma_l^2 \alpha^{2\nu}} \left|\sum_{l=1}^{L} w_l \sigma_l^2 \alpha^{2\nu}(1+O(\|\omega\|^2)) - \sum_{l=1}^{L} \widetilde{w}_l \sigma_l^2 \alpha^{2\nu}(1+O(\|\omega\|^2))\right|
$$

$$
= \frac{1}{\sum_{l=1}^{L} w_l \sigma_l^2 \alpha^{2\nu}} \left|\sum_{l=1}^{L} w_l \sigma_l^2 \alpha^{2\nu} - \sum_{l=1}^{L} \widetilde{w}_l \sigma_l^2 \alpha^{2\nu} + O(\|\omega\|^{-2})\right|.
$$

As a result, if $\sum_{l=1}^{L} w_l \sigma_l^2 \alpha^{2\nu} = \sum_{l=1}^{L} \widetilde{w}_l \sigma_l^2 \alpha^{2\nu}$,

$$
\int_{\omega\in\mathbb{R}^p:\|\omega\|>1} \left|\frac{\rho(\omega)-\widetilde{\rho}(\omega)}{\rho(\omega)}\right|^2 d\omega \approx \int_{\omega\in\mathbb{R}^p:\|\omega\|>1} \frac{1}{\|\omega^4\|} d\omega < \infty,
$$

when $p=1,2,3$, so $K \equiv \widetilde{K}$. That is, none of the parameters are identifiable, while the parameter that might be identifiable is $\sum_{l=1}^{L} w_l \sigma_l^2 \alpha_l^{2\nu}$, also known as the microergodic parameter.

Now we turn to the case for $p \geq 5$. The proof is similar to the proof of Theorem 2: it suffices to analyze the principal irregular terms for $K$. For the spectral density of each individual kernel component, the principal irregular term is

$$
G_\nu(\omega) = \frac{-\pi}{2^{2\nu}\sin(\nu\pi)\Gamma(\nu)\Gamma(\nu+1)}\omega^{2\nu} =: C_l\omega^{2\nu}.
$$

Furthermore,
$$
K_l(x+h,x') = \sigma_l^2 G_\nu(|\alpha_l h|) - \nu\sigma^2 G_{\nu+1}(|\alpha_l h|) + \iota(|\alpha_l h|),
$$

where $\iota(t) = p(|h|) + o(G_{\nu+1}(|h|))$ as $|h| \to 0$ and $p$ is a polynomial with even degree. By linearity, we have

$$
K(x+h,x') = \sum_{l=1}^{L} w_l K_l(x+h,x')
$$

$$
= \sum_{l=1}^{L} w_l \left(\sigma_l^2 G_\nu(|\alpha_l h|) - \nu\sigma_l^2 G_{\nu+1}(|\alpha_l h|) + \iota(t)\right)
$$

$$
= \sum_{l=1}^{L} w_l \sigma_l^2 G_\nu(|\alpha_l h|) - \sum_{l=1}^{L} \nu w_l \sigma_l^2 G_{\nu+1}(|\alpha_l h|) + \iota(t),
$$

where $\iota(t) = p(|h|) + o(G_{\nu+1}(|h|))$ as $|h| \to 0$. They by Theorem 4 of Anderes (2010), there exists consistent estimators for $\sum_{l=1}^{L} w_l \sigma_l^2 \alpha_l^{2\nu}$ and $\sum_{l=1}^{L} w_l \sigma_l^2 \alpha_l^{2(\nu+1)}$ when $0 < 2(2(\nu+1)-2\nu) < p$, that is $0 < 4 < p$. This completes the proof. $\qquad\square$

## 6   Proof of Theorem 3

(i) is a direct corollary of Theorem 3 of Bachoc et al. (2022), where $\sigma_{ii}\sigma_{jj}\rho_{ij} = A_{ij}\sigma^2$ and $\alpha_{ij} = \frac{1}{\alpha}$ for any $i,j = 1,\cdots,m$.

Now we prove (ii). Let $\theta$ be the microergodic parameter of $K_0$, that is, $K_0 \equiv \widetilde{K}_0 \Longleftrightarrow \theta = \widetilde{\theta}$, where $\widetilde{K}_0$ is characterized by $\widetilde{\theta}$. Let $\rho_0$ and $\widetilde{\rho}_0$ be the spectral densities of $K$ and $\widetilde{K}_0$, then the matrix

spectral densities of $K$ and $\widetilde{K}$ are $P := A\rho_0$ and $\widetilde{P} := \widetilde{A}\widetilde{\rho}_0$. By the assumption, that is, there exists constants $b$, $c_1$, $c_2 > 0$ and $\gamma \in \mathcal{W}_{[-b,b]^m}$ such that $c_1\gamma^2(\omega) \le \rho_0(\omega) \le c_2\gamma^2(\omega)$, we claim that there exists $q_1$, $q_2 > 0$ such that

$$q_1\gamma^2(\omega)\mathrm{I}_m \le P(\omega) \le q_2\gamma^2(\omega)\mathrm{I}_m, \ \forall \omega \in \mathbb{R}^p. \tag{3}$$

From the assumption,

$$P(\omega) = A\rho_0 \le c_2\gamma^2(\omega)A \le c_2\,\mathrm{eig}_1(A)\gamma^2(\omega),$$

where $\mathrm{eig}_1(A)$ is the largest eigenvalue of $A$. Similarly, $c_1\,\mathrm{eig}_m(A)\gamma^2(\omega)\mathrm{I}_m \le P(\omega)$ where $\mathrm{eig}_m(A) > 0$ is the smallest eigenvalue of $A$, which is positive by the positive definiteness of $A$. Then claim (3) holds where $q_1 := c_1\,\mathrm{eig}_m(A)$ and $q_1 := c_2\,\mathrm{eig}_1(A)$. Now we can prove (ii). If $\theta A = \widetilde{\theta}\widetilde{A}$, then for any $i, j = 1, \cdots, m$,

$$
\begin{aligned}
\int_{\mathbb{R}^p} \frac{1}{\gamma^4(\omega)}\left(P_{ij}(\omega) - \widetilde{P}_{ij}(\omega)\right)^2 \mathrm{d}\omega &= \int_{\mathbb{R}^p} \frac{1}{\gamma^4(\omega)}\left(A_{ij}\rho_0(\omega) - \widetilde{A}_{ij}\widetilde{\rho}_0(\omega)\right)^2 \mathrm{d}\omega \\
&= \int_{\mathbb{R}^p} \frac{1}{\gamma^4(\omega)}\left(\theta A_{ij}\frac{\rho_0(\omega)}{\theta} - \widetilde{\theta}\widetilde{A}_{ij}\frac{\widetilde{\rho}_0(\omega)}{\widetilde{\theta}}\right)^2 \mathrm{d}\omega \\
&= \int_{\mathbb{R}^p} \frac{1}{\gamma^4(\omega)}\left(\theta A_{ij}\frac{\rho_0(\omega)}{\theta} - \theta A_{ij}\frac{\widetilde{\rho}_0(\omega)}{\widetilde{\theta}}\right)^2 \mathrm{d}\omega \\
&= (\theta A_{ij})^2 \int_{\mathbb{R}^p} \frac{1}{\gamma^4(\omega)}\left(\frac{\rho_0(\omega)}{\theta} - \frac{\widetilde{\rho}_0(\omega)}{\widetilde{\theta}}\right)^2 \mathrm{d}\omega \\
&< \infty,
\end{aligned}
$$

where the last inequality holds from the assumption that $\theta$ is the microergodic parameter of $K_0$. As a result, by Theorem 2 of Bachoc et al. (2022), $\theta A$ is the microergodic parameter of $K$.

# 7 Experiment details

## 7.1 Simulation 1

In the first simulation, we examined a mixture kernel defined as follows: $K = \sum_{l=1}^3 w_l K_l$, wherein $K_l$ represents the Matérn kernel with parameters $(\sigma_l^2, \alpha_l, \nu_l)$. For this simulation, we assigned the values $\nu_1 = 1/2, \nu_2 = 3/2, \nu_3 = 5/2$ for the smoothness parameter.

The weights for each kernel component, represented as $(w_1, w_2, w_3)$, were chosen as $(0.03, 0.33, 0.63)$. This set-up presents the case where despite $w_1$ being the smallest, which is associated with the kernel that exhibits the least smoothness, our result further substantiates our claim about the dominance of the kernel with the lowest smoothness over the influence of weights. In terms of the scale parameters, for $(\sigma_1^2, \sigma_2^2, \sigma_3^2)$, we picked $(3, 3, 3)$. We selected $(\alpha_1, \alpha_2, \alpha_3)$ to be $(1, 1, 1)$. This uniform selection across the components aids in isolating the effect of the smoothness and weights in our analysis.

## 7.2 Simulation 2

In the second simulation, we consider the following mixture kernel: $K = \sum_{l=1}^3 w_l K_l$, where $K_l$ is the Matérn kernel with parameter $(\sigma_l^2, \alpha_l, \nu_l)$. For this simulation, the values for the smoothness parameter, $\nu$, were set as $\nu_1 = 1/2, \nu_2 = 3/2, \nu_3 = 5/2$. Regarding the choice of the weight parameters $(w_1, w_2, w_3)$, we selected $(0.1, 0.3, 0.6)$ to reflect the difference in contribution of each kernel to the overall mixture. Although the weight of the Matérn $1/2$ kernel $(w_1)$ is small, it still remains dominant due to its lesser smoothness. This setup allowed us to test our hypothesis that smoothness plays a more significant role than weight in determining the identifiability of parameters. $(\sigma_1^2, \sigma_2^2, \sigma_3^2)$ are set to $(16, 4, 1)$, and $(\alpha_1, \alpha_2, \alpha_3)$ are set to $(4, 2, 1)$. This variation further facilitated the examination of our proposition that the parameters associated with the least smooth kernel converge to their true values, while others do not.

In this scenario, we generate $X \in \mathbb{R}^2$ from an equal-spaced sequence ranging from $-10$ to $10$, with a random variation sampled from $\mathrm{unif}(-\frac{1}{5n}, \frac{1}{5n})$. The sample size, $n$, takes on values from the set

$20, 50, 100, 500$. For each sample size, we replicate the simulation 100 times. Subsequently, $Y$ is simulated from $N(0, K + \epsilon)$. We consider $\epsilon$ as a fixed value ($\epsilon = 0.1$) and include it for numerical robustness. This $\epsilon$ is also added in the training process. For parameter initialization, we initialize $w = (0.2, 0.3, 0.5)$, $\sigma^2 = (5.0067, 10, 15)$ and $\alpha = (0.7615, 0.4702, 0.3280)$. These initial values were close to the true parameters, yet sufficiently distinct to illustrate the efficacy of the learning process. Here we use the SGD optimizer with 0.005 learning rate and 1000 epochs. All parameter estimates are summarised in the Figure S1.

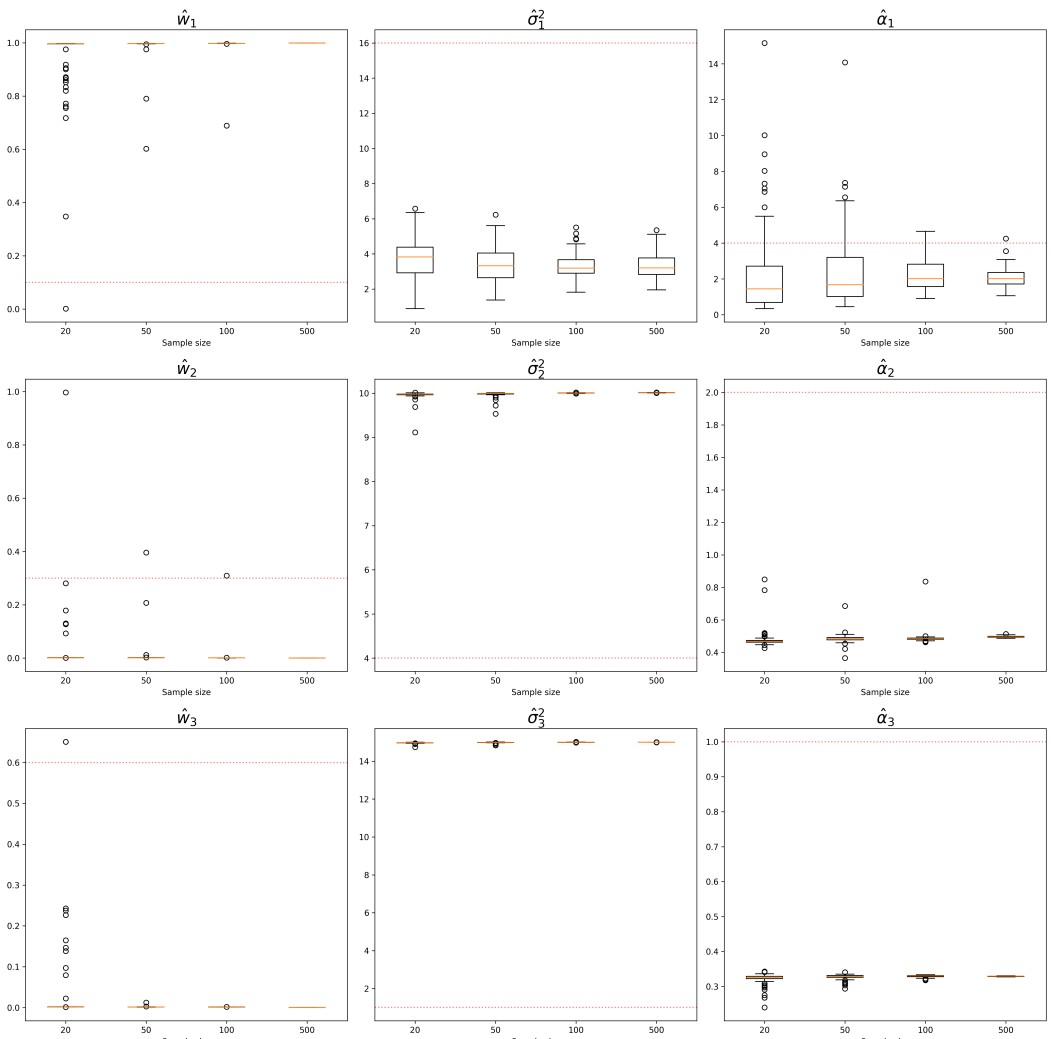

Figure S1: All parameter estimation in simulation 2.

We further examined the simulation in a larger sample size and also used a different optimizer L-BFGS (learning rate 0.5 for $n = 50, 500$, 1.0 for $n = 100$). The results are consistent with the previous results.

## 7.3 Simulation 3

In the third simulation, we consider the following mixture kernel: $K = AK_0$, where $K_0$ is the Matérn kernel with parameter $(\alpha, \sigma^2, \nu)$. For this simulation, we set $\nu = 1/2$. The ground truth for the parameters is assigned as follows: the matrix $A$ is set as $[[5, 1][1, 5]]$, which serves as a symmetric, positive-definite structure to facilitate the properties of the covariance matrix; $\sigma^2$ is set to 10, and $\alpha$ is set to 1.

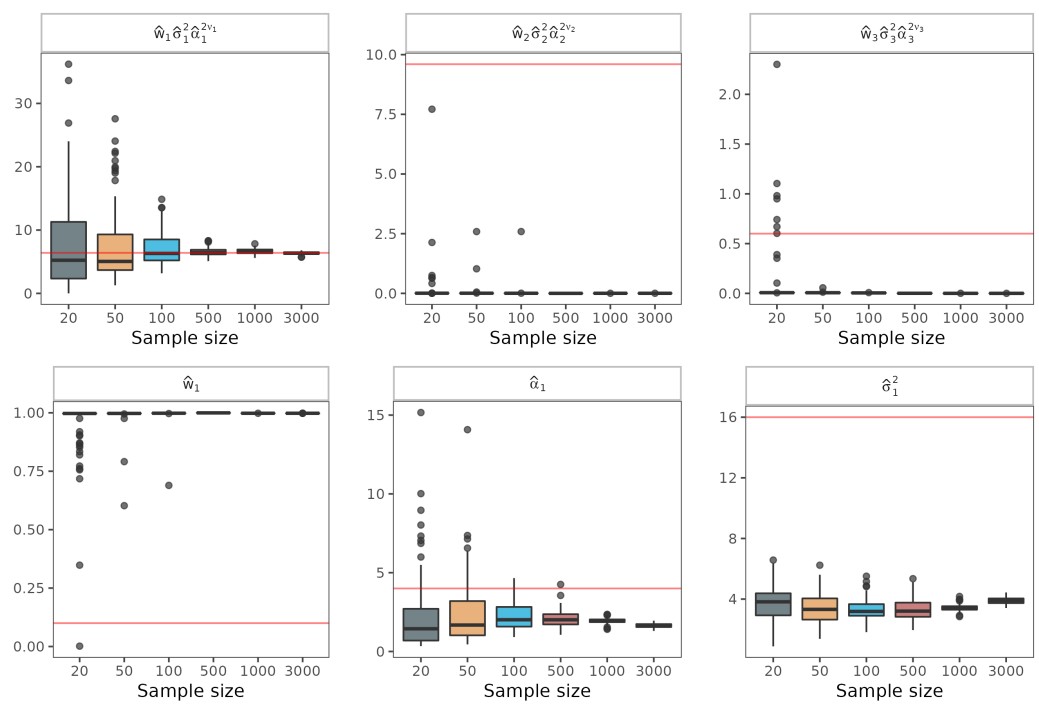

Figure S2: Parameter estimation in simulation 2 with larger sample size.

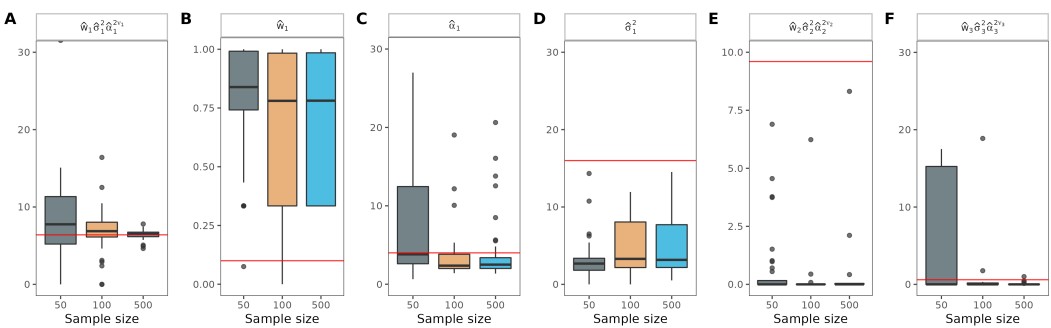

Figure S3: Parameter estimation in simulation 2 with L-BFGS optimizer.

In this simulation, we generate $X \in \mathbb{R}^2$ from an equal-spaced sequence ranging from $-10$ to $10$, with a random variation sampled from $\text{unif}(-\frac{1}{5n}, \frac{1}{5n})$. The sample size, $n$, takes on values from the set $50, 100, 200, 400$. For each sample size, we replicate the simulation 100 times. Subsequently, $Y$ is simulated from $N(0, K + \epsilon)$. We consider $\epsilon$ as a fixed value ($\epsilon = 0.5$). This $\epsilon$ is also added in the training process. In terms of parameter initialization, we elect to start with $A$ as an identity matrix, $\sigma^2 = 1$ and $\alpha = 10$. The initialization of $A$ as an identity matrix ensures a simple, non-informative starting point, while the initial $\sigma^2$ and $\alpha$ are chosen to be significantly distinct from the ground truth to assess the robustness of the learning process. Here we use the SGD optimizer with $0.001$ learning rate and 2000 epochs. The parameter estimates of $A$ are summarised in the Figure S4.

## 7.4 Additional simulation (Simulation 4) for mixture of Matérn kernels with the same smoothness

In this additional simulation 4, we aim to evaluate the parameter identifiability in a GP with a mixture kernel consisting of Matérn kernels with the same smoothness. We denote the mixture as

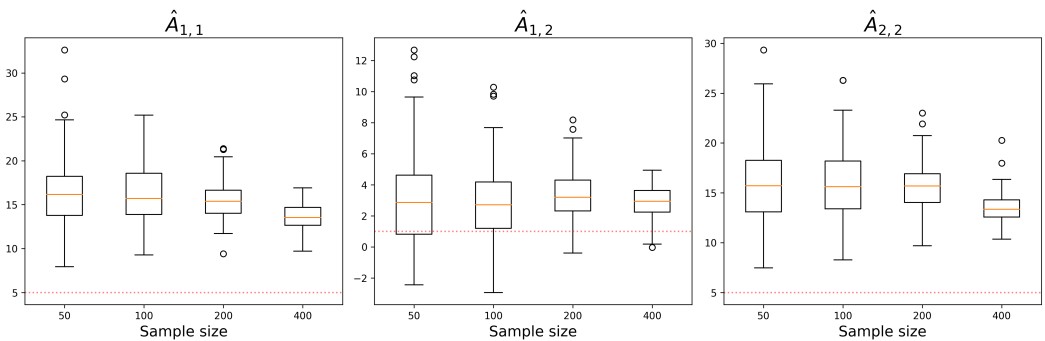

Figure S4: Parameter estimation in simulation 3.

$K = \sum_{l=1}^{L} w_l K_l$, where $K_l$ is a Matérn kernel with parameters $(\sigma_l^2, \alpha_l, \nu)$. Our theorem suggests that only the microergodic parameter $\sum_{l=1}^{L} w_l \sigma_l^2 \alpha_l$ might be identifiable, while other parameters are not identifiable. Consequently, we will evaluate the parameter estimation by comparing it with the true value for different training sample sizes.

Here we use mixture of three Matérn $1/2$ kernels. We generate $X \in \mathbb{R}^2$ from an equal-spaced sequence with values ranging from $-10$ to $10$, with a random variation sampled from $\text{unif}(-\frac{1}{10n}, \frac{1}{10n})$. The sample size $n$ takes values from $20, 50, 100, 500$. For each sample size, we replicate the simulation 100 times. Subsequently, we simulate $Y$ from a normal distribution with a mean of 0 and covariance matrix $K + \epsilon$. We treat $\epsilon$ as a fixed value ($\epsilon = 0.1$) and include it for numerical robustness. This $\epsilon$ is also added in the training process. The ground truth parameters are given as $w = (0.2, 0.3, 0.5)$, $\sigma^2 = (16, 4, 1)$ and $\alpha = (2, 1, 4)$. Here the weight and sigma are set to true value, $(\alpha_1, \alpha_2, \alpha_3)$ are set to $(4, 2, 1)$. The motivation behind our specific parameter initialization strategy is to intentionally introduce some degree of initial discrepancy. This approach allows us to critically observe the convergence behavior of the parameters. Here we use the Adam optimizer with 0.01 learning rate and 1000 epochs.

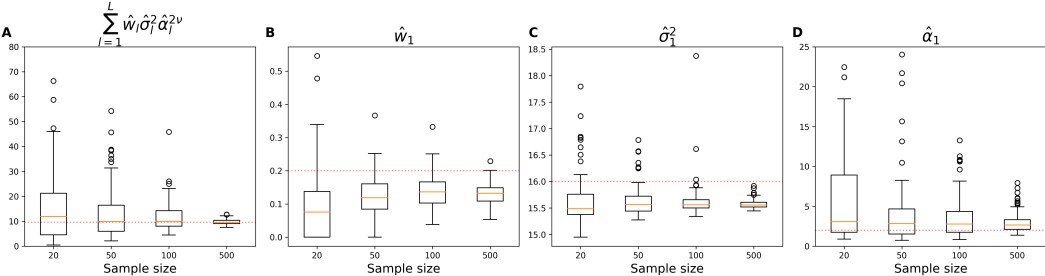

Figure S5: Parameter estimation in simulation 4. (A) Only the microergodic parameter $\sum_{l=1}^{L} w_l \sigma_l^2 \alpha_l$ might be identifiable (B-D) All other parameters $(\sigma_l^2, \alpha_l, w_l)_{l=1}^{L}$ are not identifiable.

Our results offer compelling evidence that, in the case of a mixture kernel consisting of three Matérn $1/2$ kernels, only the MLE of the microergodic parameter $\sum_{l=1}^{L} w_l \sigma_l^2 \alpha_l^{2\nu}$ converges to the true value (Figure S5A). The MLEs of all other parameters do not demonstrate convergence to their corresponding true values (Figure S5B-D). This result underscores the importance of understanding the identifiability of parameters in such mixture kernels and highlights the need for careful consideration when interpreting the estimated parameters in Gaussian process models. When treating mixture kernel consist of same smoothness, we could not interpret every single model. All parameter estimates are summarised in Figure S6.

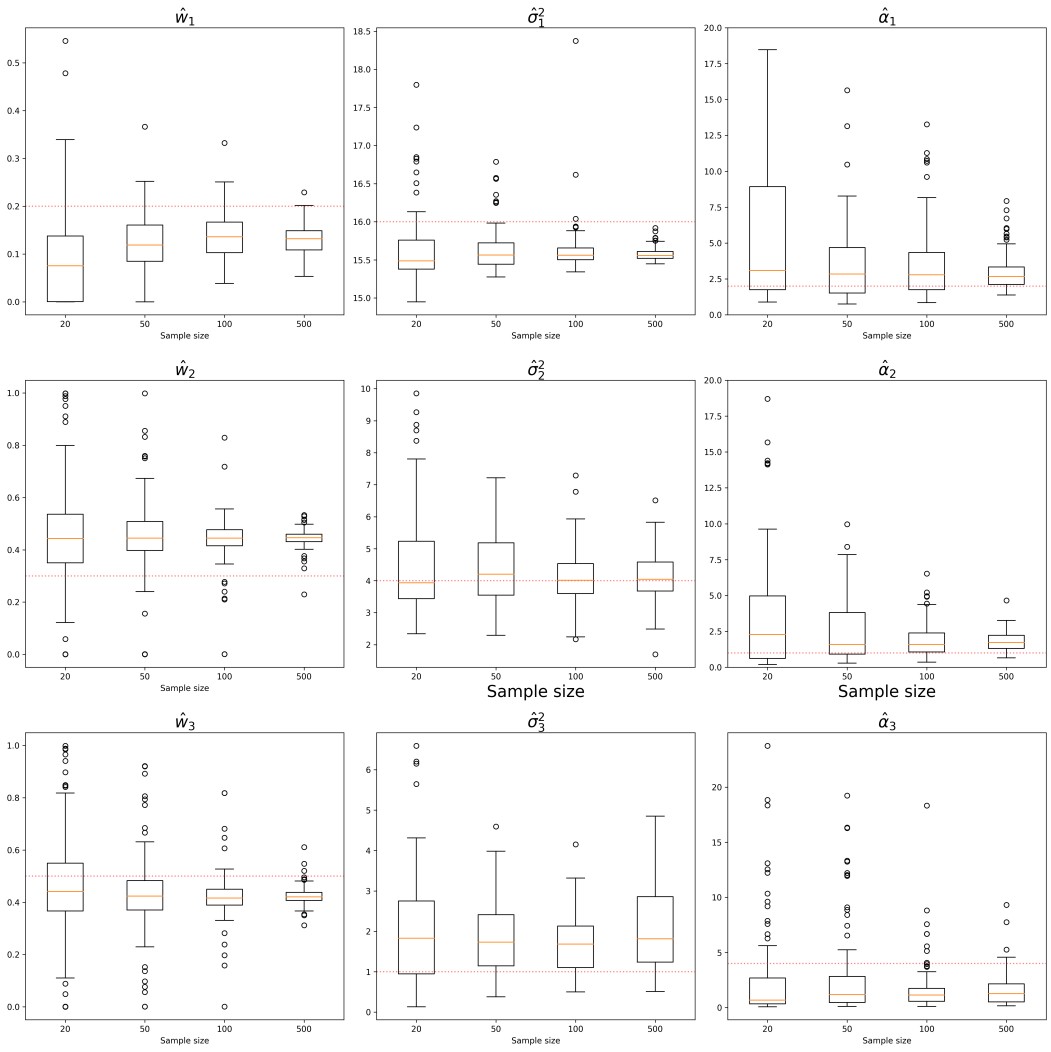

Figure S6: All parameter estimation in simulation 4.

## 7.5 Application 1

In application 1, we focus on image analysis. We employ a hand-written zero from the MNIST dataset (LeCun et al. (1998)). In this application, we use mixture kernel with Matérn kernels of smoothness $1/2$, $3/2$ and $5/2$ and compare its performance with Matérn $1/2$ kernel. For both kernels, we used the SGD optimizer with learning rate $1e^{-05}$. The training epochs are $40000$. During both the training and prediction stages, we introduced a term, $\epsilon$, into the covariance computation to guarantee its positive-definiteness. For both kernels, we set $\epsilon$ to $0.01$. For mixture kernel, the parameters are initialized as $(w_1, w_2, w_3) = (1/3, 1/3, 1/3)$, $(\sigma_1^2, \sigma_2^2, \sigma_3^2) = (1, 1, 1)$ and $(\alpha_1, \alpha_2, \alpha_3) = (1, 1, 1)$.

Here the estimated parameters for mixture kernel are $(w_1, w_2, w_3) = (9.9954e^{-01}, 2.3194e^{-04}, 2.3194e^{-04})$, $(\sigma_1^2, \sigma_2^2, \sigma_3^2) = (1.2148e^{+02}, 4.9960e^{+01}, 4.9256e^{+01})$, $(\alpha_1, \alpha_2, \alpha_3) = (1.3941e^{-01}, 1.1203e^{-02}, 1.1020e^{-02})$. The estimated parameters for Matérn $1/2$ kernel are $\alpha = 3.0639e^{-02}$, $\sigma^2 = 2.7927e^{+02}$.

## 7.6 Application 2

In application 2, we focus on the Moana Loa $CO_2$ dataset (Tans and Keeling (2023)). In this application, we compare the performance of Matérn mixture $1/2 + 3/2$, $1/2 + 3/2 + 5/2$, $3/2 + 5/2$

and three single Matérn kernels ($1/2$, $3/2$, $5/2$). We use sklearn package (Pedregosa et al. (2011)) for this analysis. The the parameters are initialized as $(\sigma_1^2, \sigma_2^2, \sigma_3^2) = (10, 500, 500)$ and $(\alpha_1, \alpha_2, \alpha_3) = (4, 0.1, 0.1)$ for both single kernel and mixture kernel. Other parameters remain default in sklearn.