# OpenReview forum: "On the Identifiability and Interpretability of Gaussian Process Models"
_NeurIPS.cc/2023/Conference — NeurIPS 2023 poster_

### Official Review · Reviewer_j39y · 2023-07-01

**Soundness:** 2 fair
**Presentation:** 3 good
**Contribution:** 3 good
**Rating:** 5
**Confidence:** 3

**Summary:**

The authors carefully examine additive mixture of Matern kernels for Gaussian processes. For the single-output case, the authors show that a mixture of Matern kernels is equivalent to the least smooth component of the mixture (Theorem 2). Consequently, "there is no advantage in including any other component apart from the least smooth component" (Corollary 1). For the multi-output case, the authors leverage results from Bachoc et al. (2022) to identify microergodic parameters of separable multi-output kernels. These theoretical results are accompanied by demonstrations on simulated data. In addition, in three experiments on real-world data, the authors provide evidence that a mixture Matern kernels performs as well as just the least smooth components.

**Strengths:**

I would like to thank the authors for their submission. I enjoyed reading the submission and certainly learned few new things.

## Strengths

* The paper is very well written and overall the presentation is good. I found hardly any typos.

* Additive mixtures of kernels are very important in GP modelling, since it allows the practitioner to model multiple length scales in the data at various levels of smoothness, so a careful assessment of additive mixtures like this work does is important.

* The main result of the paper, Theorem 2, is very interesting and might have important implications. I've read through the proof, and, modulo my comment below in the weaknesses, I think that the proof is otherwise likely alright.

* The experiments appear to nicely support the central conclusion of the paper: "the inclusion of kernels with different smoothness does not necessarily include prediction accuracy" (Section 7). (However, I have doubts about this. Please see below.)

**Weaknesses:**

## Weaknesses in the Method and Theory

### Claim of Theorem 1 Not Fully Supported by the Proof, but a Fix Appears Possible
The claim of Theorem 1 consists of two parts: (1) $K$ is $d$-times MSD, and (2) "so the smoothness of $K$ is determined by the least smooth component". For (2) to follow from (1), it is also necessary that $K$ is _not_ smoother than $d$-times MSD! That is, it is necessary that $K$ is _not_ $(d + 1)$-times MSD.

For functions, consider the following example. Suppose that $f_1$ is differentiable and that $f_2$ is non-differentiable. Is $f_1 + f_2$ then non-differentiable, so the smoothness of the sum is determined by the least smooth component? The answer is no: if $f_1 - f_2$ is non-differentiable, then a counter-example is found with $g_1 = f_1$ and $g_2 = -f_2$; and if $f_1 - f_2$ is differentiable, then a counter-example is found with $g_1 = f_1 - f_2$ and $g_2 = f_2$. Therefore, for sums of functions, the smoothness is _not necessarily_ determined by the least smooth component, because the sum can be smoother than the least smooth component! Although the claim is false for functions, kernels have very different properties, and I think the claim is true in that case.

Hence, to fully support the claim in Theorem 1, it is necessary that the author prove that $K$ is not $(d + 1)$-times MSD, which is currently not shown in the proof. Fortunately, I think that the proof is easily extended by making use of the following property: if $f_1$ is integrable and $f_2$ is non-integrable, then, by $|f_1 + f_2| >= |f_2| - |f_1|$, $f_1 + f_2$ is also non-integrable.

### Question About Proof of Theorem 2

In the proof of Theorem 2, the fourth inequality on line 25 of the appendix seems to use the following property:

$ \frac{f(\omega) + o(f(\omega))}{g(\omega) + o(g(\omega))} = \frac{f(\omega)}{g(\omega)}(1 + O(\|\omega\|^{-2})) $

where $f(\omega)$ and $g(\omega)$ are the first terms of the particular mixtures of $p_l$ and $\tilde p_l$. Now,

$ \left| \frac{(f(\omega) + o(f(\omega)))/(g(\omega) + o(g(\omega)))}{f(\omega)/g(\omega)} - 1\right| = \left|\frac{1 + o(1)}{1 + o(1)}\right| = |o(1)|$,

so

$ \frac{f(\omega) + o(f(\omega))}{g(\omega) + o(g(\omega))} = \frac{f(\omega)}{g(\omega)}(1 + o(1)) $,

which is a weaker result! If you're a little more careful, instead of $o(1)$ you might get a $O(\|\omega\|^{-q})$ where $q$ depends on the ratio of the leading term and the second term in the particular mixtures of $p_l$ and $\tilde p_l$. Importantly, $q$ might be very small, smaller than $2$. It's not at all obvious to me that $q=2$ is true! This may seem like a minor detail, but it could mean that the integrand actually depends on $O(\|\omega\|^{-2q})$ for $q$ potentially very small, which would affect the whether the integrand is integrable or not.

I would like the authors to reply to this question. Since Theorem 2 is the central result of the paper, it is important that the proof is bulletproof.

### I Disagree With Corollary 1

Corollary 1 states that "there is no advantage in including any other components apart from the least smooth component".
I strongly disagree with this conclusion, because I think that it misinterprets the concept of equivalence.

If two GPs are equivalent, then they produce _asymptotically_ similar predictions in the limit of infinite data under fixed-domain asymptotics.
_However_, for finite data, equivalent GPs may give different predictions, and this difference may be practically very significant!
Therefore, to conclude that "there is no advantage in including any other components apart from the least smooth component", in my opinion, is wrong.
Let me give a simple counter-example.

Let $k_1$ be a Matern-$1/2$ kernel with length scale $1$ and let $k_2$ a Matern-$3/2$ kernel with length scale $1000$.
Consider the mixture $k = k_1 + k_2$, and suppose that data is generated from $k$.
Suppose that we have $3$ observations randomly sampled from $[0, 2000]$.

If we make predictions with just the least smooth component, $k_1$, then, because of the short length scale of $k_1$, the predictive mean is basically zero on all of $[0, 2000]$ except near the observations.
On the other hand, if we make predictions with the mixture $k$, then, because of the long length scale of $k_2$, the predictions are much more reasonable.

(You can concoct a similar scenario by considering a Matern-$1/2$ kernel with length scale $1$ and a Matern-$v$ kernel for very high $v$ (so basically a squared-exponential kernel) also with length scale $1$. For finite data, the difference in smoothness will give very different predictive uncertainty.)

### Attribution of Theorem 3

In the main body, the authors present Theorem 3 as if it is a completely novel result.
However, Theorem 3.(i) is basically Theorem 3 by Bachoc et al. (2022), and Theorem (3).(ii) follows nearly immediately from Theorem 2 by Bachoc et al. (2022).
The authors acknowledge this in the proof in the supplement, but they should really state this attribution in the main body.
Because of the similarity to the theorems by Bachoc et al., I consider Theorem 3 by the authors only a minor contribution.

## Weaknesses in Experiments

* In Simulation 1, the authors state that the "it is evident that the mixture kernel demonstrates a degree of smoothnes similar to the Matern kernels with $\nu = 1/2$". Although I think that the result is true, I think that this is not necessarily so obvious from Figure 1. To really determine the smoothness of the samples of the mixture kernel, the authors should provide more conclusive evidence, e.g. by numerically determining whether the function is differentiable: numerically compute $|f(x) - f(x + h)|/|h|^{\alpha}$ for $h \to 0$ and _e.g._ $\alpha = 1$ (for once differentiability).

* In Simulation 2, the authors demonstrate that $w_1 \sigma_1^2 \alpha_1$ is identifiable and they claim that _only_ this parameter is identifiable. To really give evidence that only  $w_1 \sigma_1^2 \alpha_1$ is identifiable, the authors should show the parameters that would be microergodic if you would consider the kernels separately: $w_2 \sigma_2^2 \alpha_2$ and $w_3 \sigma_3^2 \alpha_3$. These parameters are also not shown in Figure S1. Without showing these additional parameters, I'm not convinced that the figure really prove that only  $w_1 \sigma_1^2 \alpha_1$ is identifiable.

* Throughout the real-world experiments, the authors use Adam/SGD. I wonder why the authors wouldn't just run L-BFGS-B until convergence at a reasonable tolerance. This would likely eliminate some difficulties in optimising the mixture kernel. In addition, the supplement says that the kernel matrices are _heavily_ regularised by a diagonal of magnitude $0.01$ or $0.1$. I think that is _much more_ than should be necessary if one uses double precision. I'm not sure if it would change any results, but it does make me suspicious.

* In Application 1, the authors compare a Matern mixture kernel against a Matern-1/2 kernel to perform texture extrapolation. Although I don't disagree with the result that the mixture Matern kernel performs similar to the Matern-1/2 kernel, I think both the mixture kernel and the Matern-1/2 kernel are a very poor fit for the data, which can be seen from the prediction in Figure 4. The point of these textures is that extrapolation is possible because they are periodic, so one should really use a (weakly) periodic kernel. I understand that Theorem 2 does not apply to mixtures of periodic Matern kernels, just to regular Matern kernels, but to me that suggests that this is not a good data set to demonstrate Theorem 2 on.

### Experiments Do Not Strongly Support Main Claim

My main issue with the experimental section is that the authors take some data sets, compare a mixture of Matern kernels versus a Matern-1/2 kernel, observe that the performance of the two is very similar, and conclude that, generally, there might be no advantage in adding additional components to the mixture. In my opinion, this is not the right conclusion. The right conclusion is that either (a) generally, there might be no advantage in adding additional components to the mixture; or (b) specifically in the chosen experiments, there is no advantage in adding additional components to the mixture.

_To really conclude (a), the authors should compare the mixture kernel to the Matern-1/2 kernel on a data set where conventially one would agree that a Matern mixture kernel would be a good choice._ An example of such a data set is a time series that has a noisy short-length-scale components and a smooth long-length-scale component. In constrast, economical data sets such as prices are traditionally modelled with an Ornstein-Uhlenbeck process (Matern-1/2 kernel), so I'm not surprised additional smooth kernels won't help there. Moreover, if you visualise the gene expression data, you will see that the data is _very_ erratic and definitely doesn't contain and smooth components. Therefore, I am not at all convinced that the authors have made a good case for (a).

**Questions:**

## Conclusion

I think that this is great submission with very interesting results. However, in my opinion the authors have not provided sufficient evidence for the central message of the paper, which is that the inclusion of kernels with different length scales does not improve prediction accuracy. (Please correct me if this is not the central message of the paper.) The authors have not provided sufficient evidence because of the following two reasons:

* I believe that the conclusion in Corollary 1 is not justified. Please see the reasoning above.

* I believe that the data sets chosen in the experiments do not contain any smooth components, so obviously modelling smooth components in the data will not get you any improvement in predictive accuracy. Please see the foregoing section. Therefore, I think that the experimental results do not support the central message of the paper.

In addition, the proof of Theorem 2 might have an error (please point out whether I'm right or wrong about this!), Simulations 1 and 2 (please see above) have flaws, and I think that Theorem 3 is only a minor contribution. If I add all this up, I'm afraid that I must recommend rejection at this point.

_However_, I'm very willing to debate the above arguments and revise my score if necessary. Therefore, authors, please carefully argue the above, because I would love to be wrong and change my recommendation to accept.

EDIT

In the discussion below, the authors have outlined changes that address the above shortcomings. These changes are substantial. Since uploading revisions is not possible, normally I would recommend that the revised PDF go through another round of review. However, I feel that the authors and I are now aligned sufficiently well, so this time I am willing to recommend a "borderline accept".

I should emphasise that I believe that this submission likely deserves a better score than "borderline accept", but I am not willing to recommend any higher score before reviewing the revised PDF.

**Limitations:**

See above.

---

> ### Author Rebuttal · Authors · 2023-08-09
>
>
> We sincerely thank the reviewer for the meticulous and insightful examination of our work. Your comprehensive feedback has been instrumental in enhancing the quality and clarity of our paper. In response, we have addressed each of the concerns raised and revised the manuscript accordingly. While we are unable to upload the revised manuscript at this juncture, please find below our point-by-point response to your comments. Additional figures that further elucidate our points are provided in the attached 1-page PDF for rebuttal. We deeply appreciate the time and effort you've invested in guiding our revisions.
>
> **About Theorem 1**
>
> A: Thank you for your keen observation. It's pivotal to also ensure that the kernel doesn't exceed this smoothness level. Your suggestion aligns with the proof's direction: since $\rho_1(\omega)\omega^{2d+2}$ is non-integrable, it implies that $\rho(\omega)\omega^{2d+2}$ is similarly non-integrable.
>
> Your example effectively distinguishes the differentiability of a deterministic function and mean-squared differentiability of random functions. We value your comprehensive feedback and have provided a clear proof of Theorem 1 in our revised paper.
>
> **About Theorem 2**
>
> A: You've identified a crucial detail. We omitted a key assumption about smoothness: $\nu_{l+1}-\nu_l\geq 1$.  This assumption is reasonable since if two consecutive $\nu$’s are too close, then the corresponding kernel components $K_1$ and $K_2$ admit the same smoothness, negating the need for a mixed kernel (see Theorem 4 in the supplement).
> As detailed in line 23 in the supplement, the order of the leading term $p_1(w)$ is $2\gamma_1$, while the order of all other terms, $2\gamma_2,\cdots,2\gamma_L$, are all smaller than $2\gamma_1$ by at least 2. As a result, if we divide both the numerator and the denominator by $\|\omega\|^{2\gamma_1}$, the denominator becomes
> $\sum_{l}{w_l}{\sigma}_l^2{\alpha}_l^{2\nu_l}{p}_l(\omega)/\|\omega\|^{2\gamma_1}=w_1\sigma_1^2\alpha_1^{2\nu_1}+O(\|\omega\|^{-2})$.
> Applying the same trick to the numerator, the ratio simplifies to $\frac{\widetilde{w}_1\widetilde{\sigma}_1^2\widetilde{\alpha}_1^{2\nu_1}}{{w_1}{\sigma}_1^2{\alpha}_1^{2\nu_1}}(1+O(\|\omega\|^{-2}))$.
>
>
> **About Corollary 1**
>
> A: We agree with your perspective on the significant difference between a model's asymptotic behavior and its finite sample performance.  Corollary 1 emphasized asymptotic equivalence, possibly downplaying the importance of finite sample scenarios. Our revision clarifies this.
>
> **About Theorem 3**
>
> A: We recognize the need for proper attribution. While Theorem 3.(i) is indeed a direct corollary of the result by Bachoc et al., Theorem 3.(ii) is a distinct extension requiring nuanced verification. We have revised the presentation of Theorem 3 to explicitly mention its relationship to Bachoc et al.
>
>
> **About Simulation 1**
>
> A: Thank you for the insight. In our context, the differentiability of GP is in the mean-square sense (Definition 3). The numerical difference of the sample path doesn’t inherently imply mean-square differentiability. Our method (Figure 6 in the 1-page PDF) appears novel and involves the following steps:
>
> For a given fixed $x_0=0$, let $x_i=1/i$ for $i=1,2,…$, and we generate $y_i^l$ from the GP, where $l=1,…,T$ denotes the index of replicates. This allows us to approximate $\lim_{x\to 0}\mathbb{E}(f(x)-f(0))^2$ with $\beta_i=\frac{1}{T}\sum_{l=1}^T (y_i^l-y_0^l)^2$. As per Definition 3, the GP is continuous if and only if $\beta_i\to0$.
>  Similarly, we can approximate $\lim_{x\to 0}\mathbb{E}\left(\frac{f(x)-f(0)}{x}\right)^2$ with $\gamma_i=\frac{1}{T}\sum_{l=1}^T \left(\frac{y_i^l-y_0^l}{x_i}\right)^2$. The GP is mean-square differentiable if and only if $\lim_{i\to\infty}\gamma_i $ exists.
>
> Specifically, for the mixture of Matérn 1/2 and 3/2 (the first column) and Matérn 1/2 (the second column), while $\beta_i\to0$, $\gamma_i$ does not converge. However, for Matérn $3/2$ (the third column), $\beta_i\to0$ and $\gamma_i$ converges,. This empirical evidence bolsters the claims made in Theorem 1.
>
> **About Simulation 2**
>
> A: Figure 7 in the 1-page PDF now includes two panels showing inconsistencies of $w_2\sigma^2_2\alpha_2^{2\nu_2}$ and $w_3\sigma^2_3\alpha_3^{2\nu_3}$.
>
> **About  L-BFGS**
>
> A: We appreciate the reviewer's keen observation regarding the use of the L-BFGS optimizer in our real-world experiments. Following the GPyTorch's guideline, our initial approach employed the Adam and SGD optimizers. However, upon recommendation, we conducted the analysis again using the L-BFGS optimizer.  This adjustment enhanced our optimization process, enabling us to diminish the diagonal magnitude to $1e-04$ (Figure 7 in the 1-page PDF was obtained by L-BFGS). It reaffirms our manuscript's conclusion that only the term $w_1\sigma^2_1\alpha_1^{2\nu_1}$ is identifiable.
>
> **About Application 1**
>
> A: We have conducted a new experiment utilizing an MNIST image (Figure 4 in the 1-page PDF) due to its absence of periodic patterns. The results demonstrate pleasing prediction accuracy for Matérn kernels, thereby reinforcing our understanding of their applicability and effectiveness in this particular context.
>
> **About experiments do not strongly support main claim.**
>
> A: We are in full agreement with the suggestion that we should include a dataset with a smooth component. Also suggested by reviewer dNVU, we conducted a new analysis on the Mauna Loa $CO_2$ dataset (Figure 3 in the 1-page PDF). In this context, the Matérn $3/2$ appears to exhibit better prediction accuracy compare to Matérn $1/2$. However, we observe similar performance between the Matérn mixture $1/2+3/2+5/2$ and Matérn $1/2$, as well as the resemblance between Matérn mixture $3/2+5/2$ and Matérn $3/2$. This observation further substantiates our argument that the performance of a Matérn mixture kernel is primarily dictated by the least smooth Matérn kernel within it.

---

> > ### Comment · Reviewer_j39y · 2023-08-10
> >
> > Thank you for your detailed reply to my review. I appreciate the effort that very clearly went into writing the rebuttal.
> >
> > I am afraid that I continue to disagree with the main message of the paper. Specifically, I continue to disagree with Corollary 1, which sets the tone of the paper, namely that there is "there is no advantage in including any other components apart from the least smooth component". Whereas this statement is a very particular statement about asymptotics, in the non-asymptotic setting, such as practical applications, in my opinion, there is most definitely benefit in including more components than the least smooth component. Please see the silly but relevant example in my review.
> >
> > (In addition, as a minor point, theoretical results that show that equivalent kernels cannot be distinguished, as far as I know, rely on fixed-domain asymptotics. What about increasing domain asymptotics, which is a very reasonable setting?)
> >
> > For the experiments, I would again like to emphasise the following point:
> >
> > > _To really conclude (a), the authors should compare the mixture kernel to the Matern-1/2 kernel on a data set where conventially one would agree that a Matern mixture kernel would be a good choice._ An example of such a data set is a time series that has a noisy short-length-scale components and a smooth long-length-scale component.
> >
> > I appreciate that you've added in an experiment on the Mauna Loa data. However, the point is not that you should have added a data set where the data is smooth, the point is that you should have added in a data set where conventionally a Matern mixture kernel would be a good choice, and a time series data set with a noisy short-length-scale component and a smooth long-length-scale component is a good choice.
> >
> > For just year 2022 of the Mauna Loa data, this doesn't hold, because there is roughly only one length scale in the data: the sinusoidal like yearly pattern.
> >
> > I would be interested to see the same comparison on 50 years of Mauna Loa data, with the $(\nu=\frac12)$-component initialised to 4 months and the $(\nu=\frac32)$-component initialised to 10 years. In this setting, compare extrapolation on the next 5 years between a $(\nu=\frac12)$-kernel and a mixture of a $(\nu=\frac12)$-kernel and a $(\nu=\frac32)$-kernel. I'm fairly convinced that the additional smooth component would yield substantially better extrapolations due to its ability to model the smooth long-length-scale component.
> >
> > Although the paper is very well written and the technical contributions are impressive, I unfortunately still think the paper's main message is not sufficiently supported by the theoretical results and experiment.

---

> > > ### Author Response · Authors · 2023-08-10
> > > **Clarifications and Further Experiments in Response to Reviewer j39y's Feedback**
> > >
> > > Thank you for your swift and insightful response. In line with your suggestions, we've conducted additional experiments and addressed the areas of concern you highlighted. Given the platform constraints, we've presented numerical results (MSEs) in this discussion. However, should it aid in your evaluation, we are more than willing to upload the associated figures and code. Your thorough feedback has been invaluable in enhancing the rigor of our work.
> > >
> > > **About Corollary 1**
> > >
> > > A: We deeply value your insights and wholeheartedly concur with your perspective on Corollary 1. The nuance was initially captured in our detailed rebuttal; however, character limitations led to unintended truncations. You've aptly highlighted our asymptotic focus. To better reflect this nuance, we've revised Corollary 1 with a follow-up note:
> > >
> > > Corollary 1: The mixing kernel $K$ is equivalent to $K_1$.
> > >
> > > Note: In an asymptotic context, particularly with a large sample size, there's limited benefit in incorporating smoother components beyond the least smooth one. Nonetheless, for smaller sample sizes, including additional components can offer advantages depending on the specific scenario.
> > >
> > > **About fixed domain**
> > >
> > > A: You've pinpointed a crucial distinction. Indeed, our investigation was anchored in the context of fixed-domain scenarios. The realm of increasing domain (or forecasting) presents a more intricate landscape and has seen relatively less exploration compared to its fixed counterpart. It's imperative to clarify that all assertions, simulations, and empirical data examples we presented adhere to this fixed domain paradigm. That said, we acknowledge the undeniable significance of increasing domain scenarios. This very topic is on our research horizon, and we aim to delve deeper into it in subsequent works. We have incorporated this limitation in our discussion section in the revised manuscript.
> > >
> > >
> > > **About experiment**
> > >
> > > A: We appreciate your suggestion and regret any misunderstanding in our initial response. Acting upon your recommendation, we undertook the following experiment: We employed Matern 1/2, Matern 3/2, and a mixture of Matern 1/2 and 3/2 on the Mauna Loa dataset spanning from 1972 to 1981, forecasting data from 1982-1987.  To provide a more discerning contrast between interpolation and extrapolation, we conducted a random split of the data from 1972 to 1987 and used one half to predict the other 50\%. Over an extended timespan of 15 years, Mauna Loa dataset displays both a noisy short-length-scale component and a smooth long-length-scale component, precisely as you highlighted. Consequently, this dataset serves as an example "where conventionally a Matern mixture kernel would be a good choice. The MSEs are summarized in the following table:
> > >
> > >                       Matern 1/2    Matern 3/2    Mixture of 1/2 and 3/2
> > >     Extrapolation      6199.689      89770.53           1360.280
> > >     Interpolation      0.6404151     41.05647           1.242232
> > >
> > > These results lend weight to your assertion that the inclusion of a smoother component is beneficial for extrapolation. Concurrently, it solidifies our stance on the veracity of fixed-domain predictions and accentuates the nuances distinguishing fixed from increasing domain scenarios.
> > >
> > > To emphasize, all claims within our manuscript pertain to the fixed domain (interpolation) in the asymptotic framework. That said, we concur with you on the significance of exploring the increasing domain (extrapolation or forecasting) and the finite sample setting in future endeavors.
> > >
> > > We trust this addresses any reservations you may have had regarding Theorem 2 and Corollary 1. Moreover, we'd like to underline that our work extends beyond just these elements. For instance, Theorem 1, along with the illustrative experiment presented in Figure 6 of the 1-page PDF, both stand as novel contributions to the field. This visualization compellingly demonstrates mean-square continuity and differentiability. We'd like to particularly acknowledge and thank you for suggesting this experiment; it greatly enriches the depth and clarity of our work. Meanwhile, Theorem 3 focuses on multi-output (multivariate, multitask) GPs, endorsing the utilization of the multiplicative (separable) kernel—a prevalent approach in contemporary machine learning research.
> > >
> > > While our paper encompasses multiple facets and findings, we trust that our clarifications and the highlighted novelty of our contributions shed more light on the value of our work. We're deeply grateful for your keen insights and constructive feedback, which have significantly enriched our study. We are eager to address any further questions you might have and are open to conducting additional experiments during this discussion period. We hope that our continued dialogue and these enhancements can be viewed favorably in your evaluation.

---

> > > > ### Comment · Reviewer_j39y · 2023-08-11
> > > > **Reply to Part 1**
> > > >
> > > > Dear authors,
> > > >
> > > > Thanks again for getting back to me.
> > > >
> > > > I appreciate the additional clarifications surround Corollary 1 and increasing-domain asymptotics, so thank you for that.
> > > >
> > > > > It's imperative to clarify that all assertions, simulations, and empirical data examples we presented adhere to this fixed domain paradigm.
> > > >
> > > > Could you explain why you think that the experiments are presented in the fixed-domain paradigm? Especially for time series we're often in the increasing-domain setting.
> > > >
> > > > **About experiment**
> > > >
> > > > I've taken some time to conduct the Mauna Loa experiment myself. I am using all Mauna Loa data between 1960 and 2020, and choose a random subset of 25% for the training data. The remaining 75% is held-out test data. I've chosen a large test set to exaggerate the point. Note I'm chosen the test data randomly, so this interpolation, not extrapolation. I train a 1/2 kernel and a 1/2+3/2 kernel on the training data and then evaluate on the test data. The hyperparameters are optimised by running L-BFGS-B until convergence. In both cases I've verified that the posterior predictions fit the data well. The results are as follows:
> > > >
> > > > ```
> > > > Initial parameters for 1/2:
> > > >     mat_12.scale:         0.25
> > > >     mat_12.var:           10.0
> > > >     noise:                0.5
> > > > Learned parameters for 1/2:
> > > >     mat_12.scale:         133.1
> > > >     mat_12.var:           1.816e+03
> > > >     noise:                2.555e-06
> > > > Initial parameters for 1/2 + 3/2:
> > > >     mat_12.scale:         0.25
> > > >     mat_12.var:           10.0
> > > >     mat_32.scale:         10.0
> > > >     mat_32.var:           500.0
> > > >     noise:                0.5
> > > > Learned parameters for 1/2 + 3/2:
> > > >     mat_12.scale:         0.2102
> > > >     mat_12.var:           4.502
> > > >     mat_32.scale:         295.3
> > > >     mat_32.var:           2.937e+04
> > > >     noise:                6.362e-07
> > > > Log-pdf (train):
> > > >     1/2:                  -491.7
> > > >     1/2 + 3/2:            -456.0
> > > > Log-pdf (test):
> > > >     1/2:                  -947.7
> > > >     1/2 + 3/2:            -934.3
> > > > RMSE (test):
> > > >     1/2:                  2.1639
> > > >     1/2 + 3/2:            1.7994
> > > > ```
> > > >
> > > > (Apologies for the messy output.)
> > > >
> > > > The above shows that the training log-marginal likelihood, test log-marginal likelihood, and  RMSE are better for the 1/2 + 3/2 model.
> > > >
> > > > I've not repeated the experiment with data generated from a mixture of Matern kernels, but I would suspect that the results would be similar.
> > > >
> > > > Would you be able to comment on the above?
> > > >
> > > > All in all, I continue to believe that the story is not right: You prove very interesting results about equivalence of GPs, and then argue that this has significant implications for practical applications. Whereas I do not doubt the significance of the theoretical results, I disagree with the implications that the paper draws for practice, i.e. that there wouldn't be an advantage in including any other components apart from the least smooth component.

---

> > > > > ### Author Response · Authors · 2023-08-11
> > > > > **Clarifications and Further Experiments for Reviewer j39y's Feedback to Part 1**
> > > > >
> > > > > Thank you for your prompt feedback and dedication to our submission. We genuinely value your thoroughness and are particularly grateful for the effort you took in running your own experiments on our work.
> > > > >
> > > > > I believe there might be a slight disconnect regarding the central theme of our paper. Our primary focus, as stated in the paper title and abstract, centers on the identifiability and interpretability of mixture kernels in GP models. Theorem 1 delves into the smoothness of the mixture kernel, while Theorems 2 and 3 examine the identifiability of specific parameters. In Theorems 1-3 and the revised Corollary 1, we haven't ventured into making strong assertions concerning prediction accuracy.
> > > > >
> > > > > It's worth noting that existing literature on GP parameter identifiability, from sources such as Zhang (JASA 2004), Anderes (AoS 2010), Kaufman, Shaby (Biometrika 2013), Tang, Zhang, Banerjee (JRSSB 2022), Li, Tang, Banerjee (JMLR 2023), Loh, Sun (Bernoulli 2023), often sidestep real data examples. This trend acknowledges the inherent gap between theory and real-world data, relying more on simulation studies. Our contribution, encapsulated in three simulation studies from the original manuscript, four in the first rebuttal, and two recent simulations upon your recommendation, stands as a testament to our theory. We genuinely believe that our comprehensive simulations hold significant merit, especially when juxtaposed with existing literature.
> > > > >
> > > > > To clarify, we don’t strictly oppose adding a smoother component in real applications. The trends observed in our Figure 5B in our manuscript do indicate occasional advantages in including the Matern 3/2, but the overall benefits, especially for the dataset in question, aren't consistently substantial.
> > > > >
> > > > > Regarding your additional experiment, although we admit the possible gap between theory and real data, we didn't observe a significant contradiction. We fully trust your experimental outcomes and concur, in line with our Figure 5B, that integrating Matern 3/2 might enhance predictions. Taking into account the inherent randomness in train/test splits, we meticulously mirrored your methodology. Allocating 25\% for training and 75\% for testing from the 1960-2020 period, we performed 20 replications. The mean and standard deviation of RMSEs are in the following table:
> > > > >
> > > > >            Matern 1/2            Mixture
> > > > >     RMSE 3.2549 (0.5713)     4.1934 (1.6716)
> > > > >
> > > > > In the 20 replicates, the mixture kernel outperformed Matern 1/2 in 8 of those instances. This means our findings are in alignment with yours. This fluctuating performance echoes patterns highlighted in our Figure 5B. We endeavored to replicate the RMSE values (2.1639, 1.7994) you provided, but to no avail. Would you be able to furnish us with more specific experimental details: whether you utilized monthly or annual data, learning rate, software or package, and whether jitter was included, among other factors.
> > > > >
> > > > > Similarly, in our most recent simulation, where we generated data from a mixture of Matern 1/2 and 3/2, a single trial might show a lower RMSE for the mixture kernel due to the randomness when generating the data. However, upon multiple replicates, the overall performance from both the Matern 1/2 and the mixture does not show a significant difference, see the table below:
> > > > >
> > > > >               Matern 1/2    Mixture
> > > > >     RMSE 5.6320 (0.5840)  5.6337 (0.5843)
> > > > >
> > > > > To further support Theorem 2, we simulated data from a mixture of Matern $1/2$ and $3/2$. We utilized the true parameters for predictions to alleviate any concerns regarding optimizer choice, initialization, local maxima, early stopping, or other potential influencing factors. Given our use of the true model, it represents the optimal prediction setup. Additionally, we executed predictions solely from Matern 1/2,  The resulting RMSEs are detailed in the table below:
> > > > >
> > > > >               Matern 1/2    Mixture
> > > > >     RMSE  5.3874 (0.5441) 5.3573 (0.5689)
> > > > >
> > > > > We are not opposed to the use of the mixture kernel, if the users' target is prediction RMSE. Our emphasis, however, remains rooted in the theoretical principles of identifiability and interpretability.
> > > > >
> > > > > In summary, we'd like to emphasize that our paper does not posit that smoother components are entirely redundant in real-world applications. In fact, your conclusion aligns well with ours. We are open to refining our manuscript to better clarify the practical implications of our theorem and would even consider scaling back on application contexts if you deem it beneficial. Importantly, our paper's essence lies in the exploration of theoretical nuances such as identifiability and interpretability, rather than purely contrasting RMSEs. We trust that this detailed explanation clarifies our stance. If you'd consider revisiting our work in light of these insights, we'd be immensely grateful. As always, we are open to continued discussions and prepared to conduct any additional experiments you may recommend during our review discussions.

---

> > > > > > ### Author Response · Authors · 2023-08-11
> > > > > > **Clarifications for fixed domain examples**
> > > > > >
> > > > > > Due to character constraints, we've segmented our response on the fixed domain versus increasing domain topic into this separate message. We appreciate your understanding and patience, and thank you for your invaluable feedback.
> > > > > >
> > > > > > To start, we'd like to clarify that our entire work, spanning theory, simulation, and real data application, operates within the fixed domain. The distinction between the fixed and increasing domains arises from the challenges they present and the respective volume of literature available for each.
> > > > > >
> > > > > > Fixed-Domain (our focus): In a fixed-domain setting, the spatial region of interest remains unchanged. This simplicity makes it easier to work with mathematical and statistical models, leading to more comprehensive theoretical results. Many real-world problems, especially in spatial statistics, lie within a fixed domain. Given the aforementioned reasons, there's a rich body of literature on GPs in the parameter inference fixed-domain setting. However, existing studies haven't covered the mixture kernel.
> > > > > >
> > > > > > Increasing-Domain: When the domain of interest grows (e.g., as time evolves in time series), the statistical and mathematical challenges multiply. For instance, considering how covariance functions evolve as the domain increases can be intricate. Due to its inherent complexities, the increasing-domain perspective is less studied in the GP literature. However, it's crucial for applications where the temporal domain naturally expands, like forecasting/predicting the future from the past data in time-series.
> > > > > >
> > > > > >
> > > > > > Detailing Our Work:
> > > > > >
> > > > > > Theory: all our theories were focused on fixed-domain setting due to the above reasons.
> > > > > >
> > > > > > Simulations: All our simulations were crafted within a fixed domain. Even as we scaled the sample size, the domain remained constant.
> > > > > >
> > > > > > Application 1 (Image) data: Here, our input domain originates from image data, naturally making it a fixed domain where pixel locations are constants.
> > > > > >
> > > > > > Application 2 (Boston housing): Using the LSTAT variable as input and the MEDV variable as output is based on the established correlation between them, as documented by Harrison and Rubinfeld (1978). It's essential to note that Boston housing isn't time-series data, thus reinforcing our fixed-domain stance. Specifically, the domain of LSTAT lies within [0,1].
> > > > > >
> > > > > > Application 3 (Genomics): Gene expression levels, as discussed by Stark, Grzelak, and Hadfield (Nature Reviews Genetics, 2019), predominantly occupy a compact space, shaped by sequencing depth and biological factors. This again places our study within a fixed domain.
> > > > > >
> > > > > > Additional application during rebuttal (Mauna Loa): Our foray into time-series data, guided by reviewer recommendations, was solely with the Mauna Loa dataset during the rebuttal phase. While the increasing-domain is applicable to the data, all our analyses, except the forecasting task you requested, fall within the fixed-domain. Such tasks in time series data can also be seen as imputing missing data. It's important to reiterate that our original manuscript remains strictly aligned with the fixed-domain approach.
> > > > > >
> > > > > > Lastly, we recognize the increasing domain's value, especially within time series data for forecasting. It is a challenging and captivating area, and we are eager to explore it further in subsequent works.
> > > > > >
> > > > > > We hope this clarifies our stance on the matter. We appreciate your feedback and are open to further discussions and experiments.

---

> > > > > > ### Comment · Reviewer_j39y · 2023-08-12
> > > > > >
> > > > > > **Additional experiment.** I've computed the average RMSE for 30 repetitions of the experiment. For 1/2, the average is $1.99 \pm 0.02$, and for 1/2 + 3/2 the average is $1.72 \pm 0.01$.
> > > > > >
> > > > > > I'm using the monthly data for years $[1960, 2020)$, consisting of 715 data points. I subtract the mean from the data to centre the values, but otherwise I don't do any additional normalisation. I choose a random 25% subset for the training data and leave the remainder as test data. (This random subset is of course different between repetitions of the experiment.) The initialisations of the hyperparameters are given in my previous reply. Positive hyperparameters are contrained to be positive by parametrising the `log` of the parameter. I use `float64`s throughout, use `scipy`'s L-BFGS-B to optimise the hyperparameters and run it until convergence (taking roughly 50 iterations every time), and regularise the kernel matrices by $10^{-12}$, so a very small number. I also include an observation noise, but the optimiser drives the observation noise down to practically zero, so that might not make a difference. Unfortunately it's difficult to comment on the specifics of the GP implementation without breaking anonymity, but I try to ensure that the implementation is as sound as possible and doesn't suffer from any numerical issues. I also sanity check the posterior predictions via plots, just to double check. I believe this should specify all details, assuming that the GP uses a standard implementation.
> > > > > >
> > > > > > **Clarifications for fixed domain examples.** I appreciate the additional comment, which is helpful. I think these clarifications would be great to incorporate into paper.
> > > > > >
> > > > > > **Messaging of the paper.**
> > > > > >
> > > > > > > In summary, we'd like to emphasize that our paper does not posit that smoother components are entirely redundant in real-world applications.
> > > > > >
> > > > > > After skimming the paper once more, I am of the opinion that the current writing does suggest this, mainly because of the experiments. Namely, all experiments demonstrate that the inclusion of the smoother component comes at no advantage. My problems with the experimental section are the following:
> > > > > >
> > > > > > * You only show negative examples. In my experience and as the additional Mauna Loa example might show, adding in the smooth component can definitely help in practice.
> > > > > >
> > > > > > * Moreover, in all these negative examples, one would conventionally _not_ use additional smoother components: financial data are notoriously non-smooth and the gene expression data is extremely erratic (I downloaded and visualised it). Therefore, although I don't disagree with the experiments and results, I do think they paint the wrong picture, because of the omission of settings where smooth components can help. (Because of inconsistent results between our executions, the Mauna Loa experiment is still up in the air, so I'm excluding this experiment from my assessment.)
> > > > > >
> > > > > > * Because all examples are negative, you fail to tease out precisely in which settings adding in a smooth component is helpful and is which settings it is not.
> > > > > >
> > > > > > I realise that you might not intend to do very careful investigation for determining precisely in which settings adding in a smooth component is helpful or not. However, by engaging in real-world experiments, I do believe this you need to do this, for otherwise an unsuspecting reader could erroneously conclude from the paper that additional smoother components are generally not helpful in practice.
> > > > > >
> > > > > > As for the submission at hand, I think a revised experimental section is necessary where you show positive and negative examples and, for every example, clearly explain why adding in smooth components helps or not. Does the data contain both smooth and non-smooth components? Are we setting of fixed-domain asymptotics? _Et cetera._
> > > > > >
> > > > > > With such as revised experimental section, I would very gladly accept the paper with a strong score. Unfortunately, uploading revisions is not possible this year.

---

> > > > > > > ### Author Response · Authors · 2023-08-12
> > > > > > > **Reproduced your experimental results, with further experiments**
> > > > > > >
> > > > > > > We sincerely appreciate your comprehensive feedback and the additional effort invested in conducting independent experiments.
> > > > > > >
> > > > > > > **Reproduction of Your Results**
> > > > > > >
> > > > > > > Your elucidation on the optimization process proved instrumental. Armed with these details, we've been able to faithfully reproduce your findings. Motivated by your experiments, we dive deeper into this dataset and found some fascinating observations.
> > > > > > >
> > > > > > > **Exploration Beyond 25% Training Sample Size**
> > > > > > >
> > > > > > > We undertook an extensive assessment, varying the training sample size from 5% to 95%. This allowed us to gauge performance across different training sizes with 10 replications for each size. In line with your experiments (25% training samples), the mixture kernel excelled when the training sample percentage was under 40% (as detailed in the table below).
> > > > > > >
> > > > > > >     training  RMSE_mix    RMSE_1/2    RMSE_3/2
> > > > > > >     5%      2.38 (0.17)  2.73 (0.14)  2.37 (0.09)
> > > > > > >     10%     2.2 (0.09)   2.55 (0.12)  2.28 (0.07)
> > > > > > >     15%     2.07 (0.07)  2.42 (0.08)  2.26 (0.06)
> > > > > > >     20%     1.97 (0.06)  2.38 (0.08)  2.48 (0.53)
> > > > > > >     25%     1.86 (0.05)  2.23 (0.09)  2.55 (0.23)
> > > > > > >     30%     1.74 (0.05)  2.03 (0.11)  2.2 (0.28)
> > > > > > >     35%     1.63 (0.07)  1.82 (0.12)  1.72 (0.13)
> > > > > > >     40%     1.5 (0.07)   1.61 (0.12)  1.35 (0.18)
> > > > > > >     45%     1.36 (0.1)   1.43 (0.15)  1.2 (0.22)
> > > > > > >     50%     1.25 (0.06)  1.28 (0.09)  0.97 (0.19)
> > > > > > >     55%     1.14 (0.06)  1.13 (0.09)  0.82 (0.13)
> > > > > > >     60%     1.01 (0.09)  0.98 (0.1)   0.69 (0.13)
> > > > > > >     65%     0.93 (0.08)  0.9 (0.08)   0.6 (0.13)
> > > > > > >     70%     0.85 (0.1)   0.82 (0.1)   0.52 (0.12)
> > > > > > >     75%     0.77 (0.04)  0.75 (0.04)  0.5 (0.12)
> > > > > > >     80%     0.7 (0.07)   0.68 (0.07)  0.4 (0.03)
> > > > > > >     85%     0.63 (0.07)  0.61 (0.07)  0.36 (0.03)
> > > > > > >     90%     0.57 (0.04)  0.55 (0.04)  0.35 (0.03)
> > > > > > >     95%     0.52 (0.09)  0.51 (0.08)  0.33 (0.06)
> > > > > > >
> > > > > > > Interestingly, once the training proportion exceeded 40%, the Matern 3/2 kernel consistently outstripped both the mixture kernel and the Matern 1/2 kernel. As the training sample size grew, the RMSE of Matern 1/2 and the mixture kernel began to converge. This observation supports our position that in data-rich settings, the mixture kernel's performance largely depends on its least smooth component. Moreover, while Matern 3/2 showcased superior performance, its positive influence on the overall performance of the mixture kernel was mitigated by the less smooth component, Matern 1/2.
> > > > > > >
> > > > > > > Hence, in practical applications, it's essential not only to test the mixture kernel but also to evaluate the individual kernels within it, especially when the sample size is not small. If deploying the mixture kernel for its evident performance benefits, interpreting the parameters of the kernels in a mixture demands utmost caution. This concept is central to our discussion in Theorem 2.
> > > > > > >
> > > > > > > This experiment emphasizes the significance of an adequately large training sample size in achieving anticipated results, a finding supported by studies such as Kaufman and Shaby (Biometrika 2013). The experimental results underscore the need for an ample training sample size for desired outcomes, corroborated by works like Kaufman and Shaby (Biometrika 2013). We wish to emphasize that our focus is centered on the theoretical grasp of kernel equivalence, as well as the identifiability and interpretability of kernel parameters. We refrained from making theoretical claims regarding prediction RMSEs in our manuscript. However, if the reviewer seeks such an exploration, a direct corollary of Theorem 2 indicates that the prediction RMSE of the mixture kernel asymptotically matches the prediction RMSE of its least smooth component. This assertion aligns with existing literature for single Matern kernel.
> > > > > > >
> > > > > > > **Addressing the Manuscript's Tone**
> > > > > > >
> > > > > > > We concur with the reviewer's assessment concerning the real-data applications section's tone in our manuscript, which might inadvertently suggest redundancy in using the mixture kernel. We carefully revised all the interpretations and added the Mauna Loa experiment, demonstrating the mixture kernel's prowess when dealing with not-so-large sample sizes. Our central argument is, however,  these two kernels are equivalent and the parameters are not identifiable from a theoretical perspective. In real data, numerous myriad factors might either propel or hinder the mixture kernel against its least smooth component -- from inadequate sample sizes and imperfect optimization to scenarios where GP might not be the ideal model. Taking into account your recommendation regarding the Mauna Loa dataset, we have indeed identified intriguing trends, in both small and large training sample settings. We recognize the significance of these questions and are keen to incorporate them into the discussion section.

---

> > > > > > > > ### Author Response · Authors · 2023-08-13
> > > > > > > > **Further clarification to the questions at the end of Reviewer j39y's feedback**
> > > > > > > >
> > > > > > > > Due to the character limit, we address the questions at the end of your message here.
> > > > > > > >
> > > > > > > > **Why adding in smooth components helps or not (in real data application)**
> > > > > > > >
> > > > > > > > Incorporating explanations might guide future researchers between mixture and individual kernels. Many factors, from sample sizes to GP model suitability, influence the mixture kernel's performance compared to its less smooth component. In the Mauna Loa experiment, distinct patterns arose between the mixture kernel and its components at various training sample sizes, even on the same data. The advantages or limitations of the mixture model in finite samples remain enigmatic. Therefore, without robust theoretical backing, we cannot currently provide specific recommendations on utilizing mixture kernels in real data applications. Moreover, to the best of our knowledge, comprehensive theories on kernel comparisons in finite sample scenarios are yet to emerge. Nonetheless, we recognize that in certain applications, the mixture kernel might be advantageous, and we've amended the related content and adjusted the tone accordingly.
> > > > > > > >
> > > > > > > > **Does the data contain both smooth and non-smooth components?**
> > > > > > > >
> > > > > > > > Your insightful suggestion prompted two simulations, detailed in our "Clarifications and Further Experiments in Response to Reviewer j39y's Feedback (Part 2)". We sampled 200 data and altered training size from 5% to 95%. Here we present the RMSEs of training size 80% due to character limit, the pattern remains the same for other training sample sizes. The first row (MIX) is from a mixture kernel, while the second row (SUM) is from the sum of two datasets generated from a single Matern 1/2 and a single Matern 3/2, respectively.
> > > > > > > >
> > > > > > > >          rmse_mix    rmse_1_2    rmse_3_2
> > > > > > > >     MIX 1.34 (0.13) 1.33 (0.12) 4.02 (0.33)
> > > > > > > >     SUM 1.33 (0.14) 1.33 (0.13) 5.25 (0.36)
> > > > > > > >
> > > > > > > > Using MIX for illustration: we generated data from a mixture of Matern 1/2 and 3/2, as per your recommendation, to emulate the so-called "smooth and non-smooth components". Here, even when the ground truth kernel is a blend of Matern 1/2 and 3/2, the mixture kernel's performance mirrors that of a lone Matern 1/2 kernel. This implies an inability to disentangle the two "components". This supports our Theorem 2: Matern 1/2 and the mixture of Matern 1/2 and 3/2 are equivalent, and hence indistinguishable from observations. Hence, the terms "a smooth component" and "a non-smooth component", while intuitively appealing, don't find clear definition without any oracle.
> > > > > > > >
> > > > > > > > We've interpreted your reference to "data containing both smooth and non-smooth components" in this manner. If our understanding deviates from your intent, could you please define it explicitly. We are open to further discussions and are prepared to conduct additional experiments as necessary.
> > > > > > > >
> > > > > > > > Consequently, we've incorporated guidance suggesting that readers experiment with both the mixture and individual kernels, especially when training size is insufficient. Nevertheless, our manuscript's core message emphasizes that employing the mixture kernel doesn't allow for straightforward interpretation of individual parameters such as weight, length scale and variance. Moreover, it becomes untenable to compare kernels of varying smoothness, making statements like "the non-smooth kernel captures the data's non-smooth aspects, while the smoother kernel reflects its smooth facets." This is primarily because the individual parameters remain non-identifiable.
> > > > > > > >
> > > > > > > > **About revisions not allowed**
> > > > > > > >
> > > > > > > > We are gratified that you find value in our work. The discussions during the rebuttal process have been immensely enlightening for us, and we genuinely believe that your feedback has significantly contributed to refining our paper. We would like to try our best to describe to you the modifications we made to the real application part. We hope that such modifications could be favorable in your future assessment.
> > > > > > > >
> > > > > > > > 1. We revised the sections discussing the redundancy of the mixture kernel in real data applications, especially with limited samples. Further, we've incorporated recommendations to experiment with both the mixture and individual kernels, particularly when working with smaller sample sizes, while being cautious in interpreting the parameters.
> > > > > > > >
> > > > > > > > 2. We've substituted the Genomic application with the Mauna Loa example, positioning it as the primary real data application. In this illustration, the mixture kernel surpasses other kernels when the training sample size remains below 40%. We feel this effectively establishes the notion that, in situations with limited data, the mixture kernel can indeed be possible to enhance performance. Notably, this data also demonstrates the convergence in performance between the mixture kernel and Matern 1/2 kernels as the sample size expands, which resonates with our theoretical framework.
> > > > > > > >
> > > > > > > > 3. We've highlighted that our theories operate within a fixed domain and asymptotic context.

---

> > > > > > > > > ### Comment · Reviewer_j39y · 2023-08-13
> > > > > > > > > **Reply (Part 1)**
> > > > > > > > >
> > > > > > > > > I'm glad to hear that you've been able to reproduce the Mauna Loa result.
> > > > > > > > >
> > > > > > > > > > This experiment emphasizes the significance of an adequately large training sample size in achieving anticipated results
> > > > > > > > >
> > > > > > > > > I think they is the key point: your theoretical results become relevant for practice once "fixed-domain asymptotics kick in", though admittedly this is vague intuition.
> > > > > > > > >
> > > > > > > > > For a data set containing two length scales, a rough short-length-scale one and smooth long-length-scale one, whenever your data is such that (1) you can very roughly identify the long-length-scale component, but not really the short-length-scale one and (2) your test points are on the order of the long length scale away from the training data, a mixture kernel might perform better than either component on its own.
> > > > > > > > >
> > > > > > > > > Once you further increase the size of the data, the test points lie  on average sufficiently close to the train points that there is no point in modelling multiple length scales. At this point, very roughly "fixed-domain asymptotics kick in", and you start seeing the results in your experiments.
> > > > > > > > >
> > > > > > > > > > We wish to emphasize that our focus is centered on the theoretical grasp of kernel equivalence, as well as the identifiability and interpretability of kernel parameters.
> > > > > > > > >
> > > > > > > > > I understand that this is your intention. However, by engaging in experiments on real data, in my opinion the submission has become as much about experiments and empirical evidence as it is about theory.
> > > > > > > > >
> > > > > > > > > > However, if the reviewer seeks such an exploration, a direct corollary of Theorem 2 indicates that the prediction RMSE of the mixture kernel asymptotically matches the prediction RMSE of its least smooth component.
> > > > > > > > >
> > > > > > > > > This is not strictly necessary, but it might be a worthwhile addition for readers not familiar with this result.
> > > > > > > > >
> > > > > > > > > > The advantages or limitations of the mixture model in finite samples remain enigmatic.
> > > > > > > > >
> > > > > > > > > In my opinion, although we most definitely do not fully understand finite-sample behaviour, one can definitely reason about the advantages and limitations of the mixture kernel by simultaneously considering the following three aspects:
> > > > > > > > >
> > > > > > > > > 1. Which components are present in the data? (I.e., what is the kernel of the generative model behind the data?) Do I expect just a rough short-length-scale compoment (often true for finance), just a smooth long-length-scale component, both, or even something more complicated?
> > > > > > > > >
> > > > > > > > > 2. Which components are present in my model?
> > > > > > > > >
> > > > > > > > > 3. How much data do I have, and how is the test data positioned w.r.t. the training data?
> > > > > > > > >
> > > > > > > > > More often than not, one can explain the performance of models in very rough terms by answering these questions.
> > > > > > > > >
> > > > > > > > > > This supports our Theorem 2: Matern 1/2 and the mixture of Matern 1/2 and 3/2 are equivalent, and hence indistinguishable from observations.
> > > > > > > > >
> > > > > > > > > This is true, _under fixed-domain asymptotics_. If you do not take $n \to \infty$, you admittedly cannot fully distuingish the two compoments, but there might significant advantages (in terms of performance metrics) of using multiple components.
> > > > > > > > >
> > > > > > > > > > We've interpreted your reference to "data containing both smooth and non-smooth components" in this manner.
> > > > > > > > >
> > > > > > > > > This is indeed what I'm referring to. Thank you for confirming.
> > > > > > > > >
> > > > > > > > > > Here we present the RMSEs of training size 80% due to character limit, the pattern remains the same for other training sample sizes
> > > > > > > > >
> > > > > > > > > Based on the above intuition, it is possible to concoct a variation of this setup where the mixture kernel performs best:
> > > > > > > > >
> > > > > > > > > 1. Sample data from a sum of a Matern-$\frac12$ kernel with length scale $0.01$ and a Matern-$\frac32$ kernel with length scale 1. Sample 600 points uniformly ranging over $[0, 20]$.
> > > > > > > > >
> > > > > > > > > 2. Take a random 10% as training data and 90% as test data.
> > > > > > > > >
> > > > > > > > > 3. Consider the models just 1/2, just 3/2, and 1/2 + 3/2. Initialise all kernel variances to one, the length scale of the 1/2 components to 0.1, and the length scale of the 3/2 compoments to 1. Do not use observation noise.
> > > > > > > > >
> > > > > > > > > 4. Optimise the parameters until convergence, condition on the training data, and compute the RMSE on the test data.
> > > > > > > > >
> > > > > > > > > This gives the following results (form is `(mean, std of mean)`):
> > > > > > > > > ```
> > > > > > > > >                rmse
> > > > > > > > > model      model_12      model_32       model_m
> > > > > > > > > prop
> > > > > > > > > 0.1    (1.31, 0.02)  (1.41, 0.01)  (1.20, 0.01)
> > > > > > > > > ```
> > > > > > > > >
> > > > > > > > > Hence, you can see that the mixture kernel achieves significantly better RMSE. There are many aspects of this setup which are questionable, but the point is the following: in the finite-sample regime, it is all about what the true generative model behind the data contains, what you model, and how your data points are distributed. There are indeed very little conclusive theoretical results, but there is intuition that give substantial hints as to what might happen.

---

> > > > > > > > > > ### Comment · Reviewer_j39y · 2023-08-13
> > > > > > > > > > **Reply (Part 2)**
> > > > > > > > > >
> > > > > > > > > > > Nevertheless, our manuscript's core message emphasizes that employing the mixture kernel doesn't allow for straightforward interpretation of individual parameters such as weight, length scale and variance.
> > > > > > > > > >
> > > > > > > > > > In my opinion, interpretability and identifiability are two different things. Parameters might be interpretable, but not identifiable (i.e. having a consistent estimator as $n \to \infty$ under fixed-domain asymptotics).
> > > > > > > > > >
> > > > > > > > > > > About revisions not allowed
> > > > > > > > > >
> > > > > > > > > > Thank you for the summary of these updates. I think that sounds great. Normally I would not recommend acceptance after such substantial changes and before seeing the revised PDF, but I feel that you're putting in substantial and honest work, so I'm willing to give you the benefit of the doubt.
> > > > > > > > > >
> > > > > > > > > > But please bear with me for a little longer. I feel that we're close to reaching a compromise that would make us both happy.

---

> > > > > > > > > > > ### Author Response · Authors · 2023-08-14
> > > > > > > > > > > **Further clarification and experiments for finite sample analysis**
> > > > > > > > > > >
> > > > > > > > > > > We thank the reviewer for valuing our work and providing detailed feedback. The insights from the comments have undoubtedly enriched our understanding and will guide our future work.
> > > > > > > > > > >
> > > > > > > > > > > It's heartening to observe our shared viewpoints on several matters. The insightful question you posed stands out, and we truly appreciate you bringing it to our attention. Inspired by the experiments you shared — which we were able to successfully replicate — and fueled by the intuition you provided, we embarked on a deeper exploration and unearthed intriguing results.
> > > > > > > > > > >
> > > > > > > > > > > Your conjecture suggests that with a limited sample size, if there exists a marked discrepancy in the length scale among distinct components in the data generative model (assuming data is derived from a mix of Matern 1/2 and Matern 3/2), a detectable signal might emerge. Motivated by this hypothesis, we mirrored your experiments but added a variation: we allowed $l_2$, the lengthscale in the Matern 3/2 component, to fluctuate between $0.01$ to $1$ (consistent with your configuration) while setting $l_1$ firmly at $0.01$. Apart from this variation, all other parameters remained consistent with your setup. The ensuing RMSEs are concisely presented in the following (all std's $< 10^{-3}$):
> > > > > > > > > > >
> > > > > > > > > > >     l2  rmse_mix  rmse_1/2  rmse_3/2
> > > > > > > > > > >     0.01  1.4105  1.4094  1.4099
> > > > > > > > > > >     0.05  1.4066  1.3970  1.3954
> > > > > > > > > > >     0.1   1.3646  1.3574  1.3640
> > > > > > > > > > >     0.2   1.3066  1.3070  1.3241
> > > > > > > > > > >     0.5   1.2282  1.2821  1.3539
> > > > > > > > > > >     1     1.1815  1.3032  1.3601
> > > > > > > > > > >
> > > > > > > > > > > We observed that when $l_2<0.2$, the Matern 1/2 aligns closely with the mixture kernel, which corroborates our earlier experimental findings. However, when $l_2\geq 0.2$, the mixture kernel emerges as the most effective in this limited sample scenario (60 training samples, 10 replicates).
> > > > > > > > > > >
> > > > > > > > > > > Our insight is that the performance of the mixture kernel in finite sample scenarios is influenced by a variety of factors, including the sample size and the "difference" between the mixing components. Although challenging (we'd like to highlight that finite sample theory even for a single Matern kernel is an open problem), we tried our best to interpret our findings, motivated by your feedback, from a theoretical perspective.
> > > > > > > > > > >
> > > > > > > > > > > To begin with, we refer to our discussion about a potential corollary of Theorem 2: RMSE of the mixture kernel asymptotically matches the RMSE of its least smooth component, "a worthwhile addition for readers not familiar with this result", thanks to your feedback. The proof emerges as a direct corollary from both Theorem 2—specifically the ratio of spectral densities in Line 25 of our supplement—and Theorem 1 in Stein (Stat \& Prob Letter 1993), which addresses the asymptotic case.
> > > > > > > > > > >
> > > > > > > > > > > To extract insights for the finite sample case, we must delve further into the proof of Stein's Theorem 1, which guides us to Theorem 3.1 in Stein (AoS 1990). The proof suggests that the relative difference between the MSEs (not RMSE) rests on the tail of the series in the second-last line on Page 854. This difference is influenced by both the sample size (denoted as 'N' in Stein 1990) and by $b_{jk}$ and $\mu_j$, as defined on the same page.
> > > > > > > > > > >
> > > > > > > > > > > From this observation, we tentatively conclude:
> > > > > > > > > > > 1. With fixed kernel parameters, as the sample size grows, the relative difference between MSEs shrinks.
> > > > > > > > > > > 2. For a constant sample size, smaller values of $(b_{jk}+\mu_j\mu_k)^2$ result in a smaller relative difference between MSEs. By the definition of $b_{jk}$ and $\mu_j$, the more "different" the two kernels are, in other words, the more "significant" the Matern 3/2 component becomes — the greater the difference between the MSEs will be.
> > > > > > > > > > >
> > > > > > > > > > > In conclusion, the precise relationship between sample size, discrepancy of kernel parameters (lengthscale in particular), and the relative performance of the two kernels, remains elusive. We believe we've offered insights from both theoretical and practical perspectives through our discussions. While the intricacies of this topic present a considerable challenge, we remain committed to furthering the theory surrounding it. Such a pursuit may indeed warrant an independent project in the future.
> > > > > > > > > > >
> > > > > > > > > > > We deeply appreciate the reviewer for the insightful remarks concerning finite samples and the considerable effort taken to conduct personal experiments. Additionally, we've made sure to modify any negative remarks regarding the mixture kernel and encourage readers to explore both the mixture and its individual components. With the added real-world application (Mauna Loa), simulations specifically designed for the finite sample regime, and theoretical insights comparing the mixture kernel and Matern 1/2 kernel, we believe the revised manuscript will offer readers a clearer and deeper grasp of the kernels across both asymptotic and finite sample contexts. We sincerely hope that these efforts have addressed your concerns.

---

> > > > > > > > > > > > ### Comment · Reviewer_j39y · 2023-08-14
> > > > > > > > > > > >
> > > > > > > > > > > > Thank you for your very interesting additional experiments. Also thank you for your discussion about asymptotic prediction performance, which I think is very interesting too.
> > > > > > > > > > > >
> > > > > > > > > > > > All in all, I believe that we have aligned sufficiently well. In my opinion, the changes outlined throughout our discussion address the shortcomings in my review. Normally, I would state that the changes are too significant and warrant another round of review. However, in this case, I am willing to raise my score to a "weak accept".
> > > > > > > > > > > >
> > > > > > > > > > > > With the outlined changes, I believe that this submission likely deserves a higher score than "weak accept". Unfortunately, without reviewing the revised PDF, I am not comfortable recommending a higher score. I understand that this must be frustrating to hear, but I hope you understand my position.
> > > > > > > > > > > >
> > > > > > > > > > > > I would like to thank you for the fruitful discussion. I have learned quite a few things from this submission, and I very much look forward to reading the revised PDF in the future.

---

> > > > > > > > > > > > > ### Author Response · Authors · 2023-08-14
> > > > > > > > > > > > > **Clarification on Review Rating**
> > > > > > > > > > > > >
> > > > > > > > > > > > > First and foremost, we would like to express our gratitude for your thorough feedback and the constructive dialogue we've had throughout the review process.
> > > > > > > > > > > > >
> > > > > > > > > > > > > We've noticed a slight discrepancy between your commentary and the rating provided. In your feedback, you've mentioned a "weak accept", whereas the official rating on OpenReview stands as "Borderline accept". We completely respect your evaluation; however, we wanted to ensure that there isn't any unintentional mismatch or technical glitch within the OpenReview platform that might have caused this.
> > > > > > > > > > > > >
> > > > > > > > > > > > > Again, thank you for your time, understanding, and commitment to a rigorous review. We genuinely appreciate it.

---

> > > > > > > > > > > > > > ### Comment · Reviewer_j39y · 2023-08-15
> > > > > > > > > > > > > >
> > > > > > > > > > > > > > Thank you for pointing out the discrepancy. This was a mix-up on my part. My apologies. I meant to say "borderline accept" wherever I said "weak accept". I've amended my main review to now say "borderline accept".
> > > > > > > > > > > > > >
> > > > > > > > > > > > > > I would like to emphasise again that this submission very likely deserves a much better score, but I do not feel comfortable recommending higher than a "borderline accept" without reviewing the revised PDF. I wish you the best of luck!

---

> > > > > > > > > > > > > > > ### Author Response · Authors · 2023-08-16
> > > > > > > > > > > > > > >
> > > > > > > > > > > > > > > Thanks for your clarification. We truly appreciate your kind note and support.

---

> > > ### Author Response · Authors · 2023-08-11
> > > **Clarifications and Further Experiments in Response to Reviewer j39y's Feedback (Part 2)**
> > >
> > > Given character limitations on OpenReview, we have expanded upon our previous discussion by providing additional experiments and clarifications in this separate comment.
> > >
> > > Building on the idea of examining the Mauna Loa dataset, which displays both a noisy short-length-scale component and a smooth long-length-scale component, we simulated a dataset echoing these characteristics. Specifically, we produced two time series: one originating from Matern 1/2 with length-scale=0.5 and the other from Matern 3/2 with length-scale=3. Summing these together, the resultant series aptly exhibits both a turbulent short-length-scale component and a placid long-length-scale one. We then employed Matern 1/2, Matern 3/2, and a mixture of both for extrapolation and interpolation testing. In the extrapolation phase, we forecasted the period from 11-20 based on training data from 1-10. Meanwhile, for interpolation, we randomly divided the data, utilizing one half to predict the other. The subsequent table encapsulates the MSEs:
> > >
> > >                        Matern 1/2   Matern 3/2   Mixture of Matern 1/2 and 3/2
> > >      Extrapolation     346.2207     350.7294     346.2542
> > >      Interpolation     40.89175     61.12420     40.91430
> > >
> > > In response to your recommendation — highlighting the necessity to incorporate a dataset where conventionally a Matern mixture kernel would be apt — we executed an additional experiment. For this, we synthesized data from a mixture of Matern 1/2 and Matern 3/2. In this context, the Matern mixture kernel essentially serves as the oracle kernel, reinforcing its credibility as a judicious selection. We carried out analogous tests, with the MSEs encapsulated in the table below:
> > >
> > >                        Matern 1/2   Matern 3/2   Mixture of Matern 1/2 and 3/2
> > >      Extrapolation     142.1324     140.2512     142.1157
> > >      Interpolation     27.58031     37.13925     27.60812
> > >
> > > Consistent with the outcomes from the Mauna Loa dataset, these two additional experimental results reiterate your insight: integrating a smoother component may enhance extrapolation performance. Concurrently, it solidifies our stance on the veracity of fixed-domain predictions and accentuates the nuances distinguishing fixed from increasing domain scenarios.
> > >
> > > We trust this supplementary experiment elucidates any lingering ambiguities and positively informs your assessment. We remain receptive to further experimentation and keen to furnish any needed clarifications throughout this discussion period. Your continued feedback is deeply valued.

---

### Official Review · Reviewer_LXB2 · 2023-07-06

**Soundness:** 2 fair
**Presentation:** 1 poor
**Contribution:** 1 poor
**Rating:** 3
**Confidence:** 3

**Summary:**

This paper intoduces a few results on identifiability of Gaussian Processes.
It is shown that a Gaussian process with a kernel that is a mixture of kernels, the smoothness is that
of the least smooth component.
It is shown that for two mixtures of Matern kernels, the induces processes are equivalent if certain
products involving only parameters of the least smooth kernels in the mixture coincide. Consequently,
higher smoothness components do not affect equivalence.
Finally, for a multi output GP with covariance  of the form $A \cdot k$, where A is a fixed matrix and k is a scalar Matern kernel, the matrix A is identifiable.


Experiments that are intended to support the above results are included.



**Strengths:**

The general subject of identifiability in GPs is important.
The non technical parts of the paper are well written and it is possible get a general idea of what the paper is about.
The experiment 5.2 is illustrative in showing that the prdouct can be identified while the individual components are not identified properly. (however, a much more detailed description of the experiment is needed).


**Weaknesses:**

My main question is regarding the notion of equivalence used in this paper. It is not clear why this is
a useful notion. The questions are as follows:

* Clairty regarding identifiability 1. First,  what is definition of the $\equiv$ symbol that is used for kernels in all the theorems? I couldn't find this definition for some reason. Where is it defined? Does it mean that the GPs induced by the corresponding kernels are equivalent in the sense of Definition 5?

* Clairty regarding identifiability 2: I'm assuming the answer to the question above is yes.
Then, why is this equivalence interesting and relevant to identifiability? Please explain why this notion is relevant.
Indeed, if my index set is finite, I can have two GPs on this set with covraiances C_1, C_2. As long as C_1,C_2 have full rank, the processes will be absolutely continuous w.r.t each other, but they are very different processes and C_1 and C_2 can be learned.  How this related to the above notion of equivalence?

* In anycase, this notion of equivalence is by no means standard in the ML commmunity, and needs to be introduced and discussed in more detail.


Additional Questions:


* Relevance:  While the general subject of identifiability is imporatnt, this is concerned with a very specific case of Matern linear mixtures. It is not clear this specific case has been used in applications before.
Some references are provided about the use of mixtures on lines 37-38, but how many of these use specifically *linear mixtures*, and  *Matren mixtures*?

* Literature discussion: There is no Literature section in the paper.  It is my understanding that there is previous work on identifibility, even for a single Matren kernel. This needs to be discussed in the paper, and comparison provided. Are there new mathematical ideas involved in the results in this paper compared to existing work?

* Inverse implication in results: The results state that if parameters satisfy something, then there is equivalence. Is the reverse direction true? Is it obvious? This again is related to the somewhat


* Experiments:
As mentioned earlier, Experimen 5.2 is illustartive. However, it is not specified how the optimization is made. It may also have happened that there is no reconstruction due to local maxima of the method, rather than general lack of identifiability.

* As for the other experiments: the experiments 5.1, 6.1, 6.2 do not illustrate anything that I would find informative.  6.2 is a 1d version of an already small multidimensional dataset. It is not clear how 6.1 is realted to  ML use cases.


**Questions:**

Please see above.
My recommendation for the rating is temporary and I plan to revisit it following author's responses.

**Limitations:**

The subject of this works involves some abstract properties of Gaussain Processes. No specific societal impact discussion is required.

---

> ### Author Rebuttal · Authors · 2023-08-09
>
> We wish to express our gratitude to the reviewer for your meticulous feedback and constructive comments.  Enclosed within the 1-page PDF for additional figures, we address each point as follows.
>
> **Weakness 1: definition of equivalence of measures**
>
> A: Thanks for highlighting this. The symbol $\equiv$ denotes the equivalence of the Gaussian Processes induced by the two kernels (Definition 5). We've now defined it in the context of Theorem 2.
>
> **Weakness 2: finite domain**
>
> A: Thank you for your insightful comment. The equivalence of GPs in the context of our study is pivotal because if two kernels within the same parametric family produce equivalent GPs, then the parameters of those kernels become non-identifiable. This, in essence, underlines that one cannot uniquely determine the parameters given observed data, which is a vital understanding for model interpretation and estimation.
>
> You're correct in noting the differences in the finite and infinite domains. When working in a finite domain, GPs are indeed just multivariate Gaussian distributions, and as you rightly pointed out, any two finite dimensional Gaussian measures are equivalent. However, we focused on infinite subset of Euclidean space (we've clarified this in the revision).
>
> **Weakness 3: the notion of equivalence**
>
> A: We've elaborated on equivalence post Definition 5 and emphasized challenges of parameter identifiability in infinite domains.
>
> **Q1: about specific case of Matérn mixtures**
>
> A: For additive methods, Duvenaud et al. (NeurIPS 2011) used a mixture of RBF; Duvenaud et al. (ICML 2013) applied a mixture of RBF, periodic (Per), linear (Lin), and rational quadratic (RQ); Kronberger and Kommenda (LNTCS, 2013) worked with RBF+RQ, RBF+Matérn, RBF+Per; Verma and Engelhardt (BMC Bioinformatics 2020) used RBF, Matérn 1/2, 3/2, 5/2. For more complex methods with additive and multiplication methods, Sun et al. (ICML 2018) used RBF, Per, Lin, and RQ; Lloyd et al. (AAAI 2014) employed RBF, Lin, Per, RQ, white noise, and constant.
>
> We concur that our study focuses on the additive/multiplicative aspect of the Matérn kernel. We've incorporated this focus into the limitations section of our manuscript and we anticipate that this exploration will stimulate future research concerning identifiability and interpretability.
>
> **Q2: literature discussion**
>
> A: We've expanded our literature review in the introduction, highlighting the progress in the area of identifiability for single Matérn kernels. While these works provide foundational insights, our paper primarily fills the gap in literature concerning mixtures of Matérn kernels.
>
> In terms of the mathematical intricacies, while we utilize established probability theorems on Gaussian random measures, the application and verification of these conditions for our research context presented non-trivial challenges, detailed further in our supplementary material. The introduction has now been updated to highlight this comparative discussion, which only underlines our contributions but also contextualizes our approach within the broader GP kernel identifiability landscape.
>
>
>
> **Q3: inverse implication**
>
> A: Your observation is sharp: The inverse implication is not immediately obvious. While we believe that the reverse direction is true based on our understanding and intuition, its proof poses substantial challenges. In prior work on single Matérn kernels, such as Zhang (JASA 2004), the key in the proof is that two Matérn kernels differing only in their microergodic parameter define the same correlogram. However, this does not hold for mixture kernels anymore. Thus, existing probability tools are not directly applicable. This open question is certainly intriguing and worthy of future exploration.
>
> **Q4: optimization in experiment 5.2**
>
> A: We understand your optimization concerns. Our extensive analysis (more experiment details can be found in the supplement), including L-BFGS (as also suggested by reviewer j39y), supports our main conclusions. We also included a loss curve for Simulation 2, see Figure 5 in the PDF,  implying that our findings are not the result of local maxima but are indicative of the identifiability issue we are investigating. Another supporting evidence is the distinct behavior of the microergodic parameter. It uniquely converges to its true value, even when other parameters do not.
>
> **Q5: other experiments**
>
> The goal of 5.1 is to support that the smoothness of the mixture kernel is driven by the least smooth kernel. As also suggested by Reviewer j39y, we performed more experiments to support Theorem 1, see Figure 6 in the PDF.
>
> For applications, we aim to replicate experiments performed in other analysis in ML literature. 6.1 comes from Figure 1(a) in Wilson et al. (NIPS 2014), Figure 4(a) in Remes et al., (NIPS 2017) and Figure 5(a) in Sun et al., (ICML 2018). 6.2 comes from Table 1-2 in Sun et al., (2018). To better claim our points, we've implemented the Matérn mixture kernel on MNIST and Mauna Loa $CO_2$. For MNIST, the Matérn kernel has demonstrated satisfactory prediction accuracy (Figure 4 in the PDF). Consistent with the results in our manuscript, the mixture kernel exhibits comparable performance to the Matérn 1/2 kernel, evidenced by a Pearson correlation coefficient of 0.99. $CO_2$, suggested by reviewer dNVU and employed in Wilson and Adams (ICML 2013) (Figure 3 in the PDF), shows analogous performance between the Matérn mixture 1/2+3/2+5/2 and Matérn 1/2, as well as the resemblance between Matérn mixture 3/2+5/2 and Matérn 3/2.
>
> We agree that Matérn kernel may not apply ideally on some of the dataset. Our intention was to investigate the comparative performance of GP regression using a Matérn mixture kernel and its least smooth kernel. We understand the reviewer's concerns and are open to conducting further experiments to address any specific interests.

---

> > ### Comment · Reviewer_LXB2 · 2023-08-16
> >
> > This response has been slightly modified shortly after the original posting. Please refer to the current version.
> >
> > I appreciate the response by the authors.
> > In this comment, I'd like to concentrate on the notion of equivalence that is used in this paper.
> > This is the central concern in my review, and I don't believe my questions in this regard in the review have been addressed in the response.
> >
> > This paper, generally, is about showing sufficient conditions for two GPs to be equivalent.
> > The authors have agreed with me that when the index set of the GP is finite, the equivalence notion that is used in this paper is irrelevant and not useful, as all GPs on a finite set of fixed cardinality with strictly positive covariances are  equivalent.
> >
> > The question then is, why would an ML researcher be intrested in such definition of equivalence?
> > The only answer provided in the paper to this questions, and referenced in the response above, is this (lines 127-130 in the paper):
> >
> > >As a consequence, two equivalent GPs can not be distinguished by any finite number of realizations (Stein, 1999). Specifically, given a family of GPs parametrized by θ, if Pθ1 ⇐⇒ Pθ2 with  θ1 = θ2, then θ is not identifiable since we cannot distinguish between θ1 and θ2. As a corollary,there does not exist any consistent estimator for θ
> >
> > To rephrase, equivalent processes can not be distinguished from a finite sample. However,
> > to my understanding,  this only means that there is no decision rule that works **with probability 1**.  This still leaves the possibility of designing decision rules that work with probability **arbitrarily close to 1**.
> > In particular, the statement
> >
> > >As a corollary,there does not exist any consistent estimator for θ
> >
> > is not clear. Can the authors provide precise reference (paper, or book chapter and page) for this statement?
> >
> > Is it true that for any two members of an equivalence class, one can not distinguish between them with any probability larger than 0.5? What does this probability depend on?
> > Please clarify this point.
> >
> > Provided that everythong above is correct, the notion of equivalence in this paper is simply too weak to be useful in ML. If the authors believe otherwise, please explain.
> >
> >
> >
> > P.S.
> >
> > As an additional note, the results in this paper are sufficent, but not necessary. But since equivalence is weak, sifficiency is easy to satisfy. The equivalence does not capture the finer structure of the process. This may explain the differences of opinion between claims in the paper regarding importance of smoothness, and contrary observations by other reviewers. However, I have not considered this point in depth so far.

---

> > > ### Author Response · Authors · 2023-08-17
> > > **Clarifications to further questions from Reviewer LXB2**
> > >
> > >
> > > Thank you for the thoughtful and detailed feedback. We address each of your queries in sequence.
> > >
> > > **About finite set**
> > >
> > > We appreciate the opportunity to clarify the distinction between a "finite index set" and "finite samples". A finite index set means that the domain of the GP possesses a finite cardinality, while the weaker notion "finite samples" means that the sample size $n$ is finite.
> > >
> > > It's worth noting when the domain is finite, employing a Gaussian process, or indeed any continuous process, might not be the most apt choice. However, situations characterized by finite samples are considerably more prevalent and realistic in the realm of ML. To illustrate, in our Application 1, the domain constitutes a square with infinite cardinality, and even though the number of pixels is finite, the overall context remains relevant for our discussion.
> > >
> > >
> > > **On the clarity of the statement and precise reference**
> > >
> > > The concepts of indistinguishability of equivalent GPs and parameter non-identifiability can be founded in foundational discussions available in Ibragimov and Rozanov (1978 Chapter III), Yadrenko (1983), Stein (1999 Chapter 4) and Zhang (2004). For a precise reference, we refer to Zhang's work in "Inconsistent estimation and asymptotically equal interpolations in model-based geostatistics," Journal of the American Statistical Association, 2004. Specifically, Page 252, left column, Line 4 addresses this concern. Zhang's discussion on the equivalence of measures aligns with our statement: "Moreover, if $\{P_\theta:\theta\in\Theta\}$ is a family of equivalent measures and $\hat{\theta}_n,n\geq 1$ is a sequence of estimators, then, irrespective of what is observed, $\theta_n$ cannot be weakly consistent estimator for all $\theta\in\Theta$."
> > >
> > > **About distinguishing equivalent GPs with probability more than 0.5**
> > >
> > > Thank you for this astute question. To the best of our knowledge, discussions have traditionally centered around the intriguing case of probability 1. We've taken the liberty to furnish a more streamlined proof, particularly addressing your query. Here's an outline of the argument to demonstrate that the scenario you've posited is untenable.
> > >
> > > Given that $P_1\equiv P_2$, let $D$ be a decision rule, a function that maps a realization of the GP to a binary output, either 1 or 2. Here, an output of $D(f)=1$ means that we believe $f$ is generated from $P_1$, and similar for $D(f)=2$. Now, assume that $Pr_{f\sim P_1}(D(f)=1)=a>0.5$, that is, the probability to correctly detect the true data-generating model is greater than 0.5 (better than random guess). However, since $P_1\equiv P_2$, we have $Pr_{f\sim P_2}(D(f)=1)=Pr_{f\sim P_1}(D(f)=1)=a>0.5$, meaning that  the probability of incorrectly identifying the true data-generating model also surpasses 0.5, which is a clear contradiction.
> > >
> > >
> > > **Addressing the utility of the notion of equivalence in ML and its sufficiency**
> > >
> > > A primary objective of our paper is to highlight that aside from $w_1\sigma_1^2\alpha_1^{2\nu_1}$, all individual parameters, $w_l$, $\sigma_l^2$, $\alpha_l$, for any $l=1,2,\cdots,L$, are non-identifiable. We respectfully offer a different perspective: to study the identifiability, this direction is precisely what we aimed for. Moreover, Gaussian random measures are either equivalent or orthogonal (Stein 1999, page 114). A corollary of our Theorem 2 states: if two GPs are orthogonal, then $w_1\sigma_1^2\alpha_1^{2\nu_1}\neq \widetilde{w}_1\widetilde{\sigma}_1^2\widetilde{\alpha}_1^{2\nu_1}$. Additionally, a significant conclusion of our research is, a mixture of Matern kernels is equivalent to its least smooth component. Even if there is a reason to use it for real data application, we recommend caution in parameter interpretation. For instance, $w_l$ does not necessarily represent the weight of the $l$-th component. We view this as a major contribution of our paper: "understanding identifiability and interpretability is so crucial in using GPs but rarely discussed" (Reviewer PY7Z).
> > >
> > >
> > >  **Regarding differing opinions on the importance of smoothness**
> > >
> > > We seem to have achieved a consensus regarding the discrepancies in certain experiments. To succinctly express the underlying reason: the MSEs from equivalent measures are asymptotically equal (Stein 1993 Theorem 1). Meaning, a substantial sample size is essential to witness the convergence of the MSEs. For the finite sample regime, we furnished empirical evidence complemented by theoretical insights for the disparities in the MSEs between equivalent GPs. Crafting a comprehensive theory remains a formidable challenge and an open problem, even for simplistic, singular kernels. For a detailed understanding, we kindly direct you to our discussion with Reviewer j39y due to the character limit.
> > >
> > > We deeply appreciate your insights. We are always open to continued discussions and the exploration of additional experiments.

---

> > > > ### Comment · Reviewer_j39y · 2023-08-17
> > > >
> > > > I'm sorry for jumping in here. This discussion is very interesting and just want to contribute my thoughts.
> > > >
> > > > > To rephrase, equivalent processes can not be distinguished from a finite sample. However, to my understanding, this only means that there is no decision rule that works with probability 1. This still leaves the possibility of designing decision rules that work with probability arbitrarily close to 1.
> > > >
> > > > I believe that this understanding is right. I have the following example in mind.
> > > >
> > > > Consider a GP with an exponential kernel with length scale $1000$ and variance $1000$ and a GP with an exponential kernel with length scale 1 and variance 1. I believe that two GPs with exponential kernels are are equivalent if $\sigma_1^2 / \ell_1 = \sigma_2^2 / \ell_2$, which here is the case. Now consider them on $[0, 10^6]$. Theory says that it isn't possible to determine with probability one, on the basis of a single sample, which one it is. However, simply by taking points $f(0), f(10^4), f(2 \cdot 10^4), f(3 \cdot 10^4), \ldots, f(10^6)$ (all roughly independent because they are spaced much further apart than the largest length scale) and computing the empirical variance, this number will either be closer to 1 or to 1000 and will give with high probability, though strictly lower than one, a very solid guess as to which one it.
> > > >
> > > > As to the proof of the authors, it is not clear to me why $P_1 \equiv P_2$ implies that $Pr_{f\sim P_2}(D(f)=1)=Pr_{f\sim P_1}(D(f)=1)$. This step may not be right. Equivalence simply means that absence of a decision rule with probability one on the basis of a single sample.

---

> > > > > ### Author Response · Authors · 2023-08-17
> > > > > **Further discussion with Reviewers LXB2 and j39y**
> > > > >
> > > > >
> > > > > Thank you, Reviewer j39y, for your keen interest in our dialogue with Reviewer LXB2. We deeply appreciated our prior discussions with you; they were enlightening and indeed motivating.
> > > > >
> > > > > The topic you and Reviewer LXB2 have broached is intriguing and could potentially pave the way for another project, focusing on the finite sample behavior of GPs and their associated prediction MSEs.
> > > > >
> > > > > We concur with the validity of the example provided by Reviewer j39y. To further this, consider an even more stark scenario: if we are presented with only a single sample, the lengthscale parameter wouldn't have an impact on the covariance matrix (which would simply be a scalar in this instance). If we hypothesize that the larger value among two observed $y$'s originates from the first kernel, we might accurately determine this with a probability in $(0.5,1)$ (open interval on both sides). In our conceptual outline of a potential proof, we used the term $f$ to represent a GP realization, which is a function over $\Omega$, not just a function observed at limited points. This concept is also resonant with the likelihood test discussions found in Stein 1999 (Pages 115-117), which requires the sample size to go to infinity.
> > > > >
> > > > > As both of you insightfully pointed out, it's impossible to distinguish equivalent GPs or identify their parameters **with probability 1**. This leads to a captivating phenomenon: in many analytical endeavors, our confidence in an estimator grows with increasing sample size. But in this specific situation, the limited sample size means our certainty isn't nearing 1. As the sample size increases, one would expect heightened confidence, but due to existing theories (even for singular Matern kernels as referred in Stein 1999 and Zhang 2004), the two realizations begin to resemble each other more closely.
> > > > >
> > > > > Yet, it's crucial to note that this disparity between asymptotic and finite sample cases does not undermine our core assertion: **interpreting individual parameters in the mixture kernel requires caution**.
> > > > >
> > > > > We are in complete agreement that our theorem 2 predominantly addresses the asymptotic domain. As previously elaborated in our discussion with Reviewer j39y, the finite sample situation, though highly compelling, remains a formidable challenge in deriving a thorough theoretical result.
> > > > >
> > > > > For clarity, we've highlighted these thought-provoking yet unresolved issues for potential future research. We believe they will be of equal interest to you:
> > > > >
> > > > > 1. For a fixed sample size, $n$, and given two equivalent kernels (even just a singular Matern 1/2 kernel, as an instance), what is the probability of distinguishing between them? The parameters of both kernels should factor into this probability.
> > > > >
> > > > > 2. In a similar setting, how can we quantitatively analyze the relationship between prediction MSEs?
> > > > >
> > > > > We are immensely grateful to both reviewers for presenting these compelling queries and for the profound discussions they've sparked.

---

> > > > > > ### Author Response · Authors · 2023-08-21
> > > > > > **Summary of Discussions and Responses: Addressing Reviewer LXB2's Feedback**
> > > > > >
> > > > > > We truly appreciate the time and effort the reviewer has invested in evaluating our work. As we approach the end of the discussion period, we feel it's apt to provide a brief recap of our discussions and the subsequent steps undertaken.
> > > > > >
> > > > > > **Regarding equivalence**
> > > > > >
> > > > > > We extend our gratitude to the reviewer for highlighting the need to clarify certain notions for a broader audience. All sections relating to equivalence and identifiability have been carefully revised to make them more accessible, together with a detailed literature review section in the supplement. This offers readers a deeper understanding of the concepts and their origins. Moreover, we have included a corollary of Theorem 2 showing that the MSEs of the Matern mixture kernel and its least smooth component are asymptotically equal. This aspect might particularly resonate with the ML audience.
> > > > > >
> > > > > > **Regarding the finite sample**
> > > > > >
> > > > > > Building upon our previous discussion, we concur with the reviewer's observation regarding the limitations of our theorem in a constrained sample size context. We wish to highlight that the literature on parameter identifiability within the Matern kernel predominantly centers around the infinite sample scenario, as evidenced by works like Stein (1990, 1991, 1993, 1999), Zhang (2004), and Tang et al. (2021). The inherent challenges in finite samples are well-documented in these contributions. While our current understanding doesn't allow us to formulate theories on this matter, we've endeavored to glean empirical insights by integrating additional experiments into our study. These experiments, elaborated upon in our discussions with Reviewer j39y, provide some compelling observations.
> > > > > >
> > > > > > Although securing a conclusive theoretical support for the finite sample regime is challenging, we managed to provide some insights to interpret our findings. To do so, we have to delve further into the proof of Stein's Theorem 1 in Stat \& Prob Letter (1991), which guides us to Theorem 3.1 in Stein (AoS 1990). The proof suggests that the relative difference between the MSEs rests on the tail of the series in the second-last line on Page 854. This difference is influenced by both the sample size (denoted as 'N' in Stein 1990) and by $b_{jk}$ and $\mu_j$, as defined on the same page. For a fixed sample size, smaller values of $(b_{ij}+\mu_j\mu_k)^2$ result in a smaller relative difference between MSEs. By the definition of $b_{jk}$ and $\mu_j$, the more "different" the two kernels are, in other words, the more "significant" the additional smoother components become — the greater the difference between the MSEs will be.
> > > > > >
> > > > > > In light of the intricate challenges presented by finite sample scenarios, we've expanded our manuscript to encompass a more nuanced discussion on this very topic. We've taken care to shed light on the distinction between MSEs and the probability of discerning between them, both of which are of paramount importance in understanding the nuances of our study.
> > > > > >
> > > > > > **Regarding new experiments**
> > > > > >
> > > > > > During the rebuttal and discussion phases, we undertook significant efforts to enhance our paper's depth and breadth. In our initial rebuttal, we incorporated six new experiments, examining our theory across larger sample sizes, utilizing different optimizers, and probing diverse data types. Subsequently, during the discussion period, we delved deeply into experiments that explore the intricacies of the finite sample scenario. We believe that these additions make our paper more robust and well-rounded, providing a comprehensive and credible account of our findings.
> > > > > >
> > > > > >
> > > > > > **Regarding the main message**
> > > > > >
> > > > > > We'd like to reiterate the core essence of our paper for clarity. While the mixture kernel is widely recognized and employed within the ML community, discussions regarding the identifiability of its parameters have been sparse. Our work seeks to bridge this gap. We assert that the mixture of Matern kernels is equivalent to its least smooth component, and that only the microergodic parameter of its least smooth kernel can be consistently estimated. Consequently, in large sample situations, there is limited benefit in including the smoother components, while in small sample regimes, even if there is a practical reason to use the mixture kernel, we recommend caution when interpreting each individual parameter, which is not identifiable. We hope this clarification underscores the unique contribution of our research.
> > > > > >
> > > > > >
> > > > > > We'd like to extend our heartfelt gratitude to the reviewer for the meticulous examination of our paper. The discussions have been both enlightening and enjoyable, significantly aiding in refining our work to resonate more clearly. We have invested significant effort into addressing concerns and enhancing our paper's depth and clarity. We hope we've successfully addressed your concerns, and it's our sincere wish that our efforts are viewed favorably in your evaluation. Once again, thank you for your invaluable insights and guidance.

---

### Official Review · Reviewer_dNVU · 2023-07-07

**Soundness:** 3 good
**Presentation:** 3 good
**Contribution:** 3 good
**Rating:** 6
**Confidence:** 3

**Summary:**

This paper investigates additive and separable mixture kernels in the context of Gaussian process regression. Concretely, it tests the intuition behind the convex combination of Matern processes and identifies limitations in the interpretability of the resulting mixture kernel that might contradict the intuition of the modeler.
It further analyzes the identifiability of parameters of separable Kernels in multi-output GPs.

**Strengths:**

- I believe this paper is a useful contribution to understanding when intuition about modeling decisions diverges from the actual impact of these decisions. It thus earns its place at a ML conference in that it strictly extends the pool of resources that should be considered when designing a GP model
- NeurIPS seems to me as the right venue to publish these results in that format, since it manages a good balance between mathematical rigor and intuition, which makes the text very approachable for more application-oriented researchers, as well.
- the overall presentation follows a very clear structure that is easy to follow. I appreciate that the authors do not obscure meaningful results behind unnecessarily complicated formulations.

**Weaknesses:**

- The plots are aesthetically rather unpleasing and the presentation of the results would, in my opinion, profit quite significantly from a polishing of the figures.
- The authors focus solely on Matern processes, to the point where the notion of "kernel" is used interchangeably with "Matern-kernel". This limitation, however, is understandable in light of the page limit and is also addressed in the discussion section. (It does not affect my score.)

**Questions:**

- I would like to understand the impact of the "smoothness-of-mixed-Matern-kernel-is-determined-by-its-least-smooth-component" argument better:
	- Is it really only about smoothness? This seems rather obvious to me (and I am certain this has been known before). But Theorem 2 states that the kernel is equivalent, and hence makes a more general statement about the structure of the functions from this kernel's RKHS, than merely smoothness, right? As somebody who is not familiar with proofs about identifiability and equivalence of kernels: apart from smoothness of the resulting sample paths, does mixing in more Matern components not add *any* structure to a single (least-smooth) Matern component? I could imagine that other readers would also appreciate a clarifying statement that brings together the "same-smoothness" and "equivalence" arguments.
	- What reinforces me in my endeavor to understand the impact of this statement better is also the fact that in Section 7, you write that "[...] the inclusion of kernels with different smoothness does not necessarily improve prediction accuracy".
	  On the other hand, Theorem 2 states that it does absolutely make no difference (not only "not *necessarily*"). Could you elaborate more on the impact of this statement and its strength?
- Suggestion: the paper includes a comparison between (i) a mixture of kernels with varying smoothness parameters and (ii) a kernel consisting only of the least-smooth component. It would be interesting to see the comparison to (iii) the other (two?) components, as well. I would expect these results would be different then. But *how* they would be different (by how much or - e.g. in the case of Figure 6 - how they differ structurally) would be quite interesting, measured by the expected effort of this addition.

**Limitations:**

Limitations of the analysis presented in the text are adequately addressed in the discussion section.

---

> ### Author Rebuttal · Authors · 2023-08-09
>
>
> We are deeply grateful to the reviewer for your thoughtful and thorough examination of our manuscript. Your observations and suggestions have been invaluable in refining the narrative and content of our paper. While we're unable to present the full revised manuscript at this time, please find our detailed responses to your comments below. Further clarifications, especially in relation to the figures, can be found in the accompanying 1-page PDF for rebuttal. Your feedback has been instrumental, and we genuinely appreciate the effort and expertise you have devoted to our work.
>
> **Weakness 1: about the plots**
>
> A: Thank you for the comment. We have meticulously revised and polished all the figures. Since we are not allowed to upload the revised manuscript, please see the additional figures provided in the one-page PDF for rebuttal.
>
> **Weakness 2: about the use of "kernel" vs "Matérn kernel"**
>
> A: Thank you for understanding the scope of our work within the constraints of the page limit. To ensure clarity and to avoid any possible confusion, we have revised the manuscript to consistently use "Matérn kernel" instead of the more generic term "kernel."
>
> **Question 1.1: about smoothness**
>
> A: Thank you for this insightful comment. While Theorem 1 primarily emphasizes the smoothness aspect, Theorem 2 delves into a deeper equivalence, which articulates that the Gaussian random measure associated with the mixture of Matérn kernels is fundamentally equivalent to that of the least smooth single Matérn. This means that beyond the superficial similarities in smoothness, the underlying structures of functions from the kernel's RKHS are also equivalent. In essence, incorporating additional Matérn components doesn't introduce new structural nuances beyond what the least-smooth component already encapsulates.
>
> To aid understanding for our readers, we've appended a paragraph after Theorem 2 to bridge the connection between the smoothness result in Theorem 1 and the deeper equivalence presented in Theorem 2. Your feedback has illuminated an important aspect, and we believe this addition will offer more clarity to the readers.
>
> **Question 1.2: about the conclusion in Section 7**
>
> A: Thank you for highlighting this discrepancy. You are absolutely right that there seems to be a contrast between the theoretical implications of Theorem 2 and the conclusion in Section 7.
>
> The fundamental insight from Theorem 2 is that, from a purely theoretical standpoint, the inclusion of additional Matérn components with varying smoothness levels does not bring about any change in the underlying structure or representation capabilities of the kernel. Hence, when our model assumptions hold, one shouldn't expect improvements in prediction accuracy merely by adding kernels of different smoothness.
>
> However, real-world data and applications often exhibit nuances and complexities that might not align perfectly with our theoretical model assumptions. One potential reason for the discrepancy in performance between a single kernel and a mixture kernel in real data settings could be the increased parameter space in the mixture model, making parameter estimation more challenging. As a result, despite the potential for greater flexibility, the mixture kernel might not always result in improved prediction due to potential sub-optimal parameter estimations.
>
> Another aspect to consider is the potential mismatch between our chosen model (GP with specific kernels) and the true underlying process of the real-world data. If the real-world function doesn't strictly adhere to a Gaussian Process or doesn't exactly reside in the presumed RKHS, discrepancies between theoretical predictions and empirical observations can emerge.
>
> In summary, while Theorem 2 provides a strong theoretical foundation asserting the equivalence of a single least-smooth Matérn kernel and its mixture counterpart, real-world applications introduce complexities that can sometimes deviate from our theoretical insights. We have further elucidated this point in the revised manuscript to ensure that the relationship between theoretical findings and empirical observations is clear to our readers.
>
> **Question 2: suggestion about additional experiments related to Figure 6**
>
> A: We are grateful for the reviewer's insightful suggestion, emphasizing the potential to further elucidate our arguments. The proposed comparison between the additional components is not only intriguing but will also serve to provide more comprehensive insight into our research.
>
> Acknowledging the feedback, and in alignment with remarks from reviewer j39y, we recognize that the Pollen dataset might not be the most optimal choice for demonstrating the nuances of the Matérn kernel. Taking this into account, we have transitioned to utilizing the Mauna Loa $CO_2$ time series data (Figure 3 in the 1-page PDF for rebuttal). We observe similar performance between Matérn mixture $1/2+3/2+5/2$ and Matérn $1/2$, as well as the resemblance between Matérn mixture $3/2+5/2$ and Matérn $3/2$. At the same time, while the performance of Matérn $3/2$ exceeds that of Matérn $1/2$, underscoring the dataset's inherent differentiability, it's noteworthy that the integration of the $3/2$ component to $1/2$ in a mixture kernel doesn't elevate its performance to match that of Matérn $3/2$. This observation supports our contention: the performance of a Matérn mixture kernel is chiefly dictated by its least smooth component.

---

> > ### Comment · Reviewer_dNVU · 2023-08-14
> >
> > Many thanks for the response.

---

> > > ### Author Response · Authors · 2023-08-15
> > > **Further clarification and experiments for Reviewer dNVU's Suggestion**
> > >
> > >
> > > We are grateful that you value our work from the first place. We'd like to update you on some recent enhancements to our manuscript.
> > >
> > > In response to your suggestion, we have expanded our comparison to include results for other components, showcased in Figure 3 in the rebuttal 1-page PDF. Building on your insights regarding performance distinction, we were motivated to delve deeper. Taking cues from both Reviewer PY7Z—who highlighted the potential context of the Mauna Loa dataset, suggesting Matern 1/2 might not be the optimal choice—and Reviewer j39y—who raised intriguing questions about finite sample behavior—we embarked on further experiments. These were crafted to affirm our previous findings and introduce new perspectives.
> > >
> > > To provide a concrete response, we applied the following six kernels to the Mauna Loa dataset, covering the years from 1960 to 2020: 1/2+3/2+5/2, 1/2+3/2 ,1/2, 3/2, 3/2+5/2, 5/2. By varying the training sample size from 10% to 90%, and performing 10 replications for each size, we gauged performance across different training sizes. The resultant RMSE (std) values for all kernels are detailed below:
> > >
> > >
> > >     %   1/2+3/2+5/2  1/2+3/2        1/2        3/2         3/2+5/2      5/2
> > >     10  2.19 (0.09) 2.2 (0.09)  2.55 (0.12) 2.28 (0.07) 2.26 (0.13) 2.28 (0.07)
> > >     20  1.96 (0.06) 1.97 (0.06) 2.38 (0.08) 2.48 (0.53) 2.04 (0.19) 2.22 (0.04)
> > >     30  1.74 (0.05) 1.74 (0.05) 2.03 (0.11) 2.20 (0.28) 1.57 (0.06) 2.19 (0.03)
> > >     40  1.49 (0.07) 1.50 (0.07) 1.61 (0.12) 1.35 (0.18) 1.90 (0.43) 1.94 (0.38)
> > >     50  1.25 (0.06) 1.25 (0.06) 1.28 (0.09) 0.97 (0.19) 1.41 (0.64) 1.56 (0.64)
> > >     60  1.01 (0.09) 1.01 (0.09) 0.98 (0.10) 0.69 (0.13) 0.96 (0.62) 1.16 (0.70)
> > >     70  0.85 (0.09) 0.85 (0.10) 0.82 (0.10) 0.52 (0.12) 0.52 (0.08) 0.81 (0.72)
> > >     80  0.70 (0.07) 0.70 (0.07) 0.68 (0.07) 0.40 (0.03) 0.40 (0.04) 1.08 (0.94)
> > >     90  0.57 (0.04) 0.57 (0.04) 0.55 (0.04) 0.35 (0.03) 0.35 (0.03) 1.42 (0.93)
> > >
> > > Key observations emerged:
> > >
> > > The mixture kernel 1/2+3/2+5/2 aligned closely with 1/2+3/2 across all experiments. Further, these three (1/2+3/2+5/2, 1/2+3/2, 1/2) converge when the sample size ratio reaches 50%. This empirical finding supports the theoretical equivalence of these three GPs as posited in Theorem 2. Similarly, the 3/2+5/2 kernel and the standalone 3/2 kernel converge from a 70% sample size, lending empirical weight to their theoretical equivalence. We added a corollary claiming that the MSE of the mixture kernel asymptotically matches the MSE of its least smooth component.
> > >
> > > A noteworthy trend is the exceptional performance of the Matern 3/2 kernel across all experiments, echoing the reviewers' conjecture of the dataset admitting a smooth structure. Its influential performance, however, gets diluted when coupled with the less smooth component, Matern 1/2. Conversely, when mixed with the smoother Matern 5/2 component, the lesser performance of the latter is overshadowed by Matern 3/2. These behaviors lend empirical weight to our Theorem 2, suggesting that in the asymptotic sense, the mixture kernel is dominated by its least smooth component.
> > >
> > > In the realm of finite samples, we observed intriguing nuances. Theoretically, while the combined kernels 1/2+3/2+5/2, 1/2+3/2, and the standalone 1/2 are deemed equivalent, and the 3/2+5/2 is akin to 3/2, their practical agreement points i.e., 50%, 70%, differed with limited samples. Although securing a conclusive theoretical support for the finite sample regime is challenging, we managed to provide some insights to interpret our findings. To do so, we have to delve further into the proof of Stein's Theorem 1 in Stat \& Prob Letter (1991), which guides us to Theorem 3.1 in Stein (AoS 1990). The proof suggests that the relative difference between the MSEs rests on the tail of the series in the second-last line on Page 854. This difference is influenced by both the sample size (denoted as 'N' in Stein 1990) and by $b_{jk}$ and $\mu_j$, as defined on the same page. For a fixed sample size, smaller values of $(b_{ij}+\mu_j\mu_k)^2$ result in a smaller relative difference between MSEs. By the definition of $b_{jk}$ and $\mu_j$, the more "different" the two kernels are, in other words, the more "significant" the additional smoother components become — the greater the difference between the MSEs will be.
> > >
> > > In conclusion, we have delved deeply into the intricacies of all six kernels, as you've rightly pointed out. In the realm of infinite samples, we've provided robust theoretical backing. For the finite sample domain, despite the inherent challenges and ambiguities, our discourse offers both theoretical and empirical insights.
> > >
> > > We are genuinely grateful for your meticulous comments. We anticipate that the newfound depth and clarity in our revised content align with the broader objectives of our paper and we hope these enhancements can be viewed favorably in your evaluation.

---

### Official Review · Reviewer_PY7Z · 2023-07-12

**Soundness:** 3 good
**Presentation:** 3 good
**Contribution:** 3 good
**Rating:** 7
**Confidence:** 3

**Summary:**

This work looks at idenfiability in the context of mixture kernels in GP regression. They look at additive mixtures of Matern kernels and multivariate separable kernels in the context of multivariate GPs (multiple outputs). They show theoretically and through simulation that ML-II learning cannot discover the hyperparameters in the mixture corresponding to the different components as they are not microergodic (Stein, 1999). On the contrary, only products of hyperparameters in specific combinations is identifiable. The simulation studies on synthetic data are interesting - through repeated experiments they show that ML-II does not converge to the true hyperparametes in a mixture kernel and only the microergodic parameters are discovered.

**Strengths:**

A lot of the theory on identifiability has been scattered in literature like Zhang (2004) and Stein's textbook (1999) but I would argue that a succint and targetted presentation such as this one might benefit the community and hence can be significant.

**Weaknesses:**

This work begs the question - what happens in the large data limit when n -> \infty, as the simulations and applications only address the finite data limit. My guess is that the parameters which are not microergodic remain unidentifiable but some comment is needed.


**Questions:**

- Theorem 2 'p' is introduced without defining. I've had to figure that it is the dimension.
- Perhaps tangential to this work but it would be interesting to reason about the identifiability of the noise variance when a mixture kernel is used.
- How does the additive GPs work where the constructed kernel is additive and considers all combinations of dimensions in the data  fit with the identifiability insights of this work.
- About the dominance of the least smooth kernel component, how does one extend the insight to mixtures of different kernel types like Matern + RQ or periodic + Matern.
- Line 281 / 282 pls cite Lalchand et al, Generalised GPLVM with Stochastic Variational Inference, AISTATS 2022.
- Line 47/ 48 pls cite Simpson et al, Kernel Identification with Transformers, NeurIPS, 2021.


**Limitations:**


The most obvious one - which the authors mention in conclusions is the extension of these insights to mixture of kernel families which are used very frequently to model different effects in univariate functions. eg. CO2 Mauna Loa kernel.

---

> ### Author Rebuttal · Authors · 2023-08-09
>
>
> Thank you for presenting us with such insightful questions and observations regarding our manuscript. Given the submission constraints, we're unable to provide an updated version of the full manuscript at this juncture. However, to facilitate a more tangible understanding of our clarifications and to provide visual support, we've attached a 1-page PDF with addditional figures. In the subsequent sections, we proceed with a point-to-point response to each of your comments.
>
> **Weakness about large sample**
>
> A: We appreciate this observation. Our theoretical framework is indeed designed for the infinite sample scenario. Consequently, as $n \to \infty$, the conclusion should remain consistent, wherein only the proposed microergodic parameter is identifiable, while others are not. We have clarified this in the manuscript and conducted additional simulations for $n=1000,3000$. The conclusion remains consistent (Figure 1 in the one-page PDF).
>
>
> **Question 1 about $p$**
>
> A: Thank you for pointing this out. Indeed, we defined all domains in this manuscript as $\Omega = \mathbb{R}^p$ in line 94. Recognizing the distance between this and Theorem 2, we have explicitly defined $p$ as the dimension of the domain within Theorem 2 in our revised manuscript.
>
> **Question 2 about noise variance**
>
> Thank you for raising this intriguing point. In the majority of existing literature, researchers often work under the assumption of zero noise (no nugget), primarily due to the complexities associated with noise inclusion (nowhere continuous). As of our current knowledge, the most recent investigations into this matter can be found in papers by Tang, Zhang, Banerjee (JRSSB 2022) and Loh and Sun (Bernoulli 2023). These studies specifically focus on a single Mat'ern kernel rather than a mixture kernel. To encapsulate their findings briefly, they concluded that the noise variance is indeed identifiable, and the presence of noise does not undermine the identifiability of the Mat'ern parameters. Extending these findings to our mixture kernel remains challenging but is a fruitful avenue for future research.
>
> **Question 3 about adding up all dimensions**
>
> A: This question delves deep into an intriguing facet of our research. We suppose you're referring to a kernel of the form $K(x,x') = \sum_{l=1}^p w_l \text{Matern}(x_l-x_l';\sigma^2_l,\alpha_l,\nu)$, where $x_l$ represents the $l$-th coordinate of $x$.
>
> If so, regrettably, the primary tool we utilized for proving equivalence of measures isn't directly applicable here. This is primarily due to the non-fulfillment of the first condition in the integral test (Yadrenko 1983), which expects the spectral density to behave as $1/\omega^r$ for some positive $r$. Our conjecture, based on our current understanding, is as follows: Individual parameters may not be identifiable. However, the term $w_l\sigma_l^2\alpha_l^{2\nu}$ remains identifiable, for each $l=1,...,p$.
>
> To underpin this conjecture, we employed distinct Mat'ern $1/2$ kernels for every dimension of a synthesized 2-dimensional dataset ($p=2$). Our observations indicate that both $w_1\sigma_1^2\alpha_1^{2\nu}$ and $w_2\sigma_2^2\alpha_2^{2\nu}$ are identifiable (Figure 2 in the one-page PDF).
>
> One can interpret these findings as follows: When our domain is restricted to a 1-D line in 2-D, for instance where $x_1\neq 0$ and all other dimension are set to zero ($x_2=...=x_p=0$), the kernel morphs into a 1-D Matern supported solely on the $x_1$-axis. Based on established theory, $w_1\sigma_1^2\alpha_1^{2\nu}$ becomes identifiable.
>
> This logic can be analogously applied to other dimensions. However, it's imperative to note that this isn't a rigorous proof given the absence of a conclusive integral test. Your question has certainly shed light on a promising direction for future research. We're grateful for this thought-provoking inquiry.
>
>
>
> **Question 4 about RQ and periodic kernels**
>
> A: Thank you for posing this insightful query. Notably, RQ and periodic kernels do not have analytic spectral densities, which are essential for a theoretical analysis of GPs. Extending our insights to such kernel mixtures would necessitate the application of more advanced mathematical techniques, particularly within probability theory. We recognize this as a promising area for future research.
>
>
> **Question 5 and 6 about references**
>
> Thank you for drawing our attention to these relevant works. We have now incorporated citations to both Lalchand et al. (AISTATS 2022) and Simpson et al. (NeurIPS 2021) in the appropriate sections of our revised manuscript.
>
>
> **Limitation about Co2 Mauna Loa kernel**
>
> A: Thank you for highlighting this pertinent point. The CO2 Mauna Loa dataset and its associated kernel combination serve as iconic examples in the Gaussian Processes literature, demonstrating the power of combining different kernels to encapsulate diverse features of a dataset. While our study emphasizes the theoretical underpinnings of mixture kernel types, integrating these findings into practical, often-used kernel combinations like those for the CO2 Mauna Loa dataset would undeniably deepen our understanding. As we noted in our discussion, broadening our framework to encapsulate other kernel families is an intriguing and vital future research direction.
>
> However, we'd like to underscore that extending these insights to some kernel families can be particularly challenging due to the absence of an analytic form of their spectral densities. Instead, we performed experiments on this dataset with Matérn kernels, see Figure 3 in the one-page PDF for rebuttal. On this new dataset, we can see that Matérn 1/2 performs similarly with the mixture of Matérn 1/2, 3/2 and 5/2; and Matérn 3/2 performs similarly with the mixture of Matérn 3/2 and 5/2, in terms of MSE, further justifying our Theorem 2.

---

> > ### Comment · Reviewer_PY7Z · 2023-08-15
> > **Post rebuttal update**
> >
> > Thank you for putting in the time and work towards the response and tackling most of the comments and questions raised. I would hope that you would add some of this discussion into the manuscript if accepted, esp. the the large data limit and identifiability of noise variance. Overall, I think understanding identifiability and interpretability is so crucial in using GPs but rarely discussed.
> >
> > I am happy to support this paper by raising my score to a 7 but keeping my confidence intact at 3.

---

> > > ### Author Response · Authors · 2023-08-16
> > > **Some updates**
> > >
> > > Thank you for recognizing the efforts we've put into addressing your comments and concerns. We genuinely value your constructive feedback, which has undeniably improved the caliber of our paper. We're excited to share further insights and updates related to the discussion section.
> > >
> > > **Regarding the Large Data Limit**
> > >
> > > Your acknowledgment of the Mauna Loa kernel's relevance is deeply appreciated. While we primarily engage with the Matern mixture kernel, the Mauna Loa dataset's significance within our theoretical framework became evident. Our preliminary investigations are captured in Figure 3 of the rebuttal's 1-page PDF. Informed by the insights from both Reviewer dNVU and Reviewer j39y, we embarked on extended experiments. These not only reinforced our earlier findings but also presented novel viewpoints.
> > >
> > > To provide a concrete response, we applied the following six kernels to the Mauna Loa dataset, covering the years from 1960 to 2020: 1/2+3/2+5/2, 1/2+3/2 ,1/2, 3/2, 3/2+5/2, 5/2. By varying the training sample size from 10% to 90%, and performing 10 replications for each size, we gauged performance across different training sizes. The resultant RMSE (std) values for all kernels are detailed below:
> > >
> > >
> > >     %   1/2+3/2        1/2        3/2
> > >     10  2.20 (0.09) 2.55 (0.12) 2.28 (0.07)
> > >     20  1.97 (0.06) 2.38 (0.08) 2.48 (0.53)
> > >     30  1.74 (0.05) 2.03 (0.11) 2.20 (0.28)
> > >     40  1.50 (0.07) 1.61 (0.12) 1.35 (0.18)
> > >     50  1.25 (0.06) 1.28 (0.09) 0.97 (0.19)
> > >     60  1.01 (0.09) 0.98 (0.10) 0.69 (0.13)
> > >     70  0.85 (0.10) 0.82 (0.10) 0.52 (0.12)
> > >     80  0.70 (0.07) 0.68 (0.07) 0.40 (0.03)
> > >     90  0.57 (0.04) 0.55 (0.04) 0.35 (0.03)
> > >
> > > Key observations emerged
> > >
> > > Here we highlighted the result between (1/2+3/2, 1/2, 3/2), (detailed results for all six kernels could refer to our comments to Reviewer dNVU: "Further clarification and experiments for Reviewer dNVU's Suggestion"). The mixture kernel 1/2+3/2 and Matern 1/2 converge when the sample size ratio reaches 50%. This empirical finding supports the theoretical equivalence of these three GPs as posited in Theorem 2. We added a corollary claiming that the MSE of the mixture kernel asymptotically matches the MSE of its least smooth component.
> > >
> > > In the realm of finite samples, we observed intriguing nuances. Theoretically, while the mixture kernels 1/2+3/2, and the standalone 1/2 are deemed equivalent, their performance could potentially distinct with limited samples. Although securing a conclusive theoretical support for the finite sample regime is challenging, we managed to provide some insights to interpret our findings. To do so, we have to delve further into the proof of Stein's Theorem 1 in Stat \& Prob Letter (1991), which guides us to Theorem 3.1 in Stein (AoS 1990). The proof suggests that the relative difference between the MSEs rests on the tail of the series in the second-last line on Page 854. This difference is influenced by both the sample size (denoted as 'N' in Stein 1990) and by $b_{jk}$ and $\mu_j$, as defined on the same page. For a fixed sample size, smaller values of $(b_{ij}+\mu_j\mu_k)^2$ result in a smaller relative difference between MSEs. By the definition of $b_{jk}$ and $\mu_j$, the more "different" the two kernels are, in other words, the more "significant" the additional smoother components become — the greater the difference between the MSEs will be.
> > >
> > > In essence, we have deepened our exploration into the differences between finite and infinite sample scenarios. While our foundational theories rest upon infinite samples, we've endeavored to give both theoretical and practical insights into the finite sample scenario.
> > >
> > > **On the Identifiability of Noise Variance**
> > >
> > > Post rigorous deliberation, we have deduced the analogue of Theorem 2 concerning noise (nuggets). Let $\tau^2$ and $\widetilde{\tau}^2$ be the noise variance of $K$ and $\widetilde{K}$, then if $\tau^2\neq\widetilde{\tau}^2$ then $K\not\equiv \widetilde{K}$; if $\tau^2=\widetilde{\tau}^2$, the previous results in Theorem 2 hold. That is, the noise variance is identifiable, and the existence of the noise does not affect our claim to the noiseless case. Due to page limit, we briefly sketch the proof in this response. First, by our Theorem 1, both $K$ and $\widetilde{K}$ are mean-square continuous. Then apply Lemma 1 in Tang, Zhang, Banerjee (JRSSB 2022), we conclude that if $\tau^2\neq\widetilde{\tau}^2$ then $K\not\equiv \widetilde{K}$. Similarly, if $\tau^2=\widetilde{\tau}^2$, Theorem 2, for noiseless case, follows.
> > >
> > > We are keen on weaving these enriched insights into our manuscript's discussion section. Your elevated score and unwavering support significantly encourage our ongoing commitment to this research.

---

### Author Rebuttal · Authors · 2023-08-09

We express our sincere gratitude to the reviewers for their meticulous examination of our manuscript and their insightful comments. Your observations and recommendations have significantly enhanced the quality and content of our paper, and we have learned much from your comments. In this global response, we will discuss some common interest and concern across all the comments, go through the new changes in the manuscript. And at the end, we will explain the detailed information about the figures in the one-page PDF.

**1. (Reviewer PY7Z, LXB2, j39y)** We wish to address the concern regarding the large data limit. Our theory is built on the asymptotic side, and as such, our results remain consistent as $n\to\infty$.

**2. (Reviewer PY7Z, dNVU, LXB2)** We noted the shared interest in extending our methods to other kernels, such as RQ and periodic kernels. These particular kernels lack the analytic spectral densities necessary for a theoretical analysis of Gaussian Processes, which would necessitate more complex mathematical approaches. While our current study does not encompass these kernels, we have initiated a new simulation in response to reviewer PY7Z's interest in the kernel $K(x,x')=\sum_{l=1}^pw_l\text{Matern}(x_l-x_l';\sigma^2_l,\alpha_l,\nu)$, where $x_l$ represents the $l$-th coordinate of $x$. Although the primary tool we used to prove equivalence of measures is not directly applicable in this context, we have formulated a conjecture that the microergodic parameter of each dimension remains identifiable, while other individual parameters may not be identifiable, along with simulation study to support this conjecture.

**3. (Reviewer dNVU, LXB2, j39y)** The importance of smoothness in our paper was another focal point. We would like to reiterate that the smoothness of a mixture kernel is influenced by the least smooth kernel within it. This property has direct implications for the conditions in our proof, and we've conducted new experiments to numerically verify Theorem 1.

**4. (Reviewer LXB2, j39y)** We agree that the limited performance of the Matérn kernel may affect the presentations and raise concern. In acknowledgment of this issue, we have conducted two new experiments: MNIST image recovery and prediction on Mauna Loa CO$_2$. For the MNIST dataset, Matérn kernels exhibit satisfactory performance in recovering the image. Mauna Loa CO$_2$ is a well known example that has been widely used on Matérn kernels and other mixture kernels. Interestingly, Matérn 3/2 performed better than Matérn 1/2 in our experiments. However, in Matérn mixture of $\nu=1/2,3/2,5/2$, adding Matérn 3/2 did not improve performance, and the mixture kernel's performance remained similar to Matérn 1/2. This finding further confirms our point. We appreciate your feedback, which has assisted in refining our study.

In response to the feedback received, we have made substantial changes to our manuscript to address the concerns raised. These revisions include the addition and clarification of terminology uncommon in the machine learning community. We have polished all the figures to enhance their clarity and presentation. While we are unable to submit a revised manuscript due to submission rules, the styles of the revised figures can be referred to the new figures included in the one-page PDF. Furthermore, we have expanded our literature review to include detailed information about previous work on the application of mixture kernels, such as the specific types of kernels used in various analyses, and the identifiability work concerning Matérn kernels. This added context should strengthen the connection between our work and existing research. We also included 6 new experiments.

Our one-page PDF includes the visual presentation of our newly conducted experiments. We conducted four new simulations and two real data applications.

**Figure 1 (Reviewer PY7Z)** Simulation 2 with larger sample sizes, where we added experiments with sample sizes of n=1000,3000 to validate our conclusions in much larger sample case, thus reinforcing our results.

**Figure 2 (Reviewer PY7Z)** New simulation concerning the kernel $K(x,x')=\sum_{l=1}^p w_l\text{Matern}(x_l-x_l';\sigma^2_l,\alpha_l,\nu)$, leading us to conjecture that microergodic parameters for each dimension are identifiable while all other parameters remain unidentifiable.

**Figure 3,4 (Reviewer PY7Z,dNVU, LXB2, j39y)** Mauna Loa CO$_2$ data in 2022 (smoother) and MNIST data (non-periodic), as suggested by reviewers. Our experiments support our claim that the performance of mixture kernel is determined by the least smooth component rather than the best-performed component.

**Figure 6 (Reviewer dNVU, LXB2, j39y)** New simulation to support Theorem 1, allowing us to confirm that Matérn 3/2 is continuous and differentiable, while Matérn 1/2 and the Matérn mixture 1/2+3/2 are continuous but not differentiable.

**Figure 5,7 (Reviewer LXB2, j39y)** Simulation 2 with the L-BFGS optimizer, responding to reviewer concerns by verifying our theorem across various optimizers and learning rates, thus reinforcing our confidence in the results. Additionally, following the reviewers' suggestions, two new panels about $w_2\sigma_2^2\alpha_2^{2\nu_2},w_3\sigma_3^2\alpha_3^{2\nu_3}$ were added to the figures of all simulations to better substantiate our point that only the microergodic parameter for the least smooth component is identifiable.

We have undertaken extensive work in this rebuttal and devoted significant effort to refining our manuscript. We hope that the newly added explanation and experiments could answer your questions and concerns. We would be more than happy to discuss more in the discussion period. Once again, we thank the reviewers for their thoughtful examination and invaluable feedback. Your contributions have not only enhanced our current work but also promise to influence and inspire further research in this field.

---

### Decision · Program_Chairs · 2023-09-21

**Decision:**

Accept (poster)

**Comment:**

Three reviewers recommend borderline accept, weak accept or accept. Reviewer LXB2 recommends Reject. The reviewer disputes the definition of equivalence used in the paper and its practical use in ML. The authors have sufficiently clarified in the rebuttal that their study applies to the infinite domain and that future work can consider the more difficult setting, the finite case, that the reviewer discusses. I believe the paper makes a sufficient contribution to the state of the art of identifiability of GPs and they have backed up their claims with extensive empirical studies. Although the reviewer's concern is valid, I do not believe it guarantees the rejection of the paper. Still, instead, I ask the authors to include such discussion in the supplemental material for the paper.